# Compress then Serve:
# Serving Thousands of LoRA Adapters with Little Overhead

**Rickard Brüel Gabrielsson** [1]   **Jiacheng Zhu** [1]   **Onkar Bhardwaj** [2]   **Leshem Choshen** [1 2]   **Kristjan Greenewald** [2]
**Mikhail Yurochkin** [2]   **Justin Solomon** [1]

## Abstract

Fine-tuning large language models (LLMs) with low-rank adaptations (LoRAs) has become common practice, often yielding numerous copies of the same LLM differing only in their LoRA updates. This paradigm presents challenges for systems that serve real-time responses to queries that each involve a different LoRA. Prior works optimize the design of such systems but still require continuous loading and offloading of LoRAs, as it is infeasible to store thousands of LoRAs in GPU memory. To mitigate this issue, we investigate the efficacy of compression when serving LoRAs. We propose a method for the joint compression of LoRAs into a shared basis paired with LoRA-specific scaling matrices. We extend our algorithm to learn clusters of LoRAs that are amenable to joint compression, allowing it to scale gracefully to large LoRA collections. Our experiments with up to 1000 LoRAs demonstrate that compressed LoRAs preserve performance while offering major throughput gains in realistic serving scenarios with over a thousand LoRAs, maintaining 80% of the throughput of serving a *single* LoRA.

## 1. Introduction

The myriad uses for foundation models (FMs) have led to a proliferation of specialized models, each fine-tuned to perform a downstream task. To avoid fine-tuning foundation models with billions of parameters, parameter-efficient fine-tuning (PEFT) algorithms were proposed. An especially successful PEFT method is low-rank adaptation (LoRA) (Hu et al., 2021), which learns low-rank additive changes to neural network matrices. Because of the

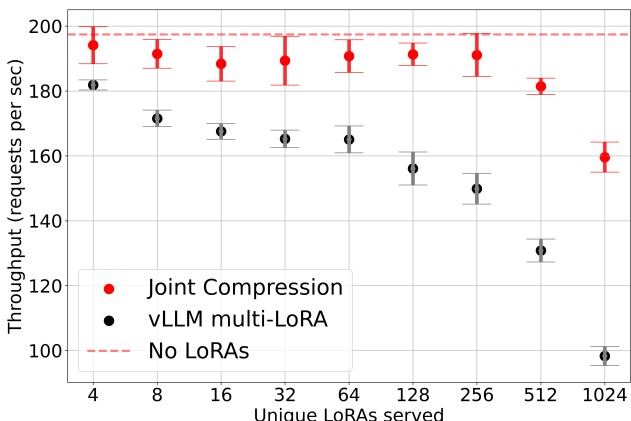

Figure 1: Throughput gains when serving 1000s of compressed LoRAs with vLLM.

low-rank parameterization, these matrices (called adapter weights) contain orders of magnitude fewer parameters than the base model. Still, LoRA can achieve performance on par with full fine-tuning (Hu et al., 2021).

LoRA's popularity has triggered a growing need to serve large collections of LoRA adapters at scale. Proprietary and open-source LLM providers offer fine-tuning services (OpenAI, 2024; TogetherAI, 2024; Predibase, 2024) with user bases likely in the thousands or even hundreds of thousands. As each user wants to use their own fine-tuned version of the LLM, serving a dedicated fine-tuned LLM per user becomes infeasible. To this end, S-LoRA (Sheng et al., 2023) proposes a system where only the base LLM is placed on an inference server and individual LoRA adapters are switched as needed at inference time. S-LoRA optimizes the system's inner workings via custom CUDA kernels and memory management to increase throughput when serving multiple LoRAs. Multi-LoRA system design has also been adopted in vLLM (Kwon et al., 2023), a state-of-the-art LLM serving engine. Despite optimized system designs, serving LoRAs still has a fundamental limitation: when the number of adapters is large, they need to be constantly loaded and offloaded from GPU memory to accommodate incoming requests, degrading throughput.

The problem of accommodating multiple LoRA adapters is also apparent when placing LLMs on edge devices,

---

*Equal contribution  [1]MIT CSAIL [2]MIT-IBM Watson AI Lab. Correspondence to: Rickard Brüel Gabrielsson <brg@mit.edu>.

*Proceedings of the 42$^{nd}$ International Conference on Machine Learning*, Vancouver, Canada. PMLR 267, 2025. Copyright 2025 by the author(s).

where smaller LLMs are fine-tuned for various tasks, and the adapters are swapped depending on the task at hand (Gunter et al., 2024). In this setting, the number of adapters is smaller, e.g., a few dozen (Gunter et al., 2024), but the memory constraints are also more stringent.

In this work, we consider the problem of compressing a collection of LoRAs. We have two key objectives: (1) preserving the performance of the original LoRAs and (2) improving the throughput of serving many LoRAs. We formulate LoRA compression as a reconstruction problem, where the goal is to approximate the original adapters via collections of matrices of a smaller size. We compress LoRAs *jointly* by finding a shared basis and LoRA-specific scaling matrices and propose a joint diagonalization-based algorithm (JD). To improve reconstruction error for large numbers of LoRAs while keeping the number of parameters in check, we propose a clustering approach where each cluster is compressed independently using the joint diagonalization algorithm. Our clustering algorithm is based on alternating between optimizing the cluster assignments and the per-cluster reconstruction error.

Figure 1 showcases the benefits of joint compression. When serving up to 64 unique LoRAs, we use JD without clustering and for 128 or more, we pick the number of clusters to match the performance of compressed and original LoRAs. In each case, the GPU memory footprint of the compressed and original LoRAs is matched for a fair comparison to vLLM's multi-LoRA inference engine. When serving over 1000 LoRAs, compression increases throughput $1.6\times$ and maintains 80% of the throughput of serving the base LLM (or a single LoRA merged into the LLM). §6 presents detailed results.

We summarize our main contributions below:

- We formulate the problem of compressing a collection of LoRAs and propose a joint compression scheme based on joint diagonalization.
- For large numbers of LoRAs, we scale joint compression by proposing a clustering algorithm where each cluster is jointly compressed to minimize reconstruction error.
- We establish theoretical guarantees for the reconstruction error of our compression formulation and relate reconstruction loss to performance empirically.
- We train a collection of more than 1000 high-quality LoRAs for `Mistral-7B-Instruct-v0.2` (Jiang et al., 2023a) on 1000 natural instruction tasks (Wang et al., 2022) and demonstrate that our compression techniques preserve the performance of the original LoRAs. We will release over a 1000 LoRAs to facilitate future work as well as the code for our method.
- We incorporate LoRA compression into a state-of-the-art LLM serving system and demonstrate that it is possible to serve over 1000 LoRAs across thousands of asyn-

chronous requests with throughput comparable to serving a *single LoRA*.

## 2. Related Work

Parameter-efficient fine-tuning (PEFT) has become prevalent for updating foundation models thanks to the need for efficiency in training and communication (Lialin et al., 2023). Many PEFT methods have been proposed, e.g. (Houlsby et al., 2019; Liu et al., 2022b) and LoRA (Hu et al., 2021) became the standard, partially due to the ease of switching between LoRAs in inference time.

Several works improve LoRA (Liu et al., 2024; Wang et al., 2024), sometimes with algebraic methods like SVD (Meng et al., 2024; Zhang et al., 2023; Jiang et al., 2023b) or by leveraging its statistical properties (Zhu et al., 2024; Zeng & Lee, 2024). Relatively few, however, accelerate inference times. S-LoRA (Sheng et al., 2023) provides an efficient means of switching between LoRAs. Wen & Chaudhuri (2024) adapt training to reduce batch multiplications, accelerating inference. Our method achieves a similar outcome (see Appendix D) without changing the LoRA formulation or requiring that LoRAs be trained in a dedicated way; future improvements to LoRA will also benefit from this aspect of our work (e.g., Meng et al. (2024)).

Punica (Chen et al., 2023) introduces Segmented Gather Matrix-Vector Multiplication (SGMV) to optimize multi-LoRA serving by parallelizing feature-weight multiplications in batches and grouping requests that use the same LoRA. Our approach, by contrast, reduces parameters as a means to serve multiple LoRAs efficiently, providing an orthogonal strategy that can be seamlessly integrated with Punica's methods to enhance performance. In our vLLM experiments, we leveraged the Punica kernel for multi-LoRA implementation, demonstrating the application of our method in conjunction with Punica's optimizations.

Other research proposes alternative PEFT methods that can be more parameter-efficient than LoRA. For example, VeRA (Kopiczko et al., 2024) fine-tunes LLMs by sharing global static parameters while learning local scaling variables; (IA)³ (Liu et al., 2022a) also reduces adapter parameter counts. However, none of these approaches has been as extensively tested or widely-adopted as LoRA. As a result, work that builds on LoRA enjoys a practical advantage due to its broad acceptance in practice.

There are many efforts to compress models (Cheng et al., 2017; Gholami et al., 2022; Sharma et al., 2024; Li et al., 2018). Predominantly, pruning and sparsification methods delete weights (Yadav et al., 2023a), and quantization methods reduce the weights' precision (Dettmers et al., 2024). Some works compress weights to reduce model size but typically require decompression and hence do not save

GPU memory (Hershcovitch et al., 2024). Similarly to our work, a few note increased performance and generalization after compression (Yadav et al., 2023a; Nadjahi et al., 2023; Hershcovitch et al., 2024; Sharma et al., 2024).

Our work also relates to model merging (Choshen et al., 2022; Wortsman et al., 2022; Matena & Raffel, 2021) and mixtures of experts (Muqeeth et al., 2024; Yadav et al., 2024). These methods reuse models trained by others (Choshen et al., 2023; Raffel, 2023), serving them together as one compressed model. Despite this similarity, these methods create a single general model that acts on any input, while ours yields more performant per-task solutions.

## 3. Rank-Based LoRA Compression

LoRA updates are parameterized by pairs of matrices $A$, $B$, whose product $BA$ updates the fixed weight matrices $W_0 \in \mathbb{R}^{d_B \times d_A}$ of a neural network foundation model. Given an input $x$ to a layer, the output of the LoRA-updated model at this layer is $(W_0 + BA)x$.

In formulating our compression algorithms, we consider a collection of given LoRA adapters $\{(A_i, B_i)\}_{i=1}^n$ that we would like to serve. We let $r_i$ refer to the rank of the LoRA adapter-pair $(A_i, B_i)$, i.e., $B_i \in \mathbb{R}^{d_B \times r_i}$, $A_i \in \mathbb{R}^{r_i \times d_A}$.

While our compression technique has access only to a collection of $\{(A_i, B_i)\}_{i=1}^n$ pairs, in our experiments we will assess the efficacy of compression by comparing how the compressed matrices perform relative to uncompressed LoRAs on typical data. For this reason, although in this section we optimize a Frobenius norm reconstruction error relative to the product $B_i A_i$, this is a proxy for the nonlinear and complex way that compression errors in the adapters impact transformer performance. Our experiments will thus focus on the performance of the compressed LoRAs against the uncompressed versions on real data in §6.

Our compression methods significantly reduce the overall number of parameters. Reducing parameters theoretically accelerates storage and serving of a collection of LoRAs. This reduction, however, alters the computational dynamics during inference, so parameter reduction alone does not immediately imply faster throughput. In light of the complexities of GPU optimization, we experimentally assess throughput under realistic conditions in §6.4.

### 3.1. Joint Diagonalization

To scale to many LoRAs, the compressed number of parameters should not scale linearly with $n$. Hence compressing each LoRA individually (e.g., via SVD as in our experimental baselines) is inherently limited. To address this, we suggest a Joint Diagonalization (JD) method, which optimizes a shared basis onto which we can project the set of

$n$ LoRAs. This allows structure to be shared, implicitly grouping and/or merging the collection of LoRAs.

In this model, each LoRA product $B_i A_i$ is factorized into the form $U \Sigma_i V$, where $U$ and $V$ are shared across all LoRAs and $\Sigma_i$ is specific to each LoRA. In this formulation, every $\Sigma_i$ shares the same rank $r$. This allows $U$ and $V$ to be pre-loaded onto the GPU, with $\Sigma_i$ loaded when necessary for each batch. The matrices $\Sigma_i$ can be either diagonal or small square matrices, thus significantly reducing the number of LoRA-specific parameters and accelerating multi-LoRA serving.

**Objective function.** Motivated by the relationship of singular value decomposition to minimizing the Frobenius norm of the reconstruction error, we also propose to minimize the Frobenius norm of the adapter matrix approximation error. Specifically, we use the following objective:

$$\min_{\{\Sigma_i\}_{i=1}^n, U, V} \sum_{i=1}^n \|B_i A_i - U \Sigma_i V^\top\|_{\text{Fro}}^2. \quad (1)$$

Note this problem is *not* solved by a single matrix SVD, since $U$ and $V$ are shared among all terms but the $\Sigma_i$'s are not. Using the Frobenius norm has the added benefit of making the objective convex in each argument separately, suggesting the possibility of efficient optimization. This objective function is underdetermined, however, so we consider two constrained regimes below.

**Full $\Sigma_i$ approximation.** The first method we call JD-Full. Without loss of generality, $U$ and $V$ can be constrained to be orthogonal, so long as $\Sigma_i$ remains an unconstrained full matrix. JD-Full adopts this restriction to make the optimization better posed, but note it does not restrict the expressiveness of the objective equation 1. This setting yields the following optimization problem:

$$
\begin{aligned}
&\text{JD-Full}_r(\{B_i A_i\}_{i=1}^n) = \\
&\underset{\substack{\{\Sigma_i\}_{i=1}^n \\ U^\top U = V^\top V = I_r}}{\arg\min} \sum_{i=1}^n \|B_i A_i - U \Sigma_i V^\top\|_{\text{Fro}}^2
\end{aligned}
$$

$$\text{(JD-Full)} \quad (2)$$

An efficient alternating algorithm to optimize this objective function can be found in Appendix A.

**Diagonal $\Sigma_i$ approximation.** As an alternative, we can leave $U$, $V$ unconstrained (other than to have $r$ columns) and instead constrain the matrices $\Sigma_i$ to be diagonal (but not necessarily positive). This formulation yields the following optimization problem:

$$\text{JD-Diag}_r(\{B_i A_i\}_{i=1}^n) =$$

$$\underset{\{\Sigma_i\}_{i=1}^n, U, V}{\text{argmin}} \sum_{i=1}^n \|B_i A_i - U \text{diag}(\Sigma_i) V^\top\|_{\text{Fro}}^2$$

$$\text{(JD-Diag)} \quad (3)$$

Appendix A provides an efficient alternating least squares algorithm for this objective. This diagonal version has per-LoRA parameter savings when compared to JD-Full, since the diagonal $\Sigma_i$ only needs $r$ parameters instead of $r^2$.

### 3.2. Clustering

As the number of LoRAs $n$ grows and becomes more diverse, the rank $r$ needed for Joint Diagonalization to achieve good performance will tend to increase. This increases the size of each $\Sigma_i$ that needs to be stored, especially for JD-Full which will require $O(nr^2)$ storage for these matrices. If the necessary $r$ grows proportionally to $n$, then this storage will eventually become the bottleneck.

To resolve this limitation with very large $n$, we propose to group the $n$ LoRAs into $k$ clusters $C_j$. Each cluster is given its own rank $r$ for JD compression, and the clusters are chosen such that the overall reconstruction error is minimized. Specifically, the overall objective is

$$\min_{\{\{C_j\}, U_j, V_j\}, \{\Sigma_i\}} \sum_j \sum_{i \in C_j} \|B_i A_i - U_j \Sigma_i V_j\|_F^2,$$

optimized by alternating between cluster assignments and the JD of each cluster; Appendix A.3 provides details. Typically, the goal with large $n$ is to have $k$ grow with $n$ as $r$ becomes fixed. Comparing $k$ rank-$r$ JD-Full clusters to a rank-$kr$ JD-Full single cluster compression, the clustered approach requires $O(dkr + nr^2)$ parameters, while the single-cluster approach requires $O(dkr + nk^2r^2)$ parameters due to the increased sizes of the $\Sigma_i$s. While these two approaches have the same rank, they may have different reconstruction abilities. Empirically, we find that multiple clusters significantly aid performance for $n \geq 100$.

## 4. Theoretical Analysis

In this section, we seek to better understand the role of the joint diagonalization method in §3.1 and how it motivates the clustering approach. We focus on the full-$\Sigma_i$ case with orthogonal $U, V$ matrices. Note that, for the same $r$, the $r$-JD-Diag has at least as large reconstruction error as $r$-JD-Full since it imposes an additional constraint on the $\Sigma_i$.

Firstly, note that perfect reconstruction can be achieved if and only if $r$ is large enough, since there exist $U, V$ such that all the $B_i, A_i$ are in the spans of $U, V$ resp. if and only if $r \geq \tilde{r}$:

**Proposition 1.** *Suppose* $\text{rank}(B_i A_i) = r_i$ *for all* $i$, *and let*

$$\tilde{r} = \max\left\{\text{rank}([A_1, \ldots, A_n]), \text{rank}([B_1^\top \ldots, B_n^\top])\right\}.$$

*Note* $\max_i r_i \leq \tilde{r} \leq \sum_{i=1}^n r_i$. *Then JD-Full (equation 2) with* $r = \tilde{r}$ *compresses losslessly (perfect reconstruction), while* $r < \tilde{r}$ *will give nonzero reconstruction error.*

Due to training noise, $\tilde{r}$ will equal $\sum_{i=1}^n r_i$ almost always. This implies that in most realistic settings, the joint diagonalization approach is a lossy reconstruction.

This reconstruction loss can be significant, as the following theorem shows (proved in Appendix B):

**Theorem 1.** *Consider* $n$ *LoRAs* $\{A_i, B_i\}_{i=1}^n$ *with* $r, n \leq d^2$, *and form the matrix* $L = \begin{bmatrix} \text{vec}(B_1 A_1) & \cdots & \text{vec}(B_n A_n) \end{bmatrix}$. *Let* $\sigma_j$ *be the singular values of* $L$, *sorted from largest to smallest, and let* $\bar{\sigma}_j$ *be the singular values of* $\sum_{i=1}^n B_i A_i$. *Then, using JD-Full (equation 2),*

$$\sum_{j=1}^r \bar{\sigma}_j^2 \leq \sum_{i=1}^n \|\Sigma_i\|_{\text{Fro}}^2 = \sum_{i=1}^n \|U \Sigma_i V^\top\|_{\text{Fro}}^2 \leq \sum_{j=1}^{\min(r^2, n)} \sigma_j^2,$$

*implying the sum of squared Frobenius norms of the reconstructed LoRAs satisfies*

$$\frac{\sum_{i=1}^n \|U \Sigma_i V^\top\|_{\text{Fro}}^2}{\sum_{i=1}^n \|B_i A_i\|_{\text{Fro}}^2} \leq \frac{\sum_{j=1}^{\min(r^2, n)} \sigma_j^2}{\sum_{j=1}^n \sigma_j^2} \leq 1, \text{ and}$$

$$\frac{\sum_{i=1}^n \|U \Sigma_i V^\top - B_i A_i\|_{\text{Fro}}^2}{\sum_{i=1}^n \|B_i A_i\|_{\text{Fro}}^2} \geq 1 - \frac{\sum_{j=1}^{\min(r^2, n)} \sigma_j^2}{\sum_{j=1}^n \sigma_j^2}.$$

In other words, reconstruction error is unavoidable if $L$'s singular values are not concentrated in the top $r^2$ entries.

**Remark 1** (Lower bound and merging). *The lower bound* $\sum_{j=1}^r \bar{\sigma}_j^2$ *could be achieved by setting all the* $\Sigma_i$ *equal, i.e., using a fully merged model instead of only merging the subspaces* $U, V$ *and allowing* $\Sigma_i$ *to vary with* $i$.

**Remark 2** (Upper bound and grouping). *The upper bound is smallest when the LoRAs are relatively* clustered, *i.e., when groups of vectors* $\text{vec}(B_i A_i)$ *are similar. This situation raises the magnitude of the largest singular values of* $L$, *raising the upper bound in the proposition. As the LoRAs are* $d \times d$ *matrices that can be thought of as points in* $\mathbb{R}^{d^2}$, *for typical values of* $d$ *well into the hundreds, it is likely that unrelated LoRAs will be* unclustered, *i.e., they will have relatively low inner products with each other.*

For orthogonal LoRAs, the singular values of $L$ are the norms of the LoRAs, suggesting the following corollary:[1]

---

[1] A result for isotropic Gaussian LoRAs could be obtained via the quantiles of the Marchenko-Pastur Law.

**Corollary 1.** *Suppose (e.g., due to normalization) that the inputs to the joint diagonalization algorithm all have unit Frobenius norm, i.e., $\|B_i A_i\|_{\mathrm{Fro}} = 1$. Moreover, assume that the LoRAs are all orthogonal in the sense $\mathrm{tr}((B_i A_i)(B_j A_j)^\top) = 0$ for $i \neq j$. Then, using the JD-Full method equation 2, we have $1 \leq \sum_{i=1}^{n} \|\Sigma_i\|_{\mathrm{Fro}}^2 \leq \min(r^2, n)$, implying that the sum of squared Frobenius norms of the reconstructed LoRAs satisfies*

$$1 - \frac{1}{n} \geq \frac{\sum_{i=1}^{n} \|U\Sigma_i V^\top - B_i A_i\|_{\mathrm{Fro}}^2}{\sum_{i=1}^{n} \|B_i A_i\|_{\mathrm{Fro}}^2} \leq 1 - \min\left(\frac{r^2}{n}, 1\right).$$

This implies that for the common setting where $r^2 \ll n$, the reconstructed LoRAs will be significantly smaller than the original LoRAs, with significant reconstruction error.

Our analysis illustrates the tradeoffs of joint diagonalization. If the LoRAs are similar or well-clustered, reconstruction error will be low. On the other hand, if the LoRAs are random and orthogonal, reconstruction error will be high.

Since the loss space of transformers is highly complex, increasing reconstruction error does not necessarily degrade LLM performance. Interestingly, Figure 3 below shows that while large reconstruction error rapidly decreases performance, moderate (but still relatively large, at around 60%) reconstruction error does not damage performance and may even slightly outperform the zero-error setting. At the same reconstruction error, clustering outperforms non-clustering. This motivates our focus on minimizing reconstruction error, while also suggesting that our approach achieves something deeper than compression. Specifically, joint diagonalization finds subspaces that are shared among many LoRAs when $r$ is large and *merges* subspaces when $r$ is small. When $r$ is particularly small, this tendency towards *averaging* all or some of the LoRAs connects to *merging* LoRAs, whose empirical success (Shah et al., 2023; Huang et al., 2024) could explain the procedure's success despite the nonlinearity of transformers.

Appendix H.11 explores this idea further, comparing reconstruction of real-world LoRAs to reconstruction of randomly sampled LoRAs. The reconstruction error is generally large, but significantly lower than the reconstruction error for random noise, indicating that a major shared component between the LoRAs is successfully retained.

That said, as the number of LoRAs grows, the shared component may not be significant enough to maintain sufficiently low reconstruction error with low rank $r$. This motivates the introduction of *clustering* in §3.2, since clustering seeks to find groups of LoRAs that are similar and better compressible by joint diagonalization. In particular, if the number of clusters $k$ grows with $n$, the reconstruction error may no longer degrade with $n$ even when $r$ is fixed.

In the extreme case where $k = n$, each LoRA is com-

pressed independently. By the Eckart-Young Theorem, JD applied to a single LoRA reduces to an SVD, replacing each rank-$r_i$ LoRA adapter $B_i A_i$ with a reduced rank-$r$ approximation, where typically $r < \frac{1}{n} \sum_{i=1}^{n} r_i$:

$$\mathrm{SVD}_r(B_i A_i) = U_i \Sigma_i V_i^\top, \quad \forall i = 1, \ldots, n. \quad (4)$$

As $\Sigma_i V_i^\top$ can be saved as a single matrix, this approach has $rn(d_A + d_B)$ parameters. We refer to this $k = n$ method as r-SVD and find that it underperforms our other methods while slightly outperforming the baseline uncompressed LoRAs. This result parallels Jiang et al. (2023b)'s observation that lowering LoRA ranks is beneficial for multi-task learning and model merging.

## 5. Training & Performance Evaluation

### 5.1. Training

We trained LoRA adapters on 1000 natural instruction tasks (Wang et al., 2022) using `Mistral-7B-Instruct-v0.2` (Jiang et al., 2023a) as the base. We set all LoRA adapter ranks to 16 (i.e., $\forall i, r_i = 16$), except for those in our ablation study (Appendix H.1), where we vary the LoRA rank.

We selected 10 diverse tasks (Table 2 in Appendix C) manually for consistent evaluation across experiments and randomly sampled an additional 990 tasks, resulting in a total of 1000 tasks (Table 3). The tasks went through a robust reviewing protocol to ensure high quality and diversity. Each task data was divided into training, validation, and test sets.

Hyperparameters, such as early stopping, were tuned using the validation sets. Table 1, Appendix C shows that on the test sets, LoRA consistently outperformed the base model in terms of Rouge scores and loss metrics.

### 5.2. Evaluation

We evaluated multiple metrics for the natural instruction tasks, including cross-entropy loss, Rouge-1, Rouge-L (Lin, 2004), exact match, and *agreement* between uncompressed and compressed LoRA. Here, *agreement* measures the exact match in task-generations between the uncompressed LoRA model and the compressed LoRA model, rather than comparing to ground truth data. While detailed results and discussions for all metrics are provided in Appendix H, our primary focus in the main text is on Rouge-L. We find that all metrics correlate, but Rouge-L correlates most strongly with downstream utility. This finding aligns with prior work (Wang et al., 2022), which demonstrates that Rouge-L correlates well with classification accuracy.

While cross-entropy is used for optimization during training, identical generation outputs across models can yield different cross-entropy losses. Exact match is too rigid and

does not account for the variability in task responses. Similarly, agreement does not capture the inexactness associated with most of our tasks, nor does it account for the performance gains or losses of the compressed LoRAs. Arguably, practitioners are primarily concerned with task performance in the settings for which the LoRA was designed, rather than exact generational agreement between models.

Joint diagonalization optimizes reconstruction error measured by the Frobenius norm, bounded by our theoretical analysis in §4. We empirically study the relation between the reconstruction error and downstream Rouge-L performance in Section 6.2.

Instead of listing absolute performance, we compute the performance difference between the base model and the LoRA model for each task via the ratio

$$\text{Performance relative to LoRA} := \frac{\text{method-performance}}{\text{LoRA-performance}}$$

for the method in question, highlighting relative improvement wrt the uncompressed LoRAs.

# 6. Experiments

## 6.1. Task Performance

For each method, we vary the number $n$ of compressed LoRAs and the compression rank $r$. We run each experiment three times with different random seeds and report the mean and standard deviation. See Table 7 for results evaluated on the same ten manually-selected tasks (Table 2) across settings. Every compressed collection of LoRAs contains these 10 tasks (i.e., in-distribution tasks), and each collection contains the smaller collections as subsets.

We normalize each LoRA adapter to have a Frobenius norm of one prior to running joint diagonalization. This normalization enhances performance and reduces the variance in reconstruction error. We restore the original norms of the LoRA adapters before reconstruction and testing.

Figure 2 illustrates the Rouge-L scores of the compressed LoRAs divided by the Rouge-L scores of the uncompressed LoRAs. JD variants often increase generalization and outperform the original LoRA. Notably, our JD methods approach the compression efficacy of a single LoRA, and with clustering, this aggressive reduction in size also maintains performance in larger collections. Appendix H includes tables of additional relative and absolute metrics.

For efficiency, we limited the JD methods to ten iterations instead of full convergence. While the alternating algorithm quickly reaches an approximate minimizer, squeezing out the last few digits of precision takes many more iterations with limited to no performance gain. Appendix H.12 also evaluates an alternative iterative algorithm that con-

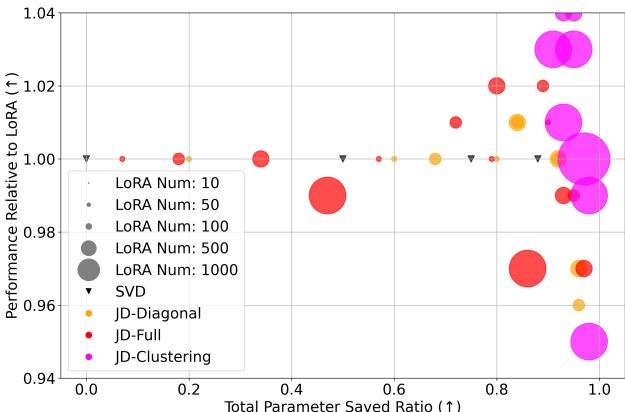

Figure 2: Performance after compression. We compare the performance of compressed LoRAs relative to uncompressed ones, with higher values on both axes reflecting better performance. The Total Parameter Saved Ratio depicts the number of parameters saved for a system with a large number $n$ of different LoRAs. It is computed as: $r_{total} := 1 - \frac{\text{num. parameters after compression}}{\text{num. parameters before compression}}$.

verges more rapidly once $U, V$ are close to a minimizer, with minimal performance differences.

## 6.2. Performance and Reconstruction Error

Figure 3 relates reconstruction error and performance. The $y$-axis measures the mean performance improvement of Rouge-L relative to uncompressed LoRA, and the $x$-axis quantifies the mean relative reconstruction error between the compressed reconstruction of the product $BA$ and the original product $BA$. Although performance and reconstruction error relate non-linearly, we see a decreasing, somewhat exponential trend. Notably, minimizing reconstruction error does not yield optimal performance, indicating that mild lossy reconstruction may enhance generalization. Interestingly, under the clustering approach, compared to non-clustering, even more aggressive lossy reconstruction can outperform less lossy reconstruction, suggesting that reconstruction error is even less critical for performance in the clustering scenario.

To select hyperparameters (compression rank and number of clusters) for the clustering experiments, we first assessed reconstruction error on a single LoRA module over a range of settings (see Appendix G). These preliminary experiments enabled efficient selection of cluster counts and rank values for compressing all LoRA modules.

## 6.3. Benefits of Compression

Compressing LoRAs reduces their parameter counts, thus lowering their overall memory footprint. While this offers

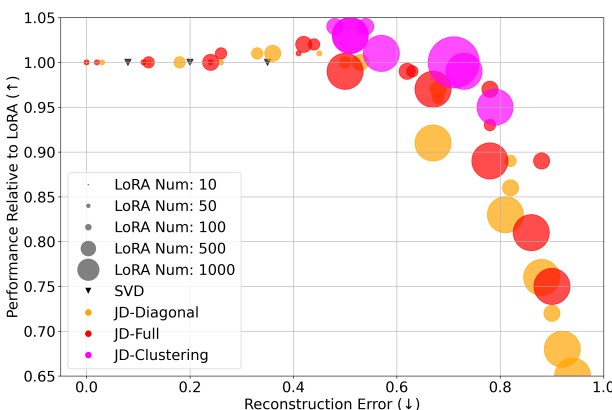

Figure 3: Reconstruction error vs. performance.

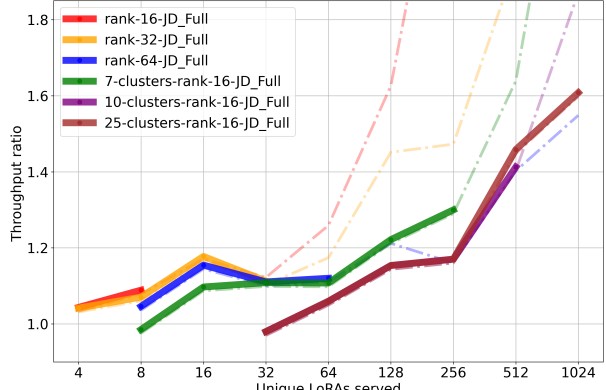

Figure 4: Throughput ratio when serving varying numbers of LoRAs with vLLM. Highlighted settings preserve at least 99% of the uncompressed LoRA performance.

many benefits, in both training and inference scenarios parameters are often transferred between different memory hierarchies (e.g., from CPU to GPU), and these transfers usually scale linearly with the amount of data moved. At the same time, compressed LoRAs alter the forward-pass formulation (unless merged with the base model weights). As shown in Figure 5 in the Appendix E, although compression greatly reduces memory usage and transfer time, it does not affect the forward-pass latency. Of the various ways to leverage these improvements, this work focuses on optimizing inference for multiple LoRAs using vLLM.

### 6.4. Throughput of Serving Compressed LoRAs

The previous sections demonstrate how to select an appropriate joint compression setting guided by the reconstruction error, such that the performance of the original LoRAs is preserved. Naturally, the rank and/or the number of clusters for the compression needs to increase as we compress larger LoRA collections to match LoRA performance.

Figure 4 studies how throughput with various compression settings compares to the vLLM multi-LoRA throughput with the matched GPU memory footprint. Specifically, for each number of unique LoRAs served and each compression setting, we compute the corresponding number of LoRAs to be placed on the GPU during serving and report the ratio of the two throughputs. For example, when serving 64 unique LoRAs and using rank 64 JD-Full compression, we report the ratio of throughputs of rank 64 JD-Full and vLLM multi-LoRA with 6 LoRAs allowed on the GPU at a time (see Appendix F for details). As the number of unique LoRAs increases, vLLM multi-LoRA throughput degrades as it needs to schedule the requests and load and offload the adapters. We note that vLLM multi-LoRA already employs advanced optimizations, such as efficient scheduling and non-blocking CPU-GPU communication when swapping LoRAs as well as techniques introduced in S-LoRA (Sheng et al., 2023; Kwon et al., 2023), but system opti-

mization alone is insufficient to mitigate throughput degradation when serving many LoRAs.

Figure 4 shows that across LoRA collection sizes our compression techniques improve the throughput of vLLM multi-LoRA. Additionally, we highlight regions for each compression setting where compression is sufficiently moderate to achieve 99%+ of LoRA performance, according to the results in §6.2. Compression with a larger rank or too many clusters does not improve baseline throughput when serving a smaller number of LoRAs and should not be used in such cases. For example, rank 16 JD-Full improves baseline throughput with 4 and 8 LoRAs, but will underperform with more LoRAs, while 25 clusters rank 15 JD-Full does not improve throughput with 32 or fewer LoRAs, but when serving 1000+ LoRAs it improves the throughput significantly while maintaining the performance. Overall, an appropriate joint compression setting improves vLLM multi-LoRA throughput and preserves performance for LoRA collections of any size between 4 and 1024, as in Figure 1. Appendix F provides compression settings for each collection size.

vLLM extensively uses custom CUDA kernels. To accommodate our compression techniques, we minimally adjusted the vLLM code to generate additional kernels needed by the compressed LoRAs and used the Punica (Chen et al., 2023) kernel to further accelerate matrix multiplication. Pseudocode is given in §F.4 to show how we use the batch multiplication kernel. There likely is room for improvement to optimize the newly added kernels.

**Additional details.** In this experiment, we considered a varying number of rank-16 LoRAs, using a dataset of Shakespeare sonnets as inputs[2] arriving *asynchronously*.

[2]https://www.kaggle.com/
datasets/shivamshinde123/
william-shakespeares-sonnet/data

We measured throughput, i.e., the number of requests served per second when generating ten tokens per request. The base was Mistral 7B Instruct; we simulated random LoRAs and assigned inputs to LoRAs at random. Experiments were conducted on H100 80GB GPU capped at 40% memory consumption to reflect situations where a service provider might want to serve many LoRAs from cheaper hardware with lower memory than higher-end GPUs. This setting applies to the scenario where the LLM is large compared to the size of GPU and yet a provider may want to serve many LoRAs efficiently using one device.

### 6.5. Recommendations

JD-Full is generally preferred over JD-Diag, although for smaller numbers of LoRAs (less than 100), the performance difference is negligible. While JD-Full alone is effective up to 100 LoRAs, incorporating clustering at scales of 500-1000 LoRAs significantly enhances performance.

We recommend the following procedure for hyperparameter selection. For $\leq$ 100 LoRAs, JD-Full can be used without substantial degradation, using a rank $\approx$ (number of LoRAs/2) + 7. Beyond 100 LoRAs, clustering becomes increasingly critical. A robust method for any number of LoRAs up to 1000 uses JD-Full with clustering. Specifically, select a LoRA module from the middle of the network, apply a compression rank of 16, and experiment with an exponentially increasing number of clusters. Compute the reconstruction error for each setting on this module across all LoRAs—a computationally efficient process. Choose the minimal number of clusters that achieves a reconstruction loss below 0.6, and then use these settings across LoRA modules. Figure 6 in the Appendix illustrates this procedure applied to 500 LoRAs.

Tuning hyperparameters as discussed above using reconstruction loss as a validation metric is convenient since it can be done efficiently on CPU without expensive LLM evaluation. As our experiments demonstrate, compression settings that achieve below 0.6 reconstruction loss reliably preserve 99% or more of the LoRA performance, sometimes even outperforming the original LoRAs.

For inference, this procedure is executed as a preprocessing step before deploying our inference server. As new LoRAs are submitted, they are initially served uncompressed. A background CPU job can periodically re-run the compression algorithm and update the served LoRA parameters with the compressed versions.

## 7. Discussion

This study introduces approaches to LoRA compression, addressing significant challenges emerging as customization of foundation models such as LLMs and diffusion models becomes increasingly popular. Our contributions include theoretical formulations, empirical validation, and practical implementations that enhance the understanding and application of LLMs in scalable environments.

Our findings have several implications. Our theoretical bounds on reconstruction error not only increase confidence in the use of compressed models but also lay a groundwork for future explorations. Demonstrating that our compression techniques can preserve up to 100% of the original LoRAs' performance highlights the effectiveness of our methods. Furthermore, integrating LoRA compression into state-of-the-art LLM serving systems demonstrates potential for resource optimization, with throughput for thousands of LoRAs nearing that of a single LoRA.

Our promising results suggest several future research directions. First, further compression may be possible via quantization, since joint-diagonalization and quantization are independent compression strategies. Second, when scaling to hundreds of thousands of LoRAs, joint compression, while effective, will be insufficient to fit all LoRAs onto the GPU, thus requiring a procedure to schedule the requests. Clustering offers opportunities for efficient scheduling that incorporates the cluster assignments of LoRAs corresponding to the incoming requests.

Privacy presents another research direction, particularly regarding the possibility of information leakage during joint compression. As a preliminary study, Appendix H.2 investigates whether a base model with an adapter $A$ for task $T_A$, after being jointly compressed alongside an adapter $B$ for task $T_B$, inadvertently improves on $T_B$. Such an outcome would indicate that adapter $A$ acquired information from adapter $B$. Our ablation study shows no performance gains on $T_B$, suggesting that the compressed adapter $A$ remains independent and does not leak—or gain—information from adapter $B$. A more detailed investigation of the privacy properties of joint compression is an interesting next step.

In conclusion, our research advances LLM deployment by providing robust, scalable, and efficient compression. The ability of compressed LoRAs to maintain high performance while saving resources opens avenues for the broad application and adoption of LLMs across various industries. We encourage the community to build upon our findings and shared LoRAs to further enhance these technologies.

## Impact Statement

This paper presents work whose goal is to advance machine learning. There are no societal consequences of our work that we feel must be specifically highlighted here.

## Acknowledgements

The MIT Geometric Data Processing Group acknowledges the generous support of Army Research Office grants W911NF2010168 and W911NF2110293, of National Science Foundation grant IIS2335492, from the CSAIL Future of Data program, from the MIT–IBM Watson AI Laboratory, from the Wistron Corporation, and from the Toyota–CSAIL Joint Research Center.

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

# A. Joint Diagonalization Algorithms

## A.1. Alternating Methods

Our goal is to derive algorithms that optimize equation 1. Common to both methods, we expand the objective functional:

$$
\sum_i \|B_i A_i - U\Sigma_i V^\top\|_{\text{Fro}}^2 = \sum_i \text{tr}((B_i A_i - U\Sigma_i V^\top)(B_i A_i - U\Sigma_i V^\top)^\top) \text{ by definition}
$$

$$
= \sum_i \left[ \text{tr}(B_i A_i A_i^\top B_i^\top) - 2\text{tr}(B_i A_i V\Sigma_i^\top U^\top) + \text{tr}(U\Sigma_i V^\top V\Sigma_i^\top U^\top) \right]
$$

$$
= \text{const.} - 2\sum_i \text{tr}(B_i A_i V\Sigma_i^\top U^\top) + \sum_i \|U\Sigma_i V^\top\|_{\text{Fro}}^2. \tag{5}
$$

Using this expansion, we now consider the two settings discussed in §3.1.

**Case 1: Non-diagonal $\Sigma_i$, orthogonal $U, V$.** Setting the derivative of equation 5 with respect to $\Sigma_i$ to zero, we find

$$
\Sigma_i = \Sigma_i^*(U, V) = U^\top B_i A_i V. \tag{6}
$$

We simplify our objective function after plugging in this expression:

$$
\sum_i \|B_i A_i - U\Sigma_i V^\top\|_{\text{Fro}}^2 + \text{const.} = \sum_i \left[ \|\Sigma_i\|_{\text{Fro}}^2 - 2\text{tr}(B_i A_i V\Sigma_i^\top U^\top) \right] \text{ from equation 5}
$$

$$
= \sum_i \left[ \text{tr}(U^\top B_i A_i VV^\top A_i^\top B_i^\top U) - 2\text{tr}(B_i A_i VV^\top A_i^\top B_i^\top UU^\top) \right] \text{ from equation 6}
$$

$$
= -\sum_i \text{tr}(B_i A_i VV^\top A_i^\top B_i^\top UU^\top).
$$

Substituting equation 6, we find

$$
U_{opt}, V_{opt} = \arg \max_{\substack{U^\top U = I \\ VV^\top = I}} \sum_{i=1}^n \|U^\top B_i A_i V\|_{\text{Fro}}^2 = \arg \max_{\substack{U^\top U = I \\ VV^\top = I}} \sum_{i=1}^n \|\Sigma_i^*(U, V)\|_{\text{Fro}}^2. \tag{7}
$$

Note that

$$
\sum_{i=1}^n \|U^\top B_i A_i V\|_{\text{Fro}}^2 = \text{tr}\left( \left( \sum_{i=1}^n B_i A_i VV^\top A_i^\top B_i^\top \right) UU^\top \right)
$$

$$
= \text{tr}\left( \left( \sum_{i=1}^n B_i^\top A_i^\top UU^\top A_i B_i \right) VV^\top \right),
$$

by the identity $\|A\|_{\text{Fro}}^2 = \text{tr}(A^\top A)$. Hence, we optimize equation 7 by alternating between $U$ and $V$:

- $U$ **iteration:** Define $M := \sum_i B_i A_i VV^\top A_i^\top B_i^\top$. Parenthesizing this expression properly requires only $O((m+n)r)$ storage/computation time. With this definition, we maximize $\text{tr}(MUU^\top)$ over $U$ satisfying $U^\top U = I$. Since $M$ is positive semidefinite, the optimum is to take $U$ to be the $r$ eigenvectors of $M$ with largest eigenvalue, equivalent to an SVD problem.
- $V$ **iteration:** Define $N := \sum_i A_i^\top B_i^\top UU^\top B_i A_i$. Similarly to the previous step, we take $V$ to contain the $r$ eigenvectors of $N$ with largest eigenvalue, again solvable using an SVD.

This method decreases the objective in each step.

**Case 2: Diagonal $\Sigma_i$.** If constrain $\Sigma_i$ to be diagonal, we interpret our objective function equation 1 as a "triple least

squares" problem. We compute gradients:

$$\nabla_U \sum_i \|B_i A_i - U\Sigma_i V^\top\|_{\text{Fro}}^2 = 2\sum_i (U\Sigma_i V^\top - B_i A_i)V\Sigma_i^\top$$

$$\nabla_V \sum_i \|B_i A_i - U\Sigma_i V^\top\|_{\text{Fro}}^2 = 2\sum_i (V\Sigma_i^\top U^\top - A_i^\top B_i^\top)U\Sigma_i$$

$$\nabla_{\Sigma_i} \sum_i \|B_i A_i - U\Sigma_i V^\top\|_{\text{Fro}}^2 = 2U^\top(U\Sigma_i V^\top - B_i A_i)V$$

These expressions suggest efficient $r \times r$ linear systems to solve for $U, V$:

$$U = \left(\sum_i B_i A_i V\Sigma_i^\top\right)\left(\sum_i \Sigma_i V^\top V\Sigma_i^\top\right)^{-1}$$

$$V = \left(\sum_i A_i^\top B_i^\top U\Sigma_i\right)\left(\sum_i \Sigma_i^\top U^\top U\Sigma_i\right)^{-1}.$$

For $\Sigma_i$, we extract the diagonal from our gradient above:

$$\begin{aligned}
\text{diag}(U^\top U\Sigma_i V^\top V)_j &= (U^\top U\Sigma_i V^\top V)_{jj}\\
&= \sum_m (U^\top U)_{jm}\Sigma_{imm}(V^\top V)_{mj}\\
&= (U^\top U \circ V^\top V)\text{diag}(\Sigma_i)\\
\text{diag}(U^\top B_i A_i V)_j &= \sum_m (U^\top B_i)_{jm}(A_i V)_{mj}\\
&= \sum_m (U^\top B_i)_{jm}(V^\top A_i^\top)_{jm}\\
&= (U^\top B_i \circ V^\top A_i^\top)\mathbf{1}\\
\implies \text{diag}(\Sigma_i) &= (U^\top U \circ V^\top V)^{-1}(U^\top B_i \circ V^\top A_i^\top)\mathbf{1}
\end{aligned}$$

Here $\circ$ denotes the Hadamard product.

Combining these expressions, we use a simple coordinate descent algorithm cycling between the following three steps:

1. Solve for $U$
2. Solve for $V$
3. Solve for the $\Sigma_i$'s
4. Optionally, normalize so $\sum_i \|\Sigma_i\|_{\text{Fro}}^2 = 1$

### A.2. Additional Eigenvalue Iteration Algorithm

For the first case in §A.1, we introduce an alternative algorithm that eschews the use of SVD. This alternative is optimized for GPU execution, enabling tractable runs to convergence.

To derive this algorithm, we employ Lagrange multipliers to formulate the derived objective from equation 7:

$$U_{opt}, V_{opt} = \arg\max_{\substack{U^\top U = I \\ VV^\top = I}} \sum_{i=1}^n \|U^\top B_i A_i V\|_{\text{Fro}}^2, \tag{8}$$

yielding the expression

$$\Lambda = -\frac{1}{2}\|U^\top B_i A_i V\|_{\text{Fro}}^2 - \frac{1}{2}\text{tr}(X^\top(I - U^\top U)) - \frac{1}{2}\text{tr}(Y^\top(I - V^\top V)). \tag{9}$$

Taking the derivatives gives

$$\nabla_U \Lambda = - \sum_i B_i (A_i V)(V^\top A_i^\top)(B_i^\top U) + UX \tag{10}$$

$$\nabla_V \Lambda = - \sum_i A_i^\top (B_i^\top U)(U^\top B_i)(A_i V) + VY \tag{11}$$

Setting these derivatives to zero shows

$$\sum_i B_i (A_i V)(V^\top A_i^\top)(B_i^\top U) = UX \tag{12}$$

$$\sum_i A_i^\top (B_i^\top U)(U^\top B_i)(A_i V) = VY. \tag{13}$$

Here, one can show that the Lagrange multiplier matrices $X$ and $Y$ are diagonal and nonnegative, since the problem reduces to an eigenvalue problem when either $U$ or $V$ is fixed; this is essentially the argument behind the alternating algorithm in Appendix A. Hence, taking inspiration from classical eigenvalue iteration, we use the following updates to improve our estimates of $U$ and $V$:

$$U_0^{(k+1)} \leftarrow \sum_i B_i (A_i V^{(k)})((V^{(k)})^\top A_i^\top)(B_i^\top U^{(k)}) \tag{14}$$

$$V_0^{(k+1)} \leftarrow \sum_i A_i^\top (B_i^\top U^{(k)})((U^{(k)})^\top B_i)(A_i V^{(k)}) \tag{15}$$

$$U^{(k+1)} \leftarrow \texttt{orthogonalize}(U_0^{(k+1)}) \tag{16}$$

$$V^{(k+1)} \leftarrow \texttt{orthogonalize}(V_0^{(k+1)}) \tag{17}$$

Here, the function `orthogonalize` orthogonalizes the columns of a matrix, e.g. by using the $Q$ part of the reduced-size $QR$ factorization. Although we lack a formal convergence proof, in practice we find that this method reliably reaches a local optimum of our problem.

By executing matrix operations in the specified sequence, these computations can be rapidly performed on GPUs. Note the expressions above are parenthesized to avoid constructing a large matrix product as an intermediate computation.

### A.3. Clustering algorithm

**Initialization**: We run joint diagonalization with a single $U, V$ then perform k-means with $k$ clusters on the space of $\Sigma_i$'s. This gives us our first clusters and we can use random initialization $U_j, V_j$ for each cluster but the $\Sigma_i$ can be maintained as initialization.

**Step 1**: Using the alternating JD algorithms from earlier in this section, we optimize the problem $\min_{U_j, V_j, \Sigma_i} \sum_{i \in C_j} ||B_i A_i - U_j \Sigma_i V_j^\top||_F^2$ for each $j$ independently.

**Step 2**: New cluster assignment for $i$ : $\min_j \min_{\Sigma_i} ||B_i A_i - U_j \Sigma_i V_j^\top||_F^2$. If any assignment changes we go to Step 1, else we have converged.

## B. Proof of Theorem 1

*Proof.* For the lower bound, note that by Jensen's inequality,

$$\sum_{i=1}^n \|U^\top B_i A_i V\|_{\text{Fro}}^2 \geq \left\| U^\top \sum_{i=1}^n B_i A_i V \right\|_{\text{Fro}}^2,$$

for any $U, V$. Hence,

$$\sup_{U,V \in \text{St}(k,d)} \sum_{i=1}^n \|U^\top B_i A_i V\|_{\text{Fro}}^2 \geq \sup_{U,V \in \text{St}(k,d)} \left\| U^\top \sum_{i=1}^n B_i A_i V \right\|_{\text{Fro}}^2. \tag{18}$$

By the definition of singular value decomposition, the right hand side of equation 18 is maximized with $U, V$ being the top $r$ singular vectors of $\sum_{i=1}^{n} B_i A_i$, yielding $\left\| U^\top \sum_{i=1}^{n} B_i A_i V \right\|_{\text{Fro}}^2 = \sum_{i=1}^{r} \bar{\sigma}_i^2$. Recalling that $\Sigma_i = U^\top B_i A_i V$ yields the lower bound.

For the upper bound, recall that $\Sigma_i = U^\top B_i A_i V$. Rearranging,

$$\text{vec}(\Sigma_i) = (V^\top \otimes U^\top)\text{vec}(B_i A_i).$$

Define

$$\bar{\Sigma} := [\text{vec}(\Sigma_1), \ldots, \text{vec}(\Sigma_n)].$$

By our previous simplification,

$$\bar{\Sigma} = (V^\top \otimes U^\top)L.$$

Now

$$\sum_{i=1}^{n} \|\Sigma_i\|_{\text{Fro}}^2 = \|\bar{\Sigma}\|_{\text{Fro}}^2 = \text{tr}\left(((V \otimes U)(V \otimes U)^\top)(LL^\top)\right)$$

Since $U, V$ are orthogonal and size $d \times r$, the top $r^2$ eigenvalues of the symmetric matrix $(V \otimes U)(V \otimes U)^\top$ will be equal to 1, and the rest will equal 0. The eigenvalues of the symmetric matrix $LL^\top$ will be equal to the squared singular values of $L$. We can then apply the Von Neumann trace inequality to obtain the upper bound.

The last statement follows from the Pythagorean theorem and the fact that the $\Sigma_i$ is a projection of $B_i A_i$ to the $U, V$ subspace. $\square$

Note that we have only used the fact that the matrix $(V \otimes U)$ has singular values equal to 1; we have not used the fact that it has Kronecker product structure. On the other hand, each vector $\text{vec}(B_i A_i)$ is a sum of $r_i$ Kronecker products and cannot be expressed as a Kronecker product. As a result, while the upper bound in the Von Neumann trace inequality is achieved if the eigenvectors of the two matrices align, the Kronecker product structure is a severe constraint and the upper bound we have provided is generous.

## C. Training LoRAs

We trained LoRA adapters on 500 natural instruction tasks (Wang et al., 2022) using `Mistral-7B-Instruct-v0.2` (Jiang et al., 2023a) as the base model. All LoRA adapters were configured with a rank of 16, i.e., $\forall i, r_i = 16$. We selected 10 diverse tasks manually for consistent evaluation across experiments and randomly sampled an additional 490 tasks, resulting in a total of 500 tasks. These tasks were exclusively in English (both input and output), ensuring higher quality and thorough review (Wang et al., 2022). Each task dataset was divided into training, validation, and test sets (80-10-10). Hyperparameters, such as early stopping, were tuned using the validation sets; that is, we train for five epochs and take the best-performing epoch-checkpoint per validation loss. Evaluation on the test sets demonstrated that LoRA consistently outperformed the base model in terms of both Rouge scores and loss metrics (see Table 1).

In Table 1, we compare metrics between base model and LoRA finetuning.

Table 1: Comparison of metrics before and after LoRA training across 1000 tasks.

| Metric | Base Model | LoRA |
|---|---|---|
| Loss | $4.14 \pm 3.07$ | $0.56 \pm 0.58$ |
| Exact Match | $1.81 \pm 6.56$ | $51.38 \pm 40.90$ |
| Rouge-1 | $21.70 \pm 19.22$ | $68.88 \pm 29.73$ |
| Rouge-L | $20.62 \pm 18.21$ | $67.80 \pm 30.15$ |

In Table 3 we include all 1000 tasks that were used.

We use Huggingface (Wolf et al., 2020) in our implementation. For the base model, we use quantization with configuration:

Table 2: Main Evaluation Tasks

| Task Number | Name | Type | Domain |
|---|---|---|---|
| task280 | stereoset_classification_stereotype_type | classification | stereoset |
| task190 | snli_classification | snli | image captions |
| task391 | causal_relationship | commonsense | cause and effect |
| task290 | tellmewhy_question_answerability | answerability | story |
| task1391 | winogrande_easy_answer_generation | commonsense | social and physical |
| task1342 | amazon_us_reviews_title | title generation | amazon reviews |
| task442 | com_qa_paraphrase_question_generation | question generation | wikipedia |
| task620 | ohsumed_medical_subject_headings_answer_generation | keyword tagging | scientific |
| task1598 | nyc_long_text_generation | data to text | restaurants |
| task039 | qasc_find_overlapping_words | overlap extraction | natural science |

```
BitsAndBytesConfig(
    load_in_4bit=True,
    bnb_4bit_use_double_quant=True,
    bnb_4bit_quant_type="nf4",
    bnb_4bit_compute_dtype=torch.bfloat16,
)
```

and LoRA configuration:

```
LoraConfig(
    r=16,
    lora_alpha=32,
    target_modules=["q_proj", "k_proj", "v_proj"],
    lora_dropout=0.05,
    bias="none",
    task_type="CAUSAL_LM",
    init_lora_weights=init_lora_weights,
)
```

## D. Avoiding Batched Matrix Multiplication (BMM)

Fast LoRA (Wen & Chaudhuri, 2024) aims to alleviate the batched matrix multiplication (BMM) bottleneck when serving many LoRAs. They propose an adapter parameterization that replaces addition with elementwise multiplication, avoiding BMM and improving LoRA throughput at lower ranks. Our JD LoRA formulation also circumvents or heavily reduces the impact of BMM as discussed below, and both individual and joint compression methods can be applied to Fast LoRAs.

In the envisioned deployment scenario, a service provider hosts a large collection of LoRAs. Upon receiving a request, each user specifies both the input data and the desired LoRA identifier. The provider then processes the base model augmented with the specified LoRA for each user's data. As a provider is batching a collection of requests for GPU parallelization, they can expect to frequently have more than one unique LoRA identifier per batch.

Traditionally, a specific LoRA is integrated into the base model by transforming $W_0 \rightarrow W_0 + B_i A_i$. Serving multiple LoRAs conventionally would necessitate maintaining and executing a separate copy of the base model for each LoRA, bringing substantial computational overhead. Alternatively, the computation for $W_0 x$ and $B_i A_i x$ can be performed independently and subsequently merged. This strategy necessitates only a single instance of $W_0 x$ computation and storage of LoRA-specific parameters rather than the entire base model.

Consider the batch processing of $\mathbf{B}\mathbf{A}\mathbf{x}$, where boldface indicates that $B_i, A_i$ are stacked into tensors of dimensions ($b \times$

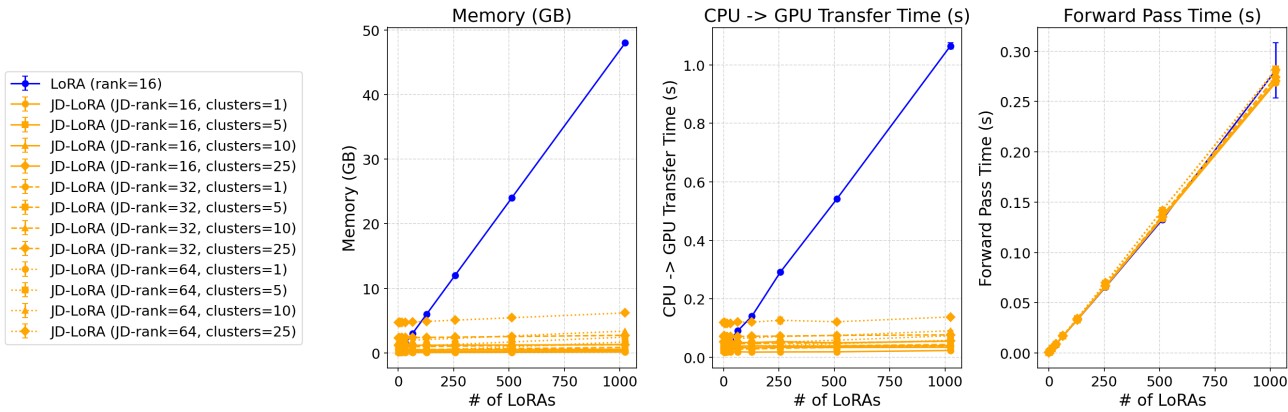

Figure 5: Memory load, transfer time, and forward-pass performance of LoRA and JD-LoRA.

$m \times r$) and $(b \times r \times n)$ respectively, with batched data $\mathbf{x}$ shaped $(b \times l \times n)$:

$$\mathbf{A}\mathbf{x} \leftrightarrow (b \times r \times n) \times (b \times l \times n) \to (b \times l \times r) \text{ bmm}$$
$$\mathbf{B}(\mathbf{A}\mathbf{x}) \leftrightarrow (b \times m \times r) \times (b \times l \times r) \to (b \times l \times m) \text{ bmm}.$$

Here, "bmm" denotes batched matrix multiplication, a known bottleneck in both throughput and latency. Consider the corresponding operations for our joint compression scheme, $U\mathbf{\Sigma}V^\top x$:

$$V^\top \mathbf{x} \leftrightarrow (\tilde{r} \times n) \times (b \times l \times n) \to (b \times l \times \tilde{r}) \text{ broadcasted}$$
$$\mathbf{\Sigma}(V^\top \mathbf{x}) \leftrightarrow (b \times \tilde{r}) \times (b \times l \times \tilde{r}) \to (b \times l \times \tilde{r}) \text{ broadcasted}$$
$$U(\mathbf{\Sigma}V^\top \mathbf{x}) \leftrightarrow (m \times \tilde{r}) \times (b \times l \times \tilde{r}) \to (b \times l \times m) \text{ broadcasted}$$

In our optimized setup, batched matrix multiplications can be completely circumvented if the $\Sigma_i$ matrices are diagonal. If not, given that $\tilde{r} \ll m, n$, any required batched matrix multiplication remains computationally inexpensive.

## E. Simple Timing Experiments

In Figure 5, we present a set of simple experiments comparing the memory load, transfer time, and forward-pass performance of LoRA and JD-LoRA. These experiments were conducted across multiple clusters and various rank configurations. For memory usage, we measured all 96 LoRA modules in the Mistral model; however, for transfer time and forward-pass performance, we tested only a single LoRA module. We also implemented F-LoRA (Wen & Chaudhuri, 2024), but contrary to their reported results, we were unable to achieve faster forward-pass performance than standard LoRA.

## F. GPU Memory Usage Computation for JD Compression.

The GPU memory consumption is primarily influenced by the number of parameters that need to be stored and processed during inference. In this section, we introduce the detail of how we compute the GPU consumption of our method, and how we find the number of vLLM multi-LoRA that share the same GPU utilization.

- $D$: Hidden dimension size (e.g., $D = 4098$).
- $r$: Rank of the shared basis matrices for compression (e.g., $r = 16, 32, 64$).
- $N$: Maximum number of LoRA modules being served simultaneously (max_lora_num).
- $c$: Number of clusters in our clustering method (e.g., $c = 7, 10, 25$).

In Figure 1, we use different JD-compression settings for serving different number of unique LoRAs. Specifically:

- **Serving 4 unique LoRAs:**
  Ours: rank 16 JD-Full.
  vLLM multiLoRA baseline: max-gpu-lora = 2.

- **Serving 8 unique LoRAs:**
  Ours: rank 16 JD-Full.
  vLLM multiLoRA baseline: max-gpu-lora = 2.
- **Serving 16 unique LoRAs:**
  Ours: rank 32 JD-Full.
  vLLM multiLoRA baseline: max-gpu-lora = 3.
- **Serving 32 unique LoRAs:**
  Ours: rank 64 JD-Full.
  vLLM multiLoRA baseline: max-gpu-lora = 5.
- **Serving 64 unique LoRAs:**
  Ours: rank 64 JD-Full.
  vLLM multiLoRA baseline: max-gpu-lora = 6.
- **Serving 128 unique LoRAs:**
  Ours: 7 clusters, rank 16 JD-Full.
  vLLM multiLoRA baseline: max-gpu-lora = 8.
- **Serving 256 unique LoRAs:**
  Ours: 10 clusters, rank 16 JD-Full.
  vLLM multiLoRA baseline: max-gpu-lora = 10.
- **Serving 512 unique LoRAs:**
  Ours: 25 clusters, rank 16 JD-Full.
  vLLM multiLoRA baseline: max-gpu-lora = 26.
- **Serving 1024 unique LoRAs:**
  Ours: 25 clusters, rank 16 JD-Full.
  vLLM multiLoRA baseline: max-gpu-lora = 28

### F.1. Baseline GPU Memory Usage

The baseline for our comparison is the standard LoRA method with a rank of 16. The total parameter count for the baseline is given by:

$$\text{Params}_{\text{baseline}} = D \times 2 \times 16.$$

This accounts for the parameters in the LoRA-adapted layers, where the factor of 2 represents the weights and biases.

### F.2. GPU Memory Usage for JD Full Method

For the Joint Decomposition (JD) Full method without clustering, the total parameter count is:

$$\text{Params}_{\text{JD\_Full}} = D \times 2 \times r + N \times r^2.$$

- $D \times 2 \times r$: Parameters for the base model adapted with rank-$r$ LoRA.
- $N \times r^2$: Additional parameters introduced by each of the $N$ LoRA modules, each of size $r \times r$.

The GPU memory usage ratio relative to the baseline is:

$$\text{GPU Usage Ratio}_{\text{JD\_Full}} = \frac{\text{Params}_{\text{JD\_Full}}}{\text{Params}_{\text{baseline}}} = \frac{D \times 2 \times r + N \times r^2}{D \times 2 \times 16}.$$

### F.3. GPU Memory Usage for Clustering Method

When employing clustering, the parameter count changes due to the addition of cluster-specific parameters:

$$\text{Params}_{\text{Clustering}} = D \times 2 \times r \times c + N \times (r^2 + 1).$$

- $D \times 2 \times r \times c$: Parameters for the base model adapted with rank-$r$ LoRA across $c$ clusters.

- $N \times (r^2 + 1)$: Additional parameters for each LoRA module and cluster assignments.

The GPU memory usage ratio is:

$$\text{GPU Usage Ratio}_{\text{Clustering}} = \frac{\text{Params}_{\text{Clustering}}}{\text{Params}_{\text{baseline}}} = \frac{D \times 2 \times r \times c + N \times (r^2 + 1)}{D \times 2 \times 16}.$$

### F.4. Punica

In our `vLLM` experiments, we specifically used the Punica kernel for implementing multi-LoRA, applying our approach in conjunction with Punica's capabilities. Our custom function, `add_lora_slice_with_sigma`, implements the following key steps:

1. **Initialize Buffers**: Creates temporary storage for intermediate calculations if not already provided.
2. **Apply Matrix A**: Transforms `x` using matrix `A`, storing the result in `buffer`.
3. **Apply Matrix Sigma**: Further transforms `buffer` using `Sigma`, storing the result in `buffer_sigma`.
4. **Apply Matrix B and Update y**: Finally, transforms `buffer_sigma` using `B`, applies scaling, and updates a slice of `y` in place.

Below is the pseudocode for `add_lora_slice_with_sigma`, illustrating the integration:

Listing 1: Pseudocode for 'add_lora_slice_with_sigma'

```
Function add_lora_slice_with_sigma(y, x, wa_t_all, wb_t_all, wsigma_t_all, indices,
    layer_idx, scale, y_offset, y_slice_size, buffer=None):
    # Initialize buffers if not provided
    if buffer is None:
        buffer = create_tensor(shape=(x.size(0), R), dtype=float32)
        buffer_sigma = create_tensor(shape=(buffer.size(0), R), dtype=float32)
    # Step 1: Apply matrix A
    dispatch_bgmv_low_level(buffer, x, wa_t_all, indices, layer_idx, scale=1.0)
    # Step 2: Apply matrix Sigma
    dispatch_bgmv_low_level(buffer_sigma, buffer, wsigma_t_all, indices, layer_idx, scale
        =1.0)
    # Step 3: Apply matrix B and update y slice
    dispatch_bgmv_low_level(y, buffer_sigma, wb_t_all, indices, layer_idx, scale, y_offset
        , y_slice_size)
End Function
```

## G. Selecting Number of Clusters

To identify optimal hyperparameters for the clusters compression method, we analyzed the relationship between reconstruction error and the parameter saved ratio for a single LoRA module, as shown in Figure 6. By comparing the results across different numbers of Low-Rank Adaptation (LoRA) configurations (100 and 500, depicted in subfigures 6a and 6b), we were able to observe the trade-off between model size reduction and reconstruction accuracy. Based on these findings, we selected the rank and number-of-clusters hyperparameters that effectively balance these two objectives. The chosen settings were then used to conduct full-scale experiments.

## H. Additional Results

This section elaborates on the results that underpin the figures presented in the main text and showcases a consistent correlation across various evaluation metrics. Additionally, we assess the significance of achieving convergence and the performance of compression on new unseen LoRA models.

### H.1. LoRAs of different ranks

In Table 4, we report the performance of our compression method on LoRAs with ranks uniformly sampled between 16 and 64 (mean rank of 43). In Table 5, we present compression results for LoRAs of rank 43, matching the average rank from Table 4. Because these same-rank LoRAs have an identical parameter count, they also exhibit identical parameter-saving

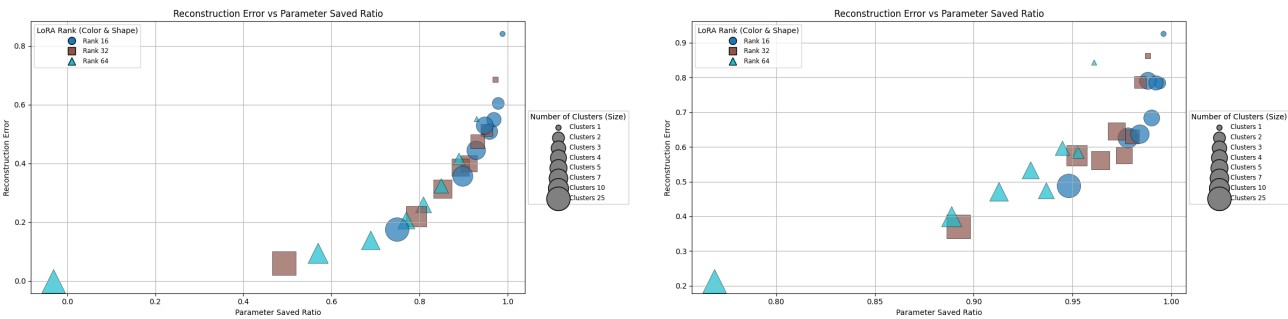

(a) Recon. Error vs Parameter Saved Ratio for 100 LoRAs

(b) Recon. Error vs Parameter Saved Ratio for 500 LoRAs

Figure 6: Comparison of reconstruction error against the parameter saved ratio for different numbers of LoRA configurations for a single LoRA module. The left subplot shows results for 100 LoRAs, while the right subplot displays results for 500 LoRAs. These plots illustrate the trade-off between reconstruction accuracy and compression efficiency, providing insights into optimal parameter settings for compression.

ratios. While performance declines slightly for LoRAs with varying ranks, our compression method still preserves over 99% of the original performance.

## H.2. Privacy Ablation

We investigate whether jointly compressing certain tasks results in improved performance on tasks within that same compressed group, compared to tasks outside of the group. In other words, we examine whether information about which tasks were compressed together could be inferred from subsequent performance differences. Table 6 presents the results.

In the Compressed Together setting, tasks (task1391, task190, task280, task290, and task391) are compressed jointly with five other tasks. We then evaluate their cross-task performance within this group. In the Compressed Separately setting, the same set of tasks is each compressed alongside nine other tasks that are not part of the original group. We again evaluate the cross-task performance using the same sets of tasks, allowing us to compare and assess any differences in performance attributable to joint versus separate compression.

Table 6: Privacy Ablation: Performance on tasks compressed together vs. performance on tasks compressed separately

| Source Task | Task Combination | Test Loss | Exact Match | Rouge1 | RougeL |
|---|---|---|---|---|---|
| **Compressed Together** | | | | | |
| task1391 | task1391 on task190 | 2.320 | 29 | 29.00 | 29.00 |
| | task1391 on task280 | 1.445 | 10 | 12.00 | 12.00 |
| | task1391 on task290 | 2.662 | 0 | 0.00 | 0.00 |
| | task1391 on task391 | 1.560 | 4 | 7.00 | 7.00 |
| | **Average** | 1.997 | 10.8 | 12.00 | 12.00 |
| task190 | task190 on task1391 | 1.392 | 0 | 0.00 | 0.00 |
| | task190 on task280 | 1.647 | 2 | 2.00 | 2.00 |
| | task190 on task290 | 1.838 | 0 | 0.00 | 0.00 |
| | task190 on task391 | 2.622 | 2 | 2.00 | 2.00 |
| | **Average** | 1.875 | 1.0 | 1.00 | 1.00 |
| task280 | task280 on task1391 | 2.259 | 14 | 16.47 | 16.47 |
| | task280 on task190 | 2.922 | 20 | 20.50 | 20.50 |
| | task280 on task290 | 0.729 | 33 | 43.07 | 43.07 |
| | task280 on task391 | 2.318 | 44 | 62.52 | 62.52 |
| | **Average** | 2.057 | 27.8 | 35.64 | 35.64 |
| task290 | task290 on task1391 | 0.867 | 36 | 46.58 | 46.58 |
| | task290 on task190 | 1.553 | 36 | 36.00 | 36.00 |
| | task290 on task280 | 1.036 | 43 | 43.40 | 43.40 |
| | task290 on task391 | 0.461 | 59 | 86.33 | 86.33 |
| | **Average** | 0.979 | 43.5 | 53.08 | 53.08 |
| task391 | task391 on task1391 | 0.502 | 65 | 65.75 | 65.75 |

Table 6: Privacy Ablation: Performance on tasks compressed together vs. performance on tasks compressed separately (continued)

| Source Task | Task Combination | Test Loss | Exact Match | Rouge1 | RougeL |
|---|---|---|---|---|---|
| | task391 on task190 | 1.421 | 31 | 31.00 | 31.00 |
| | task391 on task280 | 0.417 | 69 | 69.00 | 69.00 |
| | task391 on task290 | 0.265 | 71 | 90.33 | 90.33 |
| | **Average** | 0.651 | 59.0 | 64.02 | 64.02 |
| **Compressed Separately** | | | | | |
| | task1391 on task190 | 2.347 | 30 | 30.00 | 30.00 |
| | task1391 on task280 | 1.543 | 11 | 13.58 | 13.58 |
| task1391 | task1391 on task290 | 2.477 | 0 | 0.00 | 0.00 |
| | task1391 on task391 | 1.253 | 10 | 18.00 | 18.00 |
| | **Average** | 1.905 | 12.8 | 15.40 | 15.40 |
| | task190 on task1391 | 1.388 | 0 | 0.00 | 0.00 |
| | task190 on task280 | 1.603 | 2 | 2.00 | 2.00 |
| task190 | task190 on task290 | 1.771 | 0 | 0.00 | 0.00 |
| | task190 on task391 | 2.326 | 4 | 4.00 | 4.00 |
| | **Average** | 1.772 | 1.5 | 1.50 | 1.50 |
| | task280 on task1391 | 3.111 | 2 | 5.13 | 5.13 |
| | task280 on task190 | 2.711 | 19 | 19.00 | 19.00 |
| task280 | task280 on task290 | 0.948 | 16 | 22.09 | 22.09 |
| | task280 on task391 | 2.449 | 39 | 53.11 | 53.11 |
| | **Average** | 2.305 | 19.0 | 24.83 | 24.83 |
| | task290 on task1391 | 0.848 | 41 | 47.58 | 47.58 |
| | task290 on task190 | 1.355 | 38 | 38.00 | 38.00 |
| task290 | task290 on task280 | 1.050 | 41 | 41.00 | 41.00 |
| | task290 on task391 | 0.463 | 59 | 86.33 | 86.33 |
| | **Average** | 0.929 | 44.8 | 53.23 | 53.23 |
| | task391 on task1391 | 0.428 | 68 | 68.74 | 68.74 |
| | task391 on task190 | 1.507 | 31 | 31.00 | 31.00 |
| task391 | task391 on task280 | 0.368 | 70 | 70.00 | 70.00 |
| | task391 on task290 | 0.269 | 73 | 91.00 | 91.00 |
| | **Average** | 0.643 | 60.5 | 65.18 | 65.18 |

## H.3. Relative Rouge-L Performance and Compression Rate

Table 7 presents comprehensive results from the experiments underlying Figure 2 for each evaluation task. Additionally, we incorporate results using the Ties-merging benchmark (Yadav et al., 2023b), which consolidates all LoRA-adapters into a single adapter of identical configuration and parameter count; this integration significantly compromises performance.

## H.4. Absolute Rouge-L Performance and Compression Rate

Table 8 provides the full results behind Table 7, but with Rouge-L scores instead of relative performance compared to LoRA.

## H.5. Relative Rouge-1 Performance and Compression Rate

Table 9 provides full results for relative performance of Rouge-1, which shows the same trends as the results for relative performance of Rouge-L (Table 7).

## H.6. Absolute Rouge-1 Performance and Compression Rate

Table 10 provides full results for absolute performance of Rouge-1, which shows the same trends as the results for absolute performance of Rouge-L (Table 8).

## H.7. Relative Exact-Match Performance and Compression Rate

Table 11 provides full results for relative performance of exact-match, which shows the same trends as the results for relative performance of Rouge-L (Table 7).

## H.8. Loss and Compression Rate

Table 12 provides full results for test loss (cross-entropy), which shows the same trends as the results for relative performance of Rouge-L (Table 7).

## H.9. Agreement and Compression Rate

Table 13 provides full results for *agreement*, which shows the same trends as the results for relative performance of Rouge-L (Table 7). Note that *agreement* measures the exact match in task generations between the uncompressed LoRA model and the compressed LoRA model, rather than comparing to the task's ground truth data. The comparison is very strict and requires an exact match between the generations of the two models (LoRA and the compressed LoRA), comparing each sample one at a time.

## H.10. Reconstruction Error and Compression Rate

Table 14 provides the full results of the experiments behind Figure 3 for every evaluation task.

## H.11. Reconstruction Error: Trained vs. Random

Table 15 provides the reconstruction error on random (untrained) LoRA matrices. Comparing with Table 14, we find that reconstruction error is consistently higher on random (untrained LoRA) matrices than on trained LoRA matrices. This demonstrates that after training, LoRAs have a shared structure that JD exploits.

## H.12. Convergence

Table 16 presents outcomes where the JD-Full algorithm is executed until convergence. Our convergence criterion is defined as follows:

$$\max \left( \|U_{t+1} - U_t U_t^\top U_{t+1}\|_{\mathrm{Fro}} / \|U_{t+1}\|_{\mathrm{Fro}}, \|V_{t+1} - V_t V_t^\top V_{t+1}\|_{\mathrm{Fro}} / \|V_{t+1}\|_{\mathrm{Fro}} \right) < \tau \tag{19}$$

where the tolerance threshold $\tau$ is set to 0.001. Due to the slow per-iteration computation times of the primary JD-Full algorithm, which quickly reaches an approximate optimum but then has a long tail of convergence for final digits of precision, we devised an alternative eigenvalue iteration algorithm (Appendix A.2) optimized for GPU acceleration. Our analysis indicates that adherence to this convergence criterion does not significantly alter the results.

## H.13. Out-of-distribution Performance (LoRA-hub)

For completeness, we incorporate results using the protocol of LoRA-hub (Huang et al., 2024). That is, 100 LoRA-adapters are sampled, independent of the evaluation task, representing a measure of out-of-distribution performance. This also means that each result on a task is averaged across all 100 LoRA-adapters (as there is no *a priori* LoRA-to-task mapping). These results were obtained without normalizing the LoRA-adapters before applying the JD algorithms, a step we later identified as beneficial. We present performance comparison in Table 18. Table 17 presents the average agreement between uncompressed and compressed LoRA across 10 evaluation tasks. Results per task for JD-diagonal and JD-full are shown in Table 19 and Table 20, respectively.

From these tables, we find that the JD algorithms successfully maintain performance in this out-of-distribution context.

Table 3: List of all 1000 Tasks

| Task ID | Description | Task ID | Description | Task ID | Description |
|---|---|---|---|---|---|

Table 4: Results for Different Numbers of Clusters (different LoRA-ranks)

| Metric | Uncompressed | 1 Cluster | 2 Clusters | 4 Clusters | 8 Clusters |
|---|---|---|---|---|---|
| Loss | 0.5009 | 1.6023 | 0.9986 | 0.5603 | 0.5000 |
| Exact match | 69.6103 | 37.3392 | 60.1071 | 66.6474 | 69.6289 |
| ROUGE-1 | 79.5755 | 50.3598 | 71.5644 | 76.2930 | 79.0576 |
| ROUGE-L | 78.9355 | 49.7491 | 71.0193 | 75.6152 | 78.3958 |
| Recon. error | 0 | 0.8311 | 0.6990 | 0.4846 | 0.3246 |
| Agreement | 1.0 | 36.6706 | 62.6699 | 72.2231 | 80.5545 |
| Exact match ratio | 1.0 | 0.5724 | 0.8079 | 0.9219 | 0.9541 |
| Relative ROUGE-1 | 1.0 | 0.6220 | 0.8533 | 0.9406 | 0.9917 |
| Relative ROUGE-L | 1.0 | 0.6221 | 0.8550 | 0.9404 | 0.9915 |
| Param. saved ratio | 1.0 | 0.99 | 0.98 | 0.97 | 0.94 |

Table 5: Results for Different Numbers of Clusters (same LoRA-ranks of 43)

| Metric | Uncompressed | 1 Cluster | 2 Clusters | 4 Clusters | 8 Clusters |
|---|---|---|---|---|---|
| Loss | 0.5764 | 0.7767 | 0.6277 | 0.5841 | 0.5659 |
| Exact match | 61.7746 | 53.2000 | 60.2000 | 61.5000 | 61.3000 |
| ROUGE-1 | 79.9322 | 74.7695 | 79.4621 | 80.6950 | 80.4298 |
| ROUGE-L | 77.7961 | 72.4257 | 77.2141 | 78.3369 | 78.1883 |
| Recon. error | 0 | 0.7333 | 0.5594 | 0.3999 | 0.2502 |
| Agreement | 1.0 | 0.0000 | 6.6667 | 6.6667 | 6.6667 |
| Exact match ratio | 1.0 | 0.8175 | 0.9261 | 0.9618 | 1.0206 |
| Relative ROUGE-1 | 1.0 | 0.9560 | 1.0153 | 1.0310 | 1.0241 |
| Relative ROUGE-L | 1.0 | 0.9501 | 1.0118 | 1.0268 | 1.0231 |
| Param. saved ratio | 1.0 | 0.99 | 0.98 | 0.97 | 0.94 |

| Model Type | Method Type | Tasks | | | | | | | | | | Average | Para. Saved |
|---|---|---|---|---|---|---|---|---|---|---|---|---|---|
| | | task039 | task190 | task280 | task290 | task391 | task442 | task620 | task1342 | task1391 | task1598 | | |
| | base | 0.26 ± 0.00 | 0.02 ± 0.00 | 0.19 ± 0.00 | 0.42 ± 0.00 | 0.11 ± 0.00 | 0.47 ± 0.00 | 0.11 ± 0.00 | 0.23 ± 0.00 | 0.19 ± 0.00 | 0.77 ± 0.00 | 0.28 ± 0.21 | 1.00/1.00 |
| | lora | 1.00 ± 0.00 | 1.00 ± 0.00 | 1.00 ± 0.00 | 1.00 ± 0.00 | 1.00 ± 0.00 | 1.00 ± 0.00 | 1.00 ± 0.00 | 1.00 ± 0.00 | 1.00 ± 0.00 | 1.00 ± 0.00 | 1.00 ± 0.00 | 0.00/0.00 |
| TIES | 10 | 0.81 ± 0.00 | 0.57 ± 0.02 | 0.45 ± 0.04 | 0.10 ± 0.01 | 0.83 ± 0.01 | 0.47 ± 0.00 | 0.69 ± 0.01 | 0.57 ± 0.00 | 0.82 ± 0.01 | 0.85 ± 0.00 | 0.62 ± 0.23 | 1.00 / 1.00 |
| | 50 | 0.59 ± 0.00 | 0.41 ± 0.00 | 0.18 ± 0.05 | 0.03 ± 0.01 | 0.91 ± 0.01 | 0.31 ± 0.00 | 0.65 ± 0.00 | 0.62 ± 0.00 | 0.32 ± 0.04 | 0.84 ± 0.00 | 0.48 ± 0.28 | 1.00 / 1.00 |
| | 100 | 0.55 ± 0.00 | 0.40 ± 0.01 | 0.20 ± 0.05 | 0.01 ± 0.02 | 0.88 ± 0.00 | 0.33 ± 0.00 | 0.64 ± 0.00 | 0.57 ± 0.02 | 0.01 ± 0.00 | 0.82 ± 0.00 | 0.44 ± 0.30 | 1.00 / 1.00 |
| | 500 | 0.37 ± 0.00 | 0.26 ± 0.00 | 0.01 ± 0.00 | 0.00 ± 0.00 | 0.83 ± 0.00 | 0.29 ± 0.00 | 0.57 ± 0.00 | 0.37 ± 0.00 | 0.01 ± 0.00 | 0.43 ± 0.00 | 0.31 ± 0.26 | 1.00 / 1.00 |
| SVD | SVD 2 | 0.98 ± 0.03 | 1.07 ± 0.02 | 1.00 ± 0.00 | 1.00 ± 0.00 | 1.00 ± 0.00 | 0.98 ± 0.01 | 1.00 ± 0.01 | 1.00 ± 0.10 | 1.00 ± 0.01 | 1.00 ± 0.01 | 1.00 ± 0.04 | 0.88 / 0.88 |
| | SVD 4 | 0.99 ± 0.04 | 1.04 ± 0.01 | 1.00 ± 0.00 | 1.00 ± 0.00 | 1.00 ± 0.01 | 1.00 ± 0.00 | 0.99 ± 0.02 | 0.99 ± 0.08 | 0.99 ± 0.01 | 1.00 ± 0.01 | 1.00 ± 0.03 | 0.75 / 0.75 |
| | SVD 8 | 1.00 ± 0.00 | 1.02 ± 0.01 | 1.00 ± 0.00 | 1.00 ± 0.00 | 1.00 ± 0.00 | 1.00 ± 0.00 | 1.01 ± 0.00 | 1.00 ± 0.01 | 1.01 ± 0.01 | 1.01 ± 0.00 | 1.00 ± 0.01 | 0.50 / 0.50 |
| | SVD 16 | 1.00 ± 0.00 | 1.00 ± 0.00 | 1.00 ± 0.00 | 1.00 ± 0.00 | 1.00 ± 0.00 | 1.00 ± 0.00 | 1.00 ± 0.00 | 1.00 ± 0.00 | 1.00 ± 0.00 | 1.00 ± 0.00 | 1.00 ± 0.00 | 0.00 / 0.00 |
| 10 diagonal (D) | 16 D | 1.02 ± 0.01 | 1.01 ± 0.01 | 1.00 ± 0.00 | 1.00 ± 0.01 | 0.99 ± 0.00 | 0.96 ± 0.00 | 1.02 ± 0.02 | 1.13 ± 0.03 | 0.99 ± 0.02 | 0.98 ± 0.01 | 1.01 ± 0.05 | 1.00 / 0.90 |
| | 32 D | 1.01 ± 0.01 | 1.05 ± 0.01 | 1.00 ± 0.00 | 0.99 ± 0.00 | 1.01 ± 0.01 | 0.99 ± 0.00 | 0.97 ± 0.01 | 1.05 ± 0.03 | 1.00 ± 0.01 | 1.00 ± 0.01 | 1.00 ± 0.03 | 1.00 / 0.80 |
| | 64 D | 1.00 ± 0.00 | 1.03 ± 0.01 | 1.00 ± 0.00 | 1.00 ± 0.00 | 1.00 ± 0.00 | 1.00 ± 0.00 | 1.01 ± 0.00 | 0.99 ± 0.01 | 1.01 ± 0.00 | 1.01 ± 0.00 | 1.00 ± 0.01 | 1.00 / 0.60 |
| | 128 D | 1.00 ± 0.00 | 1.01 ± 0.01 | 1.00 ± 0.00 | 1.00 ± 0.00 | 1.00 ± 0.00 | 1.00 ± 0.00 | 1.01 ± 0.01 | 0.99 ± 0.01 | 1.00 ± 0.00 | 1.00 ± 0.00 | 1.00 ± 0.01 | 1.00 / 0.20 |
| | 256 D | 1.00 ± 0.00 | 1.00 ± 0.00 | 1.00 ± 0.00 | 1.00 ± 0.00 | 1.00 ± 0.00 | 1.00 ± 0.00 | 1.00 ± 0.00 | 1.00 ± 0.00 | 1.00 ± 0.00 | 1.00 ± 0.00 | 1.00 ± 0.00 | 1.00 / -0.60 |
| 10 full (F) | 16 F | 1.02 ± 0.00 | 1.06 ± 0.01 | 1.00 ± 0.00 | 1.00 ± 0.00 | 0.99 ± 0.01 | 0.98 ± 0.00 | 1.01 ± 0.02 | 1.07 ± 0.00 | 1.01 ± 0.01 | 1.00 ± 0.01 | 1.01 ± 0.03 | 1.00/0.90 |
| | 32 F | 1.02 ± 0.01 | 1.04 ± 0.01 | 1.00 ± 0.00 | 1.00 ± 0.00 | 1.00 ± 0.00 | 0.99 ± 0.00 | 0.96 ± 0.01 | 1.00 ± 0.02 | 1.00 ± 0.01 | 1.01 ± 0.00 | 1.00 ± 0.02 | 0.99/0.79 |
| | 64 F | 1.00 ± 0.00 | 1.03 ± 0.01 | 1.00 ± 0.00 | 1.00 ± 0.00 | 1.00 ± 0.00 | 1.00 ± 0.00 | 1.01 ± 0.01 | 0.98 ± 0.01 | 1.01 ± 0.00 | 1.01 ± 0.00 | 1.00 ± 0.01 | 0.97/0.57 |
| | 128 F | 1.00 ± 0.00 | 1.01 ± 0.01 | 1.00 ± 0.00 | 1.00 ± 0.00 | 1.00 ± 0.00 | 1.00 ± 0.00 | 1.00 ± 0.00 | 0.99 ± 0.00 | 1.00 ± 0.00 | 1.00 ± 0.00 | 1.00 ± 0.00 | 0.88/0.07 |
| | 256 F | 1.00 ± 0.00 | 1.00 ± 0.00 | 1.00 ± 0.00 | 1.00 ± 0.00 | 1.00 ± 0.00 | 1.00 ± 0.00 | 1.00 ± 0.00 | 1.00 ± 0.00 | 1.00 ± 0.00 | 1.00 ± 0.00 | 1.00 ± 0.00 | 0.50/-1.10 |
| 50 diagonal (D) | 16 D | 0.98 ± 0.04 | 0.98 ± 0.01 | 1.00 ± 0.00 | 0.92 ± 0.06 | 0.84 ± 0.07 | 0.92 ± 0.02 | 0.68 ± 0.05 | 0.87 ± 0.10 | 0.88 ± 0.07 | 0.83 ± 0.02 | 0.89 ± 0.10 | 1.00 / 0.98 |
| | 32 D | 1.00 ± 0.02 | 1.02 ± 0.02 | 1.00 ± 0.00 | 0.99 ± 0.00 | 0.96 ± 0.01 | 0.95 ± 0.02 | 0.84 ± 0.02 | 1.00 ± 0.13 | 0.98 ± 0.01 | 0.88 ± 0.01 | 0.96 ± 0.07 | 1.00 / 0.96 |
| | 64 D | 1.02 ± 0.00 | 1.05 ± 0.02 | 1.00 ± 0.00 | 1.00 ± 0.00 | 0.99 ± 0.01 | 0.97 ± 0.00 | 0.99 ± 0.01 | 1.09 ± 0.03 | 1.01 ± 0.01 | 0.90 ± 0.01 | 1.00 ± 0.05 | 1.00 / 0.92 |
| | 128 D | 1.01 ± 0.01 | 1.08 ± 0.01 | 1.00 ± 0.00 | 1.00 ± 0.00 | 0.99 ± 0.00 | 0.98 ± 0.00 | 0.98 ± 0.01 | 1.11 ± 0.03 | 1.00 ± 0.01 | 1.00 ± 0.01 | 1.01 ± 0.04 | 1.00 / 0.84 |
| | 256 D | 1.01 ± 0.01 | 1.03 ± 0.01 | 1.00 ± 0.00 | 1.00 ± 0.00 | 1.00 ± 0.01 | 1.00 ± 0.00 | 0.97 ± 0.03 | 1.01 ± 0.03 | 1.00 ± 0.01 | 1.01 ± 0.01 | 1.00 ± 0.02 | 1.00 / 0.68 |
| 50 full (F) | 16 F | 0.99 ± 0.04 | 1.00 ± 0.01 | 1.00 ± 0.01 | 0.96 ± 0.01 | 0.95 ± 0.02 | 0.94 ± 0.01 | 0.64 ± 0.10 | 1.01 ± 0.15 | 0.97 ± 0.02 | 0.87 ± 0.00 | 0.93 ± 0.12 | 1.00/0.98 |
| | 32 F | 1.02 ± 0.00 | 1.00 ± 0.02 | 1.00 ± 0.00 | 1.00 ± 0.00 | 0.98 ± 0.01 | 0.96 ± 0.00 | 0.95 ± 0.01 | 1.09 ± 0.02 | 1.01 ± 0.02 | 0.89 ± 0.01 | 0.99 ± 0.05 | 0.99/0.95 |
| | 64 F | 1.02 ± 0.00 | 1.06 ± 0.01 | 1.00 ± 0.00 | 1.00 ± 0.00 | 0.99 ± 0.01 | 0.98 ± 0.01 | 1.03 ± 0.00 | 1.11 ± 0.00 | 1.00 ± 0.01 | 0.98 ± 0.02 | 1.02 ± 0.04 | 0.97/0.89 |
| | 128 F | 1.02 ± 0.00 | 1.06 ± 0.01 | 1.00 ± 0.00 | 1.00 ± 0.00 | 1.00 ± 0.01 | 0.98 ± 0.00 | 0.98 ± 0.01 | 1.03 ± 0.04 | 1.00 ± 0.01 | 1.00 ± 0.00 | 1.01 ± 0.03 | 0.88/0.72 |
| | 256 F | 1.00 ± 0.00 | 1.02 ± 0.00 | 1.00 ± 0.00 | 1.00 ± 0.00 | 1.00 ± 0.00 | 0.99 ± 0.00 | 1.01 ± 0.01 | 1.00 ± 0.00 | 1.01 ± 0.00 | 1.01 ± 0.00 | 1.00 ± 0.01 | 0.50/0.18 |
| 100 diagonal (D) | 16 D | 0.80 ± 0.07 | 0.89 ± 0.06 | 0.93 ± 0.03 | 0.96 ± 0.01 | 0.50 ± 0.09 | 0.78 ± 0.05 | 0.28 ± 0.07 | 0.52 ± 0.10 | 0.78 ± 0.05 | 0.81 ± 0.02 | 0.72 ± 0.22 | 1.00 / 0.99 |
| | 32 D | 0.95 ± 0.06 | 0.98 ± 0.05 | 1.00 ± 0.00 | 0.91 ± 0.06 | 0.80 ± 0.14 | 0.89 ± 0.06 | 0.60 ± 0.10 | 0.77 ± 0.26 | 0.91 ± 0.02 | 0.83 ± 0.02 | 0.86 ± 0.14 | 1.00 / 0.98 |
| | 64 D | 1.01 ± 0.03 | 1.01 ± 0.01 | 1.00 ± 0.00 | 0.98 ± 0.02 | 0.96 ± 0.01 | 0.94 ± 0.01 | 0.88 ± 0.05 | 1.11 ± 0.08 | 0.96 ± 0.02 | 0.87 ± 0.03 | 0.97 ± 0.07 | 1.00 / 0.96 |
| | 128 D | 1.01 ± 0.00 | 1.02 ± 0.01 | 1.00 ± 0.00 | 1.00 ± 0.00 | 0.99 ± 0.01 | 0.97 ± 0.00 | 1.00 ± 0.03 | 1.11 ± 0.02 | 0.99 ± 0.01 | 0.89 ± 0.02 | 1.00 ± 0.05 | 1.00 / 0.92 |
| | 256 D | 1.00 ± 0.00 | 1.06 ± 0.01 | 1.00 ± 0.00 | 1.00 ± 0.00 | 0.99 ± 0.00 | 0.98 ± 0.00 | 1.00 ± 0.01 | 1.11 ± 0.03 | 1.00 ± 0.01 | 0.98 ± 0.01 | 1.01 ± 0.04 | 1.00 / 0.84 |
| 100 full (F) | 16 F | 0.95 ± 0.01 | 0.97 ± 0.03 | 0.97 ± 0.03 | 0.97 ± 0.03 | 0.93 ± 0.01 | 0.92 ± 0.01 | 0.64 ± 0.03 | 0.89 ± 0.16 | 0.87 ± 0.02 | 0.83 ± 0.01 | 0.89 ± 0.11 | 1.00/0.99 |
| | 32 F | 1.00 ± 0.02 | 0.99 ± 0.01 | 1.00 ± 0.00 | 1.00 ± 0.00 | 0.97 ± 0.01 | 0.95 ± 0.00 | 0.86 ± 0.03 | 1.12 ± 0.03 | 0.96 ± 0.01 | 0.87 ± 0.00 | 0.97 ± 0.07 | 0.99/0.97 |
| | 64 F | 1.02 ± 0.00 | 1.00 ± 0.02 | 1.00 ± 0.00 | 1.00 ± 0.00 | 0.98 ± 0.00 | 0.96 ± 0.00 | 0.99 ± 0.01 | 1.09 ± 0.01 | 0.99 ± 0.02 | 0.89 ± 0.00 | 0.99 ± 0.05 | 0.97/0.93 |
| | 128 F | 1.01 ± 0.01 | 1.05 ± 0.01 | 1.00 ± 0.00 | 0.99 ± 0.01 | 1.00 ± 0.00 | 0.98 ± 0.00 | 1.03 ± 0.01 | 1.10 ± 0.01 | 1.01 ± 0.00 | 0.99 ± 0.01 | 1.02 ± 0.04 | 0.88/0.80 |
| | 256 F | 1.01 ± 0.01 | 1.03 ± 0.01 | 1.00 ± 0.00 | 1.00 ± 0.00 | 1.01 ± 0.00 | 0.99 ± 0.00 | 0.98 ± 0.00 | 1.00 ± 0.03 | 1.01 ± 0.00 | 1.01 ± 0.00 | 1.00 ± 0.01 | 0.50/0.34 |
| 100 w/clusters (C) | 16 C 5 | 1.13 ± 0.01 | 1.03 ± 0.01 | 1.00 ± 0.00 | 1.00 ± 0.00 | 0.99 ± 0.01 | 0.96 ± 0.00 | 1.01 ± 0.02 | 1.23 ± 0.02 | 1.05 ± 0.01 | 0.99 ± 0.06 | 1.04 ± 0.08 | 1.00/0.95 |
| | 16 C 7 | 1.12 ± 0.01 | 1.01 ± 0.01 | 1.00 ± 0.00 | 1.00 ± 0.00 | 0.99 ± 0.00 | 0.96 ± 0.00 | 1.02 ± 0.02 | 1.24 ± 0.05 | 1.03 ± 0.01 | 0.99 ± 0.05 | 1.04 ± 0.08 | 1.00/0.93 |
| 500 diagonal (D) | 16 D | 0.57 ± 0.07 | 0.55 ± 0.03 | 0.83 ± 0.03 | 0.78 ± 0.16 | 0.85 ± 0.04 | 0.68 ± 0.07 | 0.24 ± 0.01 | 0.43 ± 0.01 | 0.76 ± 0.06 | 0.79 ± 0.01 | 0.65 ± 0.20 | 1.00 / 1.00 |
| | 32 D | 0.61 ± 0.12 | 0.55 ± 0.08 | 0.83 ± 0.02 | 0.84 ± 0.12 | 0.91 ± 0.02 | 0.71 ± 0.05 | 0.29 ± 0.05 | 0.47 ± 0.08 | 0.79 ± 0.04 | 0.79 ± 0.01 | 0.68 ± 0.20 | 1.00 / 1.00 |
| | 64 D | 0.73 ± 0.02 | 0.63 ± 0.11 | 0.89 ± 0.04 | 0.97 ± 0.00 | 0.94 ± 0.00 | 0.83 ± 0.05 | 0.45 ± 0.09 | 0.50 ± 0.07 | 0.82 ± 0.02 | 0.80 ± 0.02 | 0.76 ± 0.18 | 1.00 / 0.99 |
| | 128 D | 0.84 ± 0.08 | 0.92 ± 0.02 | 0.97 ± 0.03 | 0.98 ± 0.01 | 0.94 ± 0.00 | 0.88 ± 0.02 | 0.60 ± 0.15 | 0.53 ± 0.01 | 0.85 ± 0.05 | 0.80 ± 0.02 | 0.83 ± 0.15 | 1.00 / 0.98 |
| | 256 D | 0.99 ± 0.03 | 0.99 ± 0.01 | 1.00 ± 0.00 | 1.00 ± 0.00 | 0.96 ± 0.00 | 0.92 ± 0.03 | 0.66 ± 0.06 | 0.84 ± 0.14 | 0.92 ± 0.02 | 0.84 ± 0.01 | 0.91 ± 0.11 | 1.00 / 0.97 |
| 500 full (F) | 16 F | 0.57 ± 0.01 | 0.43 ± 0.07 | 0.78 ± 0.01 | 0.97 ± 0.00 | 0.96 ± 0.00 | 0.83 ± 0.01 | 0.64 ± 0.00 | 0.53 ± 0.03 | 0.83 ± 0.01 | 0.83 ± 0.00 | 0.75 ± 0.17 | 1.00/1.00 |
| | 32 F | 0.79 ± 0.05 | 0.54 ± 0.04 | 0.93 ± 0.02 | 0.98 ± 0.00 | 0.97 ± 0.00 | 0.90 ± 0.01 | 0.69 ± 0.01 | 0.50 ± 0.00 | 0.86 ± 0.02 | 0.83 ± 0.01 | 0.81 ± 0.16 | 0.99/0.99 |
| | 64 F | 1.02 ± 0.00 | 0.96 ± 0.01 | 0.94 ± 0.01 | 1.00 ± 0.01 | 0.96 ± 0.00 | 0.97 ± 0.01 | 0.73 ± 0.01 | 0.54 ± 0.01 | 0.91 ± 0.01 | 0.86 ± 0.01 | 0.89 ± 0.14 | 0.97/0.96 |
| | 128 F | 1.03 ± 0.01 | 0.97 ± 0.02 | 0.99 ± 0.02 | 1.00 ± 0.00 | 0.98 ± 0.00 | 0.96 ± 0.00 | 0.87 ± 0.01 | 1.07 ± 0.02 | 0.98 ± 0.00 | 0.87 ± 0.00 | 0.97 ± 0.06 | 0.88/0.86 |
| | 256 F | 1.03 ± 0.00 | 1.03 ± 0.01 | 1.00 ± 0.00 | 1.00 ± 0.00 | 0.99 ± 0.01 | 0.97 ± 0.01 | 0.99 ± 0.02 | 1.03 ± 0.01 | 1.00 ± 0.01 | 0.87 ± 0.00 | 0.99 ± 0.05 | 0.50/0.47 |
| 500 w/clusters (C) | 16 C 7 | 1.09 | 1.00 | 0.99 | 1.00 | 0.98 | 0.95 | 0.72 | 0.87 | 0.98 | 0.90 | 0.95 | 1.00/0.98 |
| | 16 C 10 | 1.10 | 1.01 | 1.00 | 0.99 | 0.97 | 0.93 | 0.70 | 1.30 | 1.02 | 0.88 | 0.99 | 1.00/0.98 |
| | 16 C 25 | 1.10 | 1.00 | 1.00 | 0.99 | 0.99 | 0.96 | 0.98 | 1.31 | 1.03 | 0.91 | 1.03 | 1.00/0.95 |
| | 64 C 5 | 1.09 | 0.98 | 1.00 | 1.00 | 0.99 | 0.96 | 0.99 | 1.18 | 1.04 | 0.87 | 1.01 | 0.97/0.93 |
| | 64 C 7 | 1.12 | 1.02 | 1.00 | 1.00 | 1.00 | 0.96 | 0.99 | 1.22 | 1.04 | 0.93 | 1.03 | 0.97/0.91 |
| 1000 w/clusters (C) | 16 C 25 | 1.09 | 0.98 | 1.00 | 1.00 | 0.97 | 0.96 | 0.72 | 1.30 | 1.05 | 0.91 | 1.00 | 1.00/0.97 |

Table 7: **Relative** In-Distribution *ROUGE-L* scores for various tasks and methods

| Model Type | Method Type | Tasks | | | | | | | | | | Average | Para. Saved |
|---|---|---|---|---|---|---|---|---|---|---|---|---|---|
| | | task039 | task190 | task280 | task290 | task391 | task442 | task620 | task1342 | task1391 | task1598 | | |
| | base | 24.44 ±0.00 | 1.60 ±0.00 | 19.13 ±0.00 | 39.22 ±0.00 | 10.27 ±0.00 | 35.46 ±0.00 | 7.85 ±0.00 | 6.22 ±0.00 | 17.82 ±0.00 | 38.87 ±0.00 | 20.24 ±13.27 | 1.00/1.00 |
| | lora | 95.00 ±0.00 | 86.00 ±0.00 | 99.00 ±0.00 | 93.67 ±0.00 | 94.33 ±0.00 | 74.88 ±0.00 | 74.40 ±0.00 | 26.68 ±0.00 | 95.00 ±0.00 | 50.32 ±0.00 | 78.87 ±22.56 | 0.00/0.00 |
| TIES | 10 | 76.50 ±0.00 | 49.00 ±1.73 | 44.33 ±4.04 | 9.80 ±0.58 | 78.56 ±0.96 | 35.24 ±0.00 | 51.37 ±0.67 | 15.26 ±0.12 | 77.67 ±1.15 | 42.72 ±0.01 | 48.05 ±23.61 | 1.00 / 1.00 |
| | 50 | 55.80 ±0.00 | 35.00 ±0.00 | 18.00 ±5.20 | 2.42 ±0.50 | 85.78 ±0.96 | 23.03 ±0.00 | 48.03 ±0.00 | 16.50 ±0.00 | 30.00 ±3.46 | 42.47 ±0.02 | 35.70 ±23.01 | 1.00 / 1.00 |
| | 100 | 52.43 ±0.00 | 34.00 ±0.00 | 19.67 ±4.62 | 1.09 ±1.66 | 83.33 ±0.00 | 24.89 ±0.00 | 47.52 ±0.00 | 15.18 ±0.42 | 1.00 ±0.00 | 41.19 ±0.03 | 32.03 ±24.50 | 1.00 / 1.00 |
| | 500 | 35.18 ±0.00 | 22.00 ±0.00 | 1.00 ±0.00 | 0.00 ±0.00 | 78.00 ±0.00 | 21.46 ±0.00 | 42.22 ±0.04 | 9.93 ±0.13 | 1.00 ±0.00 | 21.50 ±0.03 | 23.27 ±23.64 | 1.00 / 1.00 |
| SVD | SVD 2 | 93.15 ±2.77 | 92.24 ±1.85 | 99.09 ±0.18 | 93.44 ±0.14 | 93.89 ±0.35 | 73.74 ±0.51 | 74.55 ±0.98 | 26.80 ±2.79 | 95.06 ±1.35 | 50.21 ±0.44 | 79.11 ±22.72 | 0.88 / 0.88 |
| | SVD 4 | 94.01 ±3.60 | 89.21 ±0.71 | 99.05 ±0.09 | 93.65 ±0.03 | 94.66 ±0.63 | 74.89 ±0.33 | 73.61 ±1.15 | 26.34 ±2.13 | 93.98 ±0.77 | 50.47 ±0.54 | 78.90 ±22.68 | 0.75 / 0.75 |
| | SVD 8 | 95.00 ±0.00 | 87.40 ±0.59 | 99.05 ±0.09 | 93.65 ±0.03 | 94.36 ±0.38 | 74.58 ±0.12 | 75.07 ±0.06 | 26.71 ±0.27 | 95.51 ±1.09 | 50.89 ±0.07 | 81.01 ±21.74 | 0.50 / 0.50 |
| | SVD 16 | 95.00 ±0.00 | 86.00 ±0.00 | 99.00 ±0.00 | 93.67 ±0.00 | 94.33 ±0.00 | 74.90 ±0.03 | 74.23 ±0.18 | 26.68 ±0.00 | 95.00 ±0.00 | 50.30 ±0.02 | 78.36 ±22.97 | 0.00 / 0.00 |
| 10 diagonal (D) | 16 D | 96.67 ±0.58 | 87.00 ±1.00 | 99.00 ±0.00 | 94.00 ±0.67 | 93.11 ±0.38 | 72.08 ±0.06 | 76.26 ±1.19 | 30.11 ±0.79 | 94.00 ±1.73 | 49.30 ±0.46 | 79.15 ±22.18 | 1.00 / 0.90 |
| | 32 D | 95.67 ±0.58 | 90.00 ±1.00 | 99.00 ±0.00 | 93.00 ±0.33 | 94.89 ±0.51 | 73.86 ±0.31 | 71.92 ±0.84 | 27.89 ±0.70 | 94.67 ±0.58 | 50.36 ±0.26 | 79.13 ±22.75 | 1.00 / 0.80 |
| | 64 D | 95.00 ±0.00 | 88.33 ±0.58 | 99.00 ±0.00 | 93.67 ±0.00 | 94.78 ±0.38 | 74.61 ±0.13 | 74.97 ±0.58 | 26.35 ±0.25 | 96.00 ±0.00 | 50.99 ±0.06 | 79.37 ±22.94 | 1.00 / 0.60 |
| | 128 D | 95.00 ±0.00 | 86.67 ±0.58 | 99.00 ±0.00 | 93.67 ±0.00 | 94.33 ±0.00 | 74.92 ±0.13 | 74.96 ±0.51 | 26.45 ±0.23 | 95.00 ±0.00 | 50.21 ±0.12 | 79.02 ±22.84 | 1.00 / 0.20 |
| | 256 D | 95.00 ±0.00 | 86.00 ±0.00 | 99.00 ±0.00 | 93.67 ±0.00 | 94.88 ±0.00 | 74.88 ±0.00 | 74.90 ±0.03 | 26.68 ±0.00 | 95.00 ±0.00 | 50.27 ±0.02 | 78.92 ±22.77 | 1.00 / -0.60 |
| 10 full (F) | 16 F | 97.00 ±0.00 | 91.00 ±1.00 | 99.00 ±0.00 | 93.56 ±0.19 | 93.56 ±0.69 | 73.60 ±0.36 | 74.94 ±1.25 | 28.66 ±0.03 | 96.00 ±1.00 | 50.15 ±0.20 | 79.75 ±22.72 | 1.00/0.90 |
| | 32 F | 96.67 ±0.58 | 89.33 ±0.58 | 99.00 ±0.00 | 93.22 ±0.19 | 94.44 ±0.19 | 74.11 ±0.19 | 71.74 ±0.59 | 26.74 ±0.50 | 94.67 ±0.58 | 50.63 ±0.24 | 79.06 ±23.01 | 0.99/0.79 |
| | 64 F | 95.00 ±0.00 | 88.67 ±0.58 | 99.00 ±0.00 | 93.67 ±0.00 | 94.56 ±0.34 | 74.56 ±0.13 | 75.47 ±0.58 | 26.26 ±0.34 | 96.00 ±0.00 | 50.89 ±0.17 | 79.41 ±22.97 | 0.97/0.57 |
| | 128 F | 95.00 ±0.00 | 86.67 ±0.58 | 99.00 ±0.00 | 93.67 ±0.00 | 94.33 ±0.00 | 75.04 ±0.03 | 74.40 ±0.00 | 26.53 ±0.13 | 95.00 ±0.00 | 50.36 ±0.03 | 79.00 ±22.81 | 0.88/0.07 |
| | 256 F | 95.00 ±0.00 | 86.00 ±0.00 | 99.00 ±0.00 | 93.67 ±0.00 | 94.33 ±0.00 | 74.90 ±0.03 | 74.29 ±0.19 | 26.68 ±0.00 | 95.00 ±0.00 | 50.30 ±0.03 | 78.92 ±22.77 | 0.50/-1.10 |
| 50 diagonal (D) | 16 D | 92.76 ±3.53 | 84.67 ±1.15 | 99.00 ±0.00 | 86.17 ±5.81 | 79.68 ±6.21 | 69.07 ±1.54 | 50.65 ±3.97 | 23.27 ±2.60 | 83.90 ±6.43 | 41.86 ±0.96 | 71.10 ±23.99 | 1.00 / 0.98 |
| | 32 D | 95.33 ±2.08 | 87.33 ±2.08 | 99.00 ±0.00 | 92.60 ±0.29 | 90.32 ±1.04 | 71.16 ±1.47 | 62.51 ±1.64 | 26.60 ±3.54 | 93.33 ±1.15 | 44.35 ±0.41 | 76.25 ±23.81 | 1.00 / 0.96 |
| | 64 D | 97.00 ±0.00 | 90.33 ±1.53 | 99.00 ±0.00 | 93.78 ±0.19 | 93.00 ±0.58 | 72.37 ±0.35 | 73.39 ±0.93 | 29.06 ±0.80 | 95.67 ±0.58 | 45.43 ±0.34 | 78.90 ±23.29 | 1.00 / 0.92 |
| | 128 D | 96.33 ±0.58 | 92.67 ±0.58 | 99.00 ±0.00 | 93.56 ±0.19 | 93.00 ±0.58 | 73.32 ±0.24 | 73.03 ±1.09 | 29.51 ±0.93 | 95.00 ±0.00 | 50.73 ±0.46 | 79.56 ±22.51 | 1.00 / 0.84 |
| | 256 D | 95.67 ±0.58 | 88.33 ±0.58 | 99.00 ±0.00 | 93.56 ±0.19 | 94.67 ±0.67 | 74.82 ±0.29 | 72.36 ±2.07 | 26.90 ±0.75 | 95.33 ±0.58 | 50.73 ±0.46 | 79.14 ±22.90 | 1.00 / 0.68 |
| 50 full (F) | 16 F | 94.06 ±3.54 | 85.67 ±1.15 | 98.67 ±0.58 | 90.35 ±1.37 | 89.90 ±1.91 | 70.32 ±0.66 | 47.62 ±7.28 | 26.88 ±3.96 | 92.33 ±1.53 | 43.68 ±0.24 | 73.95 ±24.73 | 1.00/0.98 |
| | 32 F | 97.00 ±0.00 | 85.67 ±1.53 | 99.00 ±0.00 | 93.67 ±0.00 | 92.22 ±0.69 | 71.88 ±0.30 | 71.01 ±1.02 | 29.07 ±0.65 | 95.67 ±1.53 | 44.97 ±0.41 | 78.02 ±23.18 | 0.99/0.95 |
| | 64 F | 96.67 ±0.58 | 91.00 ±2.00 | 99.00 ±0.00 | 93.56 ±0.19 | 93.22 ±0.51 | 73.16 ±0.41 | 76.28 ±0.51 | 29.67 ±0.12 | 95.33 ±0.58 | 49.31 ±1.00 | 79.72 ±22.50 | 0.97/0.89 |
| | 128 F | 97.00 ±0.00 | 91.00 ±1.00 | 99.00 ±0.00 | 93.33 ±0.00 | 94.11 ±0.51 | 73.51 ±0.23 | 73.17 ±0.58 | 27.53 ±1.12 | 95.00 ±1.00 | 50.56 ±0.06 | 79.42 ±22.93 | 0.88/0.72 |
| | 256 F | 95.00 ±0.00 | 88.00 ±0.00 | 99.00 ±0.00 | 93.67 ±0.00 | 94.44 ±0.19 | 74.25 ±0.00 | 74.97 ±0.58 | 26.79 ±0.00 | 95.00 ±0.00 | 50.86 ±0.19 | 79.30 ±22.82 | 0.50/0.18 |
| 100 diagonal (D) | 16 D | 76.43 ±7.07 | 76.67 ±4.93 | 91.61 ±2.75 | 89.99 ±1.07 | 47.55 ±8.56 | 58.08 ±0.72 | 20.77 ±5.50 | 13.90 ±2.79 | 73.93 ±3.13 | 40.74 ±0.85 | 58.97 ±26.83 | 1.00 / 0.99 |
| | 32 D | 90.10 ±5.85 | 84.00 ±1.00 | 99.00 ±0.00 | 85.52 ±5.34 | 75.69 ±12.75 | 66.62 ±4.18 | 44.66 ±7.26 | 20.49 ±7.07 | 86.67 ±1.86 | 42.01 ±0.94 | 69.48 ±25.14 | 1.00 / 0.98 |
| | 64 D | 95.56 ±2.49 | 86.67 ±0.58 | 99.00 ±0.00 | 92.24 ±1.68 | 90.89 ±1.17 | 70.35 ±0.45 | 65.62 ±4.03 | 29.58 ±2.02 | 91.67 ±2.31 | 43.64 ±1.36 | 76.52 ±23.22 | 1.00 / 0.96 |
| | 128 D | 96.00 ±0.00 | 87.33 ±1.15 | 99.00 ±0.00 | 93.89 ±0.19 | 93.00 ±0.58 | 72.70 ±0.30 | 74.34 ±2.07 | 29.66 ±0.54 | 93.67 ±0.58 | 44.82 ±0.89 | 78.44 ±22.87 | 1.00 / 0.92 |
| | 256 D | 95.00 ±0.00 | 91.00 ±0.00 | 99.00 ±0.00 | 93.56 ±0.19 | 93.11 ±0.19 | 73.05 ±0.20 | 74.52 ±0.95 | 29.67 ±0.67 | 95.33 ±0.58 | 49.42 ±0.65 | 79.37 ±22.38 | 1.00 / 0.84 |
| 100 full (F) | 16 F | 90.70 ±1.07 | 83.00 ±2.65 | 96.00 ±3.00 | 91.22 ±2.94 | 87.94 ±0.54 | 68.72 ±1.05 | 47.57 ±2.54 | 23.75 ±4.33 | 82.33 ±2.08 | 41.51 ±0.67 | 71.27 ±24.23 | 1.00/0.99 |
| | 32 F | 95.33 ±1.53 | 85.00 ±1.00 | 99.00 ±0.00 | 93.50 ±0.22 | 91.44 ±0.84 | 70.94 ±0.02 | 63.64 ±1.98 | 29.82 ±0.81 | 91.67 ±0.58 | 43.94 ±0.18 | 76.43 ±23.01 | 0.99/0.97 |
| | 64 F | 97.00 ±0.00 | 85.67 ±1.53 | 99.00 ±0.00 | 93.78 ±0.19 | 92.56 ±0.19 | 72.11 ±0.08 | 73.29 ±0.64 | 29.15 ±0.24 | 94.33 ±1.53 | 44.97 ±0.05 | 78.18 ±23.03 | 0.97/0.93 |
| | 128 F | 96.33 ±0.58 | 88.67 ±0.58 | 99.00 ±0.00 | 93.00 ±0.00 | 93.89 ±0.19 | 73.11 ±0.36 | 76.56 ±1.01 | 30.45 ±0.35 | 95.00 ±0.00 | 49.81 ±0.34 | 79.74 ±22.47 | 0.88/0.80 |
| | 256 F | 96.33 ±0.58 | 88.67 ±0.58 | 99.00 ±0.00 | 93.67 ±0.00 | 94.89 ±0.19 | 74.40 ±0.16 | 72.90 ±0.12 | 26.77 ±0.68 | 96.00 ±0.00 | 50.83 ±0.09 | 79.35 ±23.04 | 0.50/0.34 |
| 100 w/clusters (C) | 16 C 5 | 98.33 ±0.47 | 89.00 ±0.82 | 99.00 ±0.00 | 93.25 ±0.40 | 92.89 ±0.87 | 72.32 ±0.36 | 77.08 ±1.67 | 28.26 ±0.38 | 96.67 ±0.47 | 68.30 ±15.72 | 81.51 ±20.63 | 1.00/0.95 |
| | 16 C 7 | 97.67 ±0.47 | 87.00 ±0.82 | 99.00 ±0.00 | 93.46 ±0.29 | 93.11 ±0.68 | 72.52 ±0.43 | 77.66 ±1.30 | 28.51 ±1.26 | 95.33 ±0.47 | 68.46 ±14.94 | 81.27 ±20.35 | 1.00/0.93 |
| 500 diagonal (D) | 16 D | 54.44 ±6.87 | 47.00 ±2.83 | 82.21 ±3.59 | 73.38 ±14.97 | 80.08 ±3.71 | 51.02 ±5.31 | 17.49 ±1.10 | 11.58 ±0.21 | 72.67 ±6.03 | 39.65 ±0.28 | 53.16 ±24.97 | 1.00 / 1.00 |
| | 32 D | 58.08 ±11.52 | 47.00 ±7.07 | 82.06 ±1.69 | 78.62 ±11.23 | 85.57 ±1.48 | 52.98 ±3.81 | 21.73 ±3.95 | 12.53 ±2.26 | 75.33 ±4.04 | 39.78 ±0.42 | 55.66 ±25.48 | 1.00 / 1.00 |
| | 64 D | 69.21 ±2.03 | 54.50 ±9.19 | 88.33 ±4.04 | 91.11 ±0.38 | 88.78 ±0.38 | 62.36 ±3.52 | 33.36 ±6.69 | 13.34 ±1.86 | 77.67 ±2.31 | 40.42 ±0.98 | 62.16 ±26.05 | 1.00 / 0.99 |
| | 128 D | 79.77 ±0.37 | 79.50 ±2.12 | 95.89 ±2.83 | 91.89 ±1.39 | 88.67 ±0.00 | 65.92 ±1.79 | 44.88 ±10.98 | 14.14 ±0.19 | 81.00 ±5.00 | 40.34 ±0.80 | 67.82 ±26.35 | 1.00 / 0.98 |
| | 256 D | 93.83 ±2.52 | 85.00 ±0.00 | 99.00 ±0.00 | 93.78 ±0.19 | 90.56 ±0.38 | 68.95 ±1.92 | 49.39 ±4.36 | 22.33 ±3.78 | 87.33 ±2.31 | 42.15 ±0.73 | 72.83 ±25.93 | 1.00 / 0.97 |
| 500 full (F) | 16 F | 54.30 ±1.13 | 37.00 ±5.66 | 77.67 ±0.58 | 91.00 ±0.00 | 90.56 ±0.19 | 62.47 ±0.79 | 47.56 ±0.29 | 14.18 ±0.67 | 79.00 ±1.00 | 41.58 ±0.23 | 60.31 ±24.42 | 1.00/1.00 |
| | 32 F | 75.10 ±4.92 | 46.50 ±3.54 | 91.67 ±1.53 | 91.56 ±0.19 | 91.56 ±0.38 | 67.37 ±0.83 | 51.17 ±0.81 | 13.44 ±0.02 | 81.67 ±1.53 | 41.92 ±0.42 | 65.84 ±25.64 | 0.99/0.99 |
| | 64 F | 96.94 ±0.42 | 82.50 ±0.71 | 93.33 ±0.58 | 93.89 ±0.69 | 90.67 ±0.00 | 72.30 ±0.71 | 54.63 ±0.79 | 14.49 ±0.27 | 86.33 ±0.58 | 43.16 ±0.08 | 72.49 ±26.64 | 0.97/0.96 |
| | 128 F | 97.67 ±0.58 | 83.50 ±2.12 | 98.00 ±0.00 | 93.56 ±0.19 | 92.00 ±0.00 | 71.92 ±0.19 | 65.02 ±0.81 | 28.49 ±0.55 | 93.00 ±0.00 | 43.85 ±0.12 | 76.47 ±23.77 | 0.88/0.86 |
| | 256 F | 98.00 ±0.00 | 88.50 ±0.71 | 99.00 ±0.00 | 93.78 ±0.19 | 93.00 ±0.24 | 72.45 ±0.38 | 73.77 ±1.21 | 27.59 ±0.39 | 93.00 ±0.58 | 43.81 ±0.17 | 78.18 ±24.16 | 0.50/0.47 |
| 500 w/clusters (C) | 16 C 7 | 95.00 | 86.00 | 98.00 | 93.67 | 91.67 | 71.19 | 54.69 | 20.03 | 90.00 | 46.34 | 74.66 | 1.00/0.98 |
| | 16 C 10 | 96.00 | 87.00 | 99.00 | 93.00 | 91.33 | 69.93 | 53.48 | 30.09 | 94.00 | 44.89 | 75.87 | 1.00/0.98 |
| | 16 C 25 | 96.00 | 86.00 | 99.00 | 92.71 | 93.00 | 72.13 | 74.59 | 30.21 | 95.00 | 46.66 | 78.53 | 1.00/0.95 |
| | 64 C 5 | 95.00 | 84.00 | 99.00 | 93.67 | 92.67 | 72.32 | 75.60 | 27.17 | 96.00 | 44.43 | 77.99 | 0.97/0.93 |
| | 64 C 7 | 98.00 | 88.00 | 99.00 | 94.00 | 93.33 | 72.18 | 75.83 | 28.14 | 96.00 | 47.68 | 79.22 | 0.97/0.91 |
| 1000 w/clusters (C) | 16 C 25 | 95.00 | 84.00 | 99.00 | 93.67 | 90.67 | 72.20 | 55.04 | 29.97 | 97.00 | 46.86 | 76.34 | 1.00/0.97 |

Table 8: **Absolute** In-Distribution ROUGE-L scores for various tasks and methods

| Model Type | Method Type | task039 | task190 | task280 | task290 | task391 | task442 | task620 | task1342 | task1391 | task1598 | Average | Para. Saved |
|---|---|---|---|---|---|---|---|---|---|---|---|---|---|
| | base | 0.26 ± 0.00 | 0.02 ± 0.00 | 0.19 ± 0.00 | 0.42 ± 0.00 | 0.11 ± 0.00 | 0.51 ± 0.00 | 0.11 ± 0.00 | 0.26 ± 0.00 | 0.19 ± 0.00 | 0.80 ± 0.00 | 0.29 ± 0.22 | 1.00/1.00 |
| | lora | 1.00 ± 0.00 | 1.00 ± 0.00 | 1.00 ± 0.00 | 1.00 ± 0.00 | 1.00 ± 0.00 | 1.00 ± 0.00 | 1.00 ± 0.00 | 1.00 ± 0.00 | 1.00 ± 0.00 | 1.00 ± 0.00 | 1.00 ± 0.00 | 0.00/0.00 |
| TIES | 10 | 0.81 ± 0.00 | 0.57 ± 0.02 | 0.45 ± 0.04 | 0.10 ± 0.01 | 0.83 ± 0.01 | 0.52 ± 0.00 | 0.71 ± 0.01 | 0.58 ± 0.00 | 0.82 ± 0.01 | 0.80 ± 0.00 | 0.62 ± 0.22 | 1.00 / 1.00 |
| | 50 | 0.59 ± 0.00 | 0.41 ± 0.00 | 0.18 ± 0.05 | 0.03 ± 0.01 | 0.91 ± 0.01 | 0.34 ± 0.00 | 0.67 ± 0.00 | 0.62 ± 0.00 | 0.32 ± 0.04 | 0.78 ± 0.00 | 0.48 ± 0.27 | 1.00 / 1.00 |
| | 100 | 0.55 ± 0.00 | 0.40 ± 0.01 | 0.20 ± 0.05 | 0.01 ± 0.02 | 0.88 ± 0.00 | 0.36 ± 0.00 | 0.65 ± 0.00 | 0.57 ± 0.02 | 0.01 ± 0.00 | 0.78 ± 0.00 | 0.44 ± 0.29 | 1.00 / 1.00 |
| | 500 | 0.37 ± 0.00 | 0.26 ± 0.00 | 0.01 ± 0.00 | 0.00 ± 0.00 | 0.83 ± 0.00 | 0.31 ± 0.00 | 0.58 ± 0.00 | 0.37 ± 0.00 | 0.01 ± 0.00 | 0.41 ± 0.00 | 0.32 ± 0.26 | 1.00 / 1.00 |
| SVD | SVD 2 | 0.98 ± 0.03 | 1.07 ± 0.02 | 1.00 ± 0.00 | 1.00 ± 0.00 | 1.00 ± 0.00 | 0.99 ± 0.00 | 1.01 ± 0.01 | 1.00 ± 0.10 | 1.00 ± 0.01 | 0.99 ± 0.01 | 1.00 ± 0.04 | 0.88 / 0.88 |
| | SVD 4 | 0.99 ± 0.04 | 1.04 ± 0.01 | 1.00 ± 0.00 | 1.00 ± 0.00 | 1.00 ± 0.01 | 1.00 ± 0.00 | 0.99 ± 0.01 | 0.99 ± 0.08 | 0.99 ± 0.01 | 1.01 ± 0.00 | 1.00 ± 0.03 | 0.75 / 0.75 |
| | SVD 8 | 1.00 ± 0.00 | 1.02 ± 0.01 | 1.00 ± 0.00 | 1.00 ± 0.00 | 1.00 ± 0.00 | 1.00 ± 0.00 | 1.01 ± 0.00 | 1.00 ± 0.01 | 1.01 ± 0.00 | 1.01 ± 0.00 | 1.00 ± 0.01 | 0.50 / 0.50 |
| | SVD 16 | 1.00 ± 0.00 | 1.00 ± 0.00 | 1.00 ± 0.00 | 1.00 ± 0.00 | 1.00 ± 0.00 | 1.00 ± 0.00 | 1.00 ± 0.00 | 1.00 ± 0.00 | 1.00 ± 0.00 | 1.00 ± 0.00 | 1.00 ± 0.00 | 0.00 / 0.00 |
| 10 diagonal (D) | 16 D | 1.02 ± 0.01 | 1.01 ± 0.01 | 1.00 ± 0.00 | 1.00 ± 0.01 | 0.99 ± 0.00 | 0.97 ± 0.00 | 1.03 ± 0.02 | 1.12 ± 0.03 | 0.99 ± 0.02 | 0.99 ± 0.00 | 1.01 ± 0.04 | 1.00 / 0.90 |
| | 32 D | 1.01 ± 0.01 | 1.05 ± 0.01 | 1.00 ± 0.00 | 0.99 ± 0.00 | 1.01 ± 0.01 | 0.99 ± 0.00 | 0.97 ± 0.01 | 1.04 ± 0.03 | 1.00 ± 0.01 | 1.01 ± 0.01 | 1.01 ± 0.02 | 1.00 / 0.80 |
| | 64 D | 1.00 ± 0.00 | 1.03 ± 0.01 | 1.00 ± 0.00 | 1.00 ± 0.00 | 1.00 ± 0.00 | 1.00 ± 0.00 | 1.01 ± 0.04 | 0.99 ± 0.01 | 1.01 ± 0.00 | 1.00 ± 0.01 | 1.00 ± 0.01 | 1.00 / 0.60 |
| | 128 D | 1.00 ± 0.00 | 1.01 ± 0.01 | 1.00 ± 0.00 | 1.00 ± 0.00 | 1.00 ± 0.00 | 1.00 ± 0.00 | 1.01 ± 0.01 | 0.99 ± 0.01 | 1.00 ± 0.00 | 1.00 ± 0.00 | 1.00 ± 0.01 | 1.00 / 0.20 |
| | 256 D | 1.00 ± 0.00 | 1.00 ± 0.00 | 1.00 ± 0.00 | 1.00 ± 0.00 | 1.00 ± 0.00 | 1.00 ± 0.00 | 1.00 ± 0.00 | 1.00 ± 0.00 | 1.00 ± 0.00 | 1.00 ± 0.00 | 1.00 ± 0.00 | 1.00 / -0.60 |
| 10 full (F) | 16 F | 1.02 ± 0.00 | 1.06 ± 0.01 | 1.00 ± 0.00 | 1.00 ± 0.00 | 0.99 ± 0.01 | 0.99 ± 0.00 | 1.01 ± 0.02 | 1.07 ± 0.00 | 1.01 ± 0.01 | 1.00 ± 0.01 | 1.02 ± 0.03 | 1.00/0.90 |
| | 32 F | 1.02 ± 0.01 | 1.04 ± 0.01 | 1.00 ± 0.00 | 1.00 ± 0.00 | 1.00 ± 0.00 | 0.99 ± 0.00 | 0.96 ± 0.01 | 1.00 ± 0.02 | 1.00 ± 0.01 | 1.01 ± 0.00 | 1.00 ± 0.02 | 0.99/0.79 |
| | 64 F | 1.00 ± 0.00 | 1.03 ± 0.01 | 1.00 ± 0.00 | 1.00 ± 0.00 | 1.00 ± 0.00 | 1.00 ± 0.00 | 1.01 ± 0.01 | 0.98 ± 0.01 | 1.01 ± 0.00 | 1.01 ± 0.00 | 1.00 ± 0.01 | 0.97/0.57 |
| | 128 F | 1.00 ± 0.00 | 1.01 ± 0.01 | 1.00 ± 0.00 | 1.00 ± 0.00 | 1.00 ± 0.00 | 1.00 ± 0.00 | 1.00 ± 0.00 | 0.99 ± 0.00 | 1.00 ± 0.00 | 1.00 ± 0.00 | 1.00 ± 0.00 | 0.88/0.07 |
| | 256 F | 1.00 ± 0.00 | 1.00 ± 0.00 | 1.00 ± 0.00 | 1.00 ± 0.00 | 1.00 ± 0.00 | 1.00 ± 0.00 | 1.00 ± 0.00 | 1.00 ± 0.00 | 1.00 ± 0.00 | 1.00 ± 0.00 | 1.00 ± 0.00 | 0.50/-1.10 |
| 50 diagonal (D) | 16 D | 0.98 ± 0.04 | 0.98 ± 0.01 | 1.00 ± 0.00 | 0.92 ± 0.06 | 0.85 ± 0.06 | 0.94 ± 0.02 | 0.69 ± 0.05 | 0.88 ± 0.10 | 0.88 ± 0.07 | 0.86 ± 0.01 | 0.90 ± 0.10 | 1.00 / 0.98 |
| | 32 D | 1.00 ± 0.02 | 1.02 ± 0.02 | 1.00 ± 0.00 | 0.99 ± 0.00 | 0.96 ± 0.01 | 0.96 ± 0.02 | 0.85 ± 0.02 | 1.00 ± 0.12 | 0.98 ± 0.01 | 0.90 ± 0.00 | 0.97 ± 0.06 | 1.00 / 0.96 |
| | 64 D | 1.02 ± 0.03 | 1.05 ± 0.02 | 1.00 ± 0.00 | 1.00 ± 0.00 | 0.99 ± 0.01 | 0.97 ± 0.01 | 0.99 ± 0.01 | 1.03 ± 0.02 | 1.01 ± 0.01 | 0.94 ± 0.00 | 1.01 ± 0.04 | 1.00 / 0.92 |
| | 128 D | 1.01 ± 0.01 | 1.08 ± 0.03 | 1.00 ± 0.00 | 1.00 ± 0.00 | 0.99 ± 0.00 | 0.98 ± 0.00 | 0.98 ± 0.02 | 1.10 ± 0.03 | 1.00 ± 0.00 | 1.01 ± 0.01 | 1.02 ± 0.04 | 1.00 / 0.84 |
| | 256 D | 1.01 ± 0.01 | 1.03 ± 0.01 | 1.00 ± 0.00 | 1.00 ± 0.00 | 1.00 ± 0.01 | 1.00 ± 0.00 | 0.97 ± 0.03 | 1.00 ± 0.03 | 1.00 ± 0.01 | 1.01 ± 0.00 | 1.00 ± 0.02 | 1.00 / 0.68 |
| 50 full (F) | 16 F | 0.99 ± 0.04 | 1.00 ± 0.01 | 1.00 ± 0.01 | 0.96 ± 0.01 | 0.95 ± 0.02 | 0.95 ± 0.01 | 0.65 ± 0.09 | 1.01 ± 0.15 | 0.97 ± 0.02 | 0.88 ± 0.01 | 0.94 ± 0.11 | 1.00/0.98 |
| | 32 F | 1.02 ± 0.00 | 1.00 ± 0.02 | 1.00 ± 0.00 | 1.00 ± 0.00 | 0.98 ± 0.01 | 0.97 ± 0.00 | 0.96 ± 0.01 | 1.09 ± 0.03 | 1.01 ± 0.02 | 0.93 ± 0.00 | 0.99 ± 0.04 | 0.99/0.95 |
| | 64 F | 1.02 ± 0.01 | 1.06 ± 0.01 | 1.00 ± 0.00 | 1.00 ± 0.00 | 0.99 ± 0.01 | 0.98 ± 0.00 | 1.03 ± 0.00 | 1.11 ± 0.00 | 1.00 ± 0.01 | 0.99 ± 0.01 | 1.02 ± 0.04 | 0.97/0.89 |
| | 128 F | 1.02 ± 0.00 | 1.06 ± 0.01 | 1.00 ± 0.00 | 1.00 ± 0.00 | 1.00 ± 0.01 | 0.98 ± 0.00 | 0.98 ± 0.01 | 1.03 ± 0.04 | 1.00 ± 0.01 | 1.01 ± 0.00 | 1.01 ± 0.02 | 0.88/0.72 |
| | 256 F | 1.00 ± 0.00 | 1.02 ± 0.00 | 1.00 ± 0.00 | 1.00 ± 0.00 | 1.00 ± 0.00 | 0.99 ± 0.00 | 1.01 ± 0.01 | 1.00 ± 0.03 | 1.00 ± 0.01 | 1.01 ± 0.00 | 1.00 ± 0.01 | 0.50/0.18 |
| 100 diagonal (D) | 16 D | 0.80 ± 0.07 | 0.89 ± 0.06 | 0.93 ± 0.03 | 0.96 ± 0.01 | 0.51 ± 0.09 | 0.81 ± 0.07 | 0.30 ± 0.07 | 0.54 ± 0.11 | 0.78 ± 0.03 | 0.83 ± 0.02 | 0.73 ± 0.21 | 1.00 / 0.99 |
| | 32 D | 0.95 ± 0.06 | 0.98 ± 0.51 | 1.00 ± 0.00 | 0.91 ± 0.06 | 0.80 ± 0.13 | 0.91 ± 0.05 | 0.62 ± 0.10 | 0.78 ± 0.25 | 0.91 ± 0.02 | 0.85 ± 0.01 | 0.87 ± 0.14 | 1.00 / 0.98 |
| | 64 D | 1.01 ± 0.03 | 1.01 ± 0.01 | 1.00 ± 0.00 | 0.98 ± 0.02 | 0.96 ± 0.01 | 0.95 ± 0.01 | 0.90 ± 0.05 | 1.11 ± 0.07 | 0.96 ± 0.02 | 0.88 ± 0.02 | 0.98 ± 0.07 | 1.00 / 0.96 |
| | 128 D | 1.01 ± 0.00 | 1.02 ± 0.01 | 1.00 ± 0.00 | 1.00 ± 0.00 | 0.99 ± 0.01 | 0.98 ± 0.00 | 1.00 ± 0.03 | 1.11 ± 0.02 | 0.99 ± 0.01 | 0.92 ± 0.00 | 1.00 ± 0.05 | 1.00 / 0.92 |
| | 256 D | 1.00 ± 0.00 | 1.06 ± 0.00 | 1.00 ± 0.00 | 1.00 ± 0.00 | 0.99 ± 0.00 | 0.98 ± 0.00 | 1.00 ± 0.01 | 1.11 ± 0.03 | 1.00 ± 0.00 | 0.99 ± 0.02 | 1.01 ± 0.04 | 1.00 / 0.84 |
| 100 full (F) | 16 F | 0.95 ± 0.01 | 0.97 ± 0.03 | 0.97 ± 0.03 | 0.97 ± 0.03 | 0.93 ± 0.01 | 0.93 ± 0.01 | 0.66 ± 0.03 | 0.90 ± 0.16 | 0.87 ± 0.02 | 0.85 ± 0.01 | 0.90 ± 0.10 | 1.00/0.99 |
| | 32 F | 1.00 ± 0.02 | 0.99 ± 0.01 | 1.00 ± 0.00 | 1.00 ± 0.00 | 0.97 ± 0.01 | 0.96 ± 0.00 | 0.87 ± 0.03 | 1.12 ± 0.03 | 0.96 ± 0.01 | 0.89 ± 0.00 | 0.98 ± 0.07 | 0.99/0.97 |
| | 64 F | 1.02 ± 0.01 | 1.00 ± 0.02 | 1.00 ± 0.00 | 1.00 ± 0.00 | 0.98 ± 0.00 | 0.97 ± 0.00 | 0.99 ± 0.01 | 1.10 ± 0.01 | 0.99 ± 0.02 | 0.93 ± 0.01 | 1.00 ± 0.04 | 0.97/0.93 |
| | 128 F | 1.01 ± 0.01 | 1.05 ± 0.03 | 1.00 ± 0.00 | 0.99 ± 0.00 | 1.00 ± 0.00 | 0.98 ± 0.00 | 1.03 ± 0.00 | 1.10 ± 0.01 | 1.01 ± 0.00 | 1.00 ± 0.00 | 1.02 ± 0.03 | 0.88/0.80 |
| | 256 F | 1.01 ± 0.01 | 1.03 ± 0.01 | 1.00 ± 0.00 | 1.00 ± 0.00 | 1.01 ± 0.00 | 1.00 ± 0.00 | 0.98 ± 0.00 | 1.00 ± 0.03 | 1.01 ± 0.00 | 1.01 ± 0.00 | 1.00 ± 0.01 | 0.50/0.34 |
| 100 w/clusters (C) | 16 C 5 | 1.13 ± 0.01 | 1.03 ± 0.01 | 1.00 ± 0.00 | 1.00 ± 0.00 | 0.99 ± 0.01 | 0.97 ± 0.00 | 1.01 ± 0.02 | 1.22 ± 0.02 | 1.05 ± 0.01 | 1.00 ± 0.05 | 1.04 ± 0.07 | 1.00/0.95 |
| | 16 C 7 | 1.12 ± 0.01 | 1.01 ± 0.01 | 1.00 ± 0.00 | 1.00 ± 0.00 | 0.99 ± 0.00 | 0.97 ± 0.00 | 1.02 ± 0.02 | 1.22 ± 0.05 | 1.03 ± 0.01 | 1.01 ± 0.03 | 1.04 ± 0.07 | 1.00/0.93 |
| 500 diagonal (D) | 16 D | 0.57 ± 0.07 | 0.55 ± 0.03 | 0.83 ± 0.04 | 0.78 ± 0.16 | 0.85 ± 0.04 | 0.73 ± 0.07 | 0.24 ± 0.02 | 0.45 ± 0.01 | 0.76 ± 0.06 | 0.81 ± 0.00 | 0.66 ± 0.20 | 1.00 / 1.00 |
| | 32 D | 0.61 ± 0.12 | 0.55 ± 0.08 | 0.83 ± 0.02 | 0.84 ± 0.12 | 0.91 ± 0.02 | 0.75 ± 0.05 | 0.30 ± 0.05 | 0.49 ± 0.07 | 0.79 ± 0.04 | 0.82 ± 0.01 | 0.69 ± 0.20 | 1.00 / 1.00 |
| | 64 D | 0.73 ± 0.02 | 0.63 ± 0.11 | 0.89 ± 0.04 | 0.97 ± 0.00 | 0.94 ± 0.00 | 0.86 ± 0.03 | 0.46 ± 0.09 | 0.51 ± 0.07 | 0.82 ± 0.02 | 0.83 ± 0.01 | 0.77 ± 0.18 | 1.00 / 0.99 |
| | 128 D | 0.84 ± 0.00 | 0.92 ± 0.02 | 0.97 ± 0.03 | 0.98 ± 0.01 | 0.94 ± 0.00 | 0.90 ± 0.02 | 0.62 ± 0.14 | 0.54 ± 0.01 | 0.85 ± 0.05 | 0.83 ± 0.01 | 0.84 ± 0.15 | 1.00 / 0.98 |
| | 256 D | 0.99 ± 0.03 | 0.99 ± 0.00 | 1.00 ± 0.00 | 1.00 ± 0.00 | 0.96 ± 0.00 | 0.93 ± 0.02 | 0.68 ± 0.05 | 0.85 ± 0.14 | 0.92 ± 0.02 | 0.85 ± 0.00 | 0.92 ± 0.11 | 1.00 / 0.97 |
| 500 full (F) | 16 F | 0.57 ± 0.01 | 0.43 ± 0.07 | 0.78 ± 0.01 | 0.97 ± 0.00 | 0.96 ± 0.00 | 0.86 ± 0.01 | 0.65 ± 0.02 | 0.55 ± 0.02 | 0.83 ± 0.01 | 0.84 ± 0.00 | 0.76 ± 0.17 | 1.00/1.00 |
| | 32 F | 0.79 ± 0.05 | 0.54 ± 0.04 | 0.93 ± 0.02 | 0.98 ± 0.00 | 0.97 ± 0.00 | 0.92 ± 0.00 | 0.70 ± 0.01 | 0.52 ± 0.00 | 0.86 ± 0.02 | 0.85 ± 0.00 | 0.81 ± 0.16 | 0.99/0.99 |
| | 64 F | 1.02 ± 0.00 | 0.96 ± 0.01 | 0.94 ± 0.01 | 1.00 ± 0.01 | 0.96 ± 0.00 | 0.97 ± 0.01 | 0.74 ± 0.01 | 0.87 ± 0.00 | 0.87 ± 0.00 | 0.89 ± 0.14 | 0.89 ± 0.14 | 0.97/0.96 |
| | 128 F | 1.03 ± 0.01 | 0.97 ± 0.02 | 0.99 ± 0.02 | 1.00 ± 0.00 | 0.98 ± 0.00 | 0.97 ± 0.00 | 0.88 ± 0.01 | 1.07 ± 0.02 | 0.98 ± 0.00 | 0.90 ± 0.00 | 0.98 ± 0.05 | 0.88/0.86 |
| | 256 F | 1.03 ± 0.00 | 1.03 ± 0.01 | 1.00 ± 0.00 | 1.00 ± 0.00 | 0.99 ± 0.01 | 0.97 ± 0.00 | 1.00 ± 0.02 | 1.04 ± 0.02 | 1.00 ± 0.01 | 0.93 ± 0.00 | 1.00 ± 0.03 | 0.50/0.47 |
| 500 w/clusters (C) | 16 C 7 | 1.09 | 1.00 | 0.99 | 1.00 | 0.98 | 0.96 | 0.72 | 0.88 | 0.98 | 0.93 | 0.95 | 1.00/0.98 |
| | 16 C 10 | 1.10 | 1.01 | 1.00 | 0.99 | 0.97 | 0.94 | 0.72 | 1.29 | 1.02 | 0.92 | 1.00 | 1.00/0.98 |
| | 16 C 25 | 1.10 | 1.00 | 1.00 | 0.99 | 0.99 | 0.97 | 0.98 | 1.30 | 1.03 | 0.96 | 1.03 | 1.00/0.95 |
| | 64 C 5 | 1.09 | 0.98 | 1.00 | 1.00 | 0.99 | 0.97 | 0.99 | 1.17 | 1.04 | 0.93 | 1.02 | 0.97/0.93 |
| | 64 C 7 | 1.12 | 1.02 | 1.00 | 1.00 | 1.00 | 0.97 | 1.00 | 1.22 | 1.04 | 0.99 | 1.04 | 0.97/0.91 |
| 1000 w/clusters (C) | 16 C 25 | 1.09 | 0.98 | 1.00 | 1.00 | 0.97 | 0.97 | 0.74 | 1.29 | 1.05 | 0.94 | 1.00 | 1.00/0.97 |

Table 9: **Relative** In-Distribution ROUGE-1 scores for various tasks and methods

| Model Type | Method Type | Tasks | | | | | | | | | | Average | Para. Saved |
|---|---|---|---|---|---|---|---|---|---|---|---|---|---|
| | | task039 | task190 | task280 | task290 | task391 | task442 | task620 | task1342 | task1391 | task1598 | | |
| | base | 24.44 ± 0.00 | 1.60 ± 0.00 | 19.13 ± 0.00 | 39.22 ± 0.00 | 10.42 ± 0.00 | 39.88 ± 0.00 | 8.05 ± 0.00 | 6.96 ± 0.00 | 17.82 ± 0.00 | 55.03 ± 0.00 | 22.43 ± 16.49 | 1.00 / 1.00 |
| | lora | 95.00 ± 0.00 | 86.00 ± 0.00 | 99.00 ± 0.00 | 93.67 ± 0.00 | 94.33 ± 0.00 | 78.43 ± 0.00 | 74.90 ± 0.00 | 26.87 ± 0.00 | 95.00 ± 0.00 | 68.66 ± 0.00 | 81.14 ± 20.67 | 0.00/0.00 |
| TIES | 10 | 76.50 ± 0.00 | 49.00 ± 1.73 | 44.33 ± 4.04 | 9.80 ± 0.58 | 78.56 ± 0.96 | 40.44 ± 0.00 | 53.10 ± 0.67 | 15.48 ± 0.12 | 77.67 ± 1.15 | 54.89 ± 0.06 | 49.98 ± 23.33 | 1.00 / 1.00 |
| | 50 | 55.80 ± 0.00 | 35.00 ± 0.00 | 18.00 ± 5.20 | 2.42 ± 0.50 | 85.78 ± 0.96 | 26.75 ± 0.00 | 49.96 ± 0.00 | 16.73 ± 0.00 | 30.00 ± 3.46 | 53.87 ± 0.02 | 37.43 ± 23.49 | 1.00 / 1.00 |
| | 100 | 52.43 ± 0.00 | 34.00 ± 0.00 | 19.67 ± 4.62 | 1.09 ± 1.66 | 83.33 ± 0.00 | 28.57 ± 0.00 | 48.89 ± 0.00 | 15.18 ± 0.42 | 1.00 ± 0.00 | 53.44 ± 0.02 | 33.76 ± 25.22 | 1.00 / 1.00 |
| | 500 | 35.18 ± 0.00 | 22.00 ± 0.00 | 1.00 ± 0.00 | 0.00 ± 0.00 | 78.00 ± 0.00 | 24.32 ± 0.00 | 43.80 ± 0.04 | 9.96 ± 0.13 | 1.00 ± 0.00 | 27.90 ± 0.03 | 24.40 ± 23.79 | 1.00 / 1.00 |
| SVD | SVD 2 | 93.15 ± 2.77 | 92.24 ± 1.85 | 99.09 ± 0.18 | 93.44 ± 0.14 | 93.89 ± 0.35 | 77.33 ± 0.29 | 75.40 ± 1.01 | 26.90 ± 2.68 | 95.06 ± 1.35 | 67.71 ± 0.49 | 81.33 ± 20.85 | 0.88 / 0.88 |
| | SVD 4 | 94.01 ± 3.60 | 89.21 ± 0.71 | 99.05 ± 0.09 | 93.65 ± 0.03 | 94.66 ± 0.63 | 78.42 ± 0.23 | 74.09 ± 1.12 | 26.47 ± 2.06 | 93.98 ± 0.77 | 69.37 ± 0.21 | 81.22 ± 20.80 | 0.75 / 0.75 |
| | SVD 8 | 95.00 ± 0.00 | 87.40 ± 0.59 | 99.05 ± 0.09 | 93.65 ± 0.03 | 94.36 ± 0.38 | 78.21 ± 0.10 | 75.57 ± 0.00 | 26.88 ± 0.27 | 95.51 ± 1.09 | 69.33 ± 0.08 | 83.02 ± 19.93 | 0.50 / 0.50 |
| | SVD 16 | 95.00 ± 0.00 | 86.00 ± 0.00 | 99.00 ± 0.00 | 93.67 ± 0.00 | 94.33 ± 0.00 | 78.44 ± 0.03 | 74.73 ± 0.18 | 26.87 ± 0.00 | 95.00 ± 0.00 | 68.62 ± 0.04 | 80.76 ± 21.05 | 0.00 / 0.00 |
| 10 diagonal (D) | 16 D | 96.67 ± 0.58 | 87.00 ± 1.00 | 99.00 ± 0.00 | 94.00 ± 0.67 | 93.11 ± 0.38 | 76.08 ± 0.17 | 77.26 ± 1.47 | 30.15 ± 0.72 | 94.00 ± 1.73 | 68.25 ± 0.18 | 81.55 ± 20.03 | 1.00 / 0.90 |
| | 32 D | 95.67 ± 0.58 | 90.00 ± 1.00 | 99.00 ± 0.00 | 93.00 ± 0.33 | 94.89 ± 0.51 | 77.46 ± 0.36 | 72.53 ± 1.00 | 27.98 ± 0.71 | 94.67 ± 0.58 | 69.16 ± 0.41 | 81.44 ± 20.80 | 1.00 / 0.80 |
| | 64 D | 95.00 ± 0.00 | 88.33 ± 0.58 | 99.00 ± 0.00 | 93.67 ± 0.00 | 94.78 ± 0.38 | 78.28 ± 0.07 | 75.47 ± 0.58 | 26.53 ± 0.25 | 96.00 ± 0.00 | 69.36 ± 0.05 | 81.64 ± 21.06 | 1.00 / 0.60 |
| | 128 D | 95.00 ± 0.00 | 86.67 ± 0.58 | 99.00 ± 0.00 | 93.67 ± 0.00 | 94.33 ± 0.00 | 78.45 ± 0.16 | 75.46 ± 0.51 | 26.64 ± 0.23 | 95.00 ± 0.00 | 68.70 ± 0.14 | 81.29 ± 20.92 | 1.00 / 0.20 |
| | 256 D | 95.00 ± 0.00 | 86.00 ± 0.00 | 99.00 ± 0.00 | 93.67 ± 0.00 | 94.33 ± 0.00 | 78.43 ± 0.00 | 74.90 ± 0.00 | 26.87 ± 0.00 | 95.00 ± 0.00 | 68.59 ± 0.03 | 81.18 ± 20.86 | 1.00 / -0.60 |
| 10 full (F) | 16 F | 97.00 ± 0.00 | 91.00 ± 1.00 | 99.00 ± 0.00 | 93.56 ± 0.19 | 93.56 ± 0.69 | 77.64 ± 0.25 | 75.78 ± 1.25 | 28.71 ± 0.09 | 96.00 ± 1.00 | 68.69 ± 0.08 | 82.09 ± 20.68 | 1.00/0.90 |
| | 32 F | 96.67 ± 0.58 | 89.33 ± 0.58 | 99.00 ± 0.00 | 93.22 ± 0.19 | 94.44 ± 0.19 | 77.84 ± 0.21 | 72.24 ± 0.59 | 26.84 ± 0.50 | 94.67 ± 0.58 | 69.55 ± 0.08 | 81.38 ± 21.11 | 0.99/0.79 |
| | 64 F | 95.00 ± 0.00 | 88.67 ± 0.58 | 99.00 ± 0.00 | 93.67 ± 0.00 | 94.56 ± 0.38 | 78.19 ± 0.08 | 76.58 ± 0.00 | 26.43 ± 0.34 | 96.00 ± 0.00 | 69.38 ± 0.11 | 81.69 ± 21.07 | 0.97/0.57 |
| | 128 F | 95.00 ± 0.00 | 86.67 ± 0.58 | 99.00 ± 0.00 | 93.67 ± 0.00 | 94.33 ± 0.00 | 78.46 ± 0.03 | 74.90 ± 0.00 | 26.72 ± 0.13 | 95.00 ± 0.00 | 68.65 ± 0.03 | 81.24 ± 20.91 | 0.88/0.07 |
| | 256 F | 95.00 ± 0.00 | 86.00 ± 0.00 | 99.00 ± 0.00 | 93.67 ± 0.00 | 94.33 ± 0.00 | 78.44 ± 0.03 | 74.79 ± 0.19 | 26.87 ± 0.00 | 95.00 ± 0.00 | 68.64 ± 0.03 | 81.17 ± 20.86 | 0.50/-1.10 |
| 50 diagonal (D) | 16 D | 92.76 ± 3.53 | 84.67 ± 1.15 | 99.00 ± 0.00 | 86.17 ± 5.81 | 79.83 ± 6.08 | 73.55 ± 1.39 | 51.72 ± 3.78 | 23.75 ± 2.66 | 83.90 ± 6.43 | 59.05 ± 0.94 | 73.44 ± 22.08 | 1.00 / 0.98 |
| | 32 D | 95.33 ± 2.08 | 87.33 ± 2.08 | 99.00 ± 0.00 | 92.60 ± 0.29 | 90.35 ± 1.00 | 75.43 ± 1.33 | 63.84 ± 1.64 | 26.97 ± 3.21 | 93.33 ± 1.15 | 61.94 ± 0.32 | 78.61 ± 21.60 | 1.00 / 0.96 |
| | 64 D | 97.00 ± 0.00 | 90.33 ± 1.53 | 99.00 ± 0.00 | 93.78 ± 0.19 | 93.00 ± 0.58 | 76.27 ± 0.49 | 74.39 ± 0.90 | 29.28 ± 0.81 | 95.67 ± 0.58 | 64.84 ± 0.27 | 81.36 ± 20.83 | 1.00 / 0.92 |
| | 128 D | 96.33 ± 0.58 | 92.67 ± 0.58 | 99.00 ± 0.00 | 93.56 ± 0.19 | 93.00 ± 0.58 | 77.24 ± 0.19 | 73.76 ± 1.25 | 29.58 ± 0.93 | 95.00 ± 0.00 | 69.04 ± 0.54 | 81.92 ± 20.44 | 1.00 / 0.84 |
| | 256 D | 95.67 ± 0.58 | 88.33 ± 0.58 | 99.00 ± 0.00 | 93.56 ± 0.19 | 94.67 ± 0.67 | 78.45 ± 0.14 | 72.86 ± 2.07 | 27.00 ± 0.77 | 95.33 ± 0.58 | 69.61 ± 0.18 | 81.45 ± 21.00 | 1.00 / 0.68 |
| 50 full (F) | 16 F | 94.06 ± 3.54 | 85.67 ± 1.15 | 98.67 ± 0.58 | 90.35 ± 1.37 | 89.97 ± 1.78 | 74.46 ± 0.58 | 49.03 ± 7.07 | 27.14 ± 3.94 | 92.33 ± 1.53 | 60.26 ± 1.03 | 76.19 ± 22.80 | 1.00/0.98 |
| | 32 F | 97.00 ± 0.00 | 86.67 ± 1.53 | 99.00 ± 0.00 | 93.67 ± 0.00 | 92.22 ± 0.69 | 75.86 ± 0.22 | 71.68 ± 0.65 | 29.26 ± 0.70 | 95.67 ± 1.53 | 63.88 ± 0.10 | 80.39 ± 20.81 | 0.99/0.95 |
| | 64 F | 96.67 ± 0.58 | 91.00 ± 2.00 | 99.00 ± 0.00 | 93.56 ± 0.19 | 93.22 ± 0.51 | 77.17 ± 0.38 | 77.11 ± 0.51 | 29.75 ± 0.03 | 95.33 ± 0.58 | 68.13 ± 0.75 | 82.09 ± 20.33 | 0.97/0.89 |
| | 128 F | 97.00 ± 0.00 | 91.00 ± 1.00 | 99.00 ± 0.00 | 93.33 ± 0.00 | 94.11 ± 0.51 | 77.23 ± 0.17 | 73.67 ± 0.58 | 27.62 ± 1.12 | 95.00 ± 1.00 | 69.40 ± 0.16 | 81.74 ± 20.97 | 0.88/0.72 |
| | 256 F | 95.00 ± 0.00 | 86.00 ± 0.00 | 99.00 ± 0.00 | 93.67 ± 0.00 | 94.44 ± 0.19 | 77.97 ± 0.24 | 75.47 ± 0.58 | 26.96 ± 0.09 | 95.00 ± 0.00 | 69.28 ± 0.05 | 81.58 ± 20.92 | 0.50/0.18 |
| 100 diagonal (D) | 16 D | 76.43 ± 7.07 | 76.67 ± 4.93 | 91.61 ± 2.75 | 89.99 ± 1.07 | 47.89 ± 8.62 | 63.17 ± 1.31 | 22.23 ± 5.27 | 14.46 ± 2.89 | 73.93 ± 3.13 | 57.17 ± 1.05 | 61.35 ± 25.78 | 1.00 / 0.99 |
| | 32 D | 90.10 ± 5.85 | 84.00 ± 1.00 | 99.00 ± 0.00 | 85.52 ± 5.34 | 75.88 ± 12.57 | 71.15 ± 3.61 | 46.10 ± 7.39 | 21.04 ± 6.76 | 86.67 ± 1.86 | 58.64 ± 1.02 | 71.81 ± 23.39 | 1.00 / 0.98 |
| | 64 D | 95.56 ± 2.49 | 86.67 ± 0.58 | 99.00 ± 0.00 | 92.24 ± 1.68 | 90.89 ± 1.17 | 74.57 ± 0.50 | 67.07 ± 3.81 | 29.78 ± 1.92 | 91.67 ± 2.31 | 60.28 ± 1.51 | 78.77 ± 20.77 | 1.00 / 0.96 |
| | 128 D | 96.00 ± 0.00 | 87.33 ± 1.15 | 99.00 ± 0.00 | 93.89 ± 0.19 | 93.00 ± 0.58 | 76.68 ± 0.18 | 74.84 ± 2.23 | 29.79 ± 0.50 | 93.67 ± 0.58 | 63.49 ± 0.34 | 80.77 ± 20.47 | 1.00 / 0.92 |
| | 256 D | 95.00 ± 0.00 | 91.00 ± 0.00 | 99.00 ± 0.00 | 93.56 ± 0.19 | 93.11 ± 0.23 | 76.93 ± 0.23 | 75.13 ± 0.84 | 29.75 ± 0.73 | 95.33 ± 0.58 | 67.89 ± 1.34 | 81.67 ± 20.28 | 1.00 / 0.84 |
| 100 full (F) | 16 F | 90.70 ± 1.07 | 83.00 ± 2.65 | 96.00 ± 3.00 | 91.22 ± 2.94 | 87.94 ± 0.54 | 73.07 ± 0.93 | 49.41 ± 2.04 | 24.17 ± 4.22 | 82.33 ± 2.08 | 58.18 ± 0.44 | 73.60 ± 22.23 | 1.00/0.99 |
| | 32 F | 95.33 ± 1.53 | 85.00 ± 1.00 | 99.00 ± 0.00 | 93.50 ± 0.22 | 91.44 ± 0.84 | 75.00 ± 0.19 | 65.09 ± 2.23 | 30.20 ± 0.81 | 81.67 ± 1.53 | 60.92 ± 0.26 | 78.72 ± 20.72 | 0.99/0.97 |
| | 64 F | 97.00 ± 0.00 | 85.67 ± 1.53 | 99.00 ± 0.00 | 93.78 ± 0.19 | 92.56 ± 0.19 | 76.01 ± 0.13 | 73.96 ± 0.89 | 29.46 ± 0.21 | 94.33 ± 1.53 | 64.07 ± 0.37 | 80.58 ± 20.59 | 0.97/0.93 |
| | 128 F | 96.33 ± 0.58 | 88.67 ± 0.58 | 99.00 ± 0.00 | 93.00 ± 0.00 | 93.89 ± 0.19 | 77.04 ± 0.30 | 77.33 ± 1.01 | 29.49 ± 0.35 | 96.00 ± 0.00 | 68.76 ± 0.25 | 82.12 ± 20.35 | 0.88/0.80 |
| | 256 F | 96.33 ± 0.58 | 88.67 ± 0.58 | 99.00 ± 0.00 | 93.67 ± 0.00 | 94.89 ± 0.19 | 78.16 ± 0.18 | 73.40 ± 0.12 | 26.86 ± 0.68 | 96.00 ± 0.00 | 69.47 ± 0.23 | 81.64 ± 21.15 | 0.50/0.34 |
| 100 w/clusters (C) | 16 C 5 | 98.33 ± 0.47 | 89.00 ± 0.82 | 99.00 ± 0.00 | 93.25 ± 0.40 | 92.89 ± 0.87 | 76.33 ± 0.28 | 78.24 ± 2.26 | 28.45 ± 0.40 | 96.67 ± 0.47 | 75.93 ± 8.31 | 82.81 ± 20.02 | 1.00/0.95 |
| | 16 C 7 | 97.67 ± 0.47 | 87.00 ± 0.82 | 99.00 ± 0.00 | 93.46 ± 0.29 | 93.11 ± 0.68 | 76.55 ± 0.29 | 79.03 ± 2.05 | 28.62 ± 1.29 | 95.33 ± 0.47 | 76.55 ± 6.91 | 82.63 ± 19.76 | 1.00/0.93 |
| 500 diagonal (D) | 16 D | 54.44 ± 6.87 | 47.00 ± 2.83 | 82.21 ± 3.59 | 73.38 ± 14.97 | 80.13 ± 3.68 | 57.42 ± 5.29 | 18.33 ± 1.33 | 12.19 ± 0.30 | 72.67 ± 6.03 | 55.79 ± 0.20 | 55.64 ± 24.25 | 1.00 / 1.00 |
| | 32 D | 58.08 ± 11.52 | 47.00 ± 7.07 | 82.06 ± 1.69 | 78.62 ± 11.23 | 85.57 ± 1.48 | 59.19 ± 3.70 | 22.76 ± 3.95 | 13.15 ± 1.94 | 75.33 ± 4.04 | 56.07 ± 0.52 | 58.16 ± 24.56 | 1.00 / 1.00 |
| | 64 D | 69.21 ± 2.03 | 54.50 ± 9.19 | 88.33 ± 4.04 | 91.11 ± 0.38 | 88.78 ± 0.38 | 67.71 ± 2.59 | 34.79 ± 6.86 | 13.80 ± 1.95 | 77.67 ± 2.31 | 56.78 ± 0.73 | 64.61 ± 24.79 | 1.00 / 0.99 |
| | 128 D | 79.77 ± 0.37 | 79.50 ± 2.12 | 95.89 ± 2.83 | 91.89 ± 1.39 | 88.67 ± 0.00 | 70.27 ± 1.73 | 46.64 ± 10.58 | 14.63 ± 0.25 | 81.00 ± 5.00 | 56.88 ± 0.55 | 70.20 ± 24.63 | 1.00 / 0.98 |
| | 256 D | 93.83 ± 2.52 | 85.00 ± 0.00 | 99.00 ± 0.00 | 93.78 ± 0.19 | 90.56 ± 0.38 | 73.25 ± 1.86 | 51.14 ± 3.86 | 22.93 ± 3.86 | 87.33 ± 2.31 | 58.48 ± 0.20 | 75.20 ± 23.90 | 1.00 / 0.97 |
| 500 full (F) | 16 F | 54.30 ± 1.13 | 37.00 ± 5.66 | 77.67 ± 0.58 | 91.00 ± 0.00 | 90.56 ± 0.19 | 67.63 ± 0.45 | 48.81 ± 0.35 | 14.70 ± 0.65 | 79.00 ± 1.00 | 57.66 ± 0.19 | 62.69 ± 23.46 | 1.00/1.00 |
| | 32 F | 75.10 ± 4.92 | 46.50 ± 3.54 | 91.67 ± 1.53 | 91.56 ± 0.19 | 91.56 ± 0.38 | 72.03 ± 0.15 | 52.53 ± 0.86 | 13.93 ± 0.02 | 81.67 ± 1.53 | 58.50 ± 0.20 | 68.24 ± 24.29 | 0.99/0.99 |
| | 64 F | 96.94 ± 0.42 | 82.50 ± 0.71 | 93.33 ± 0.58 | 93.89 ± 0.69 | 90.67 ± 0.00 | 75.99 ± 0.64 | 55.63 ± 1.07 | 14.74 ± 0.27 | 86.33 ± 0.58 | 59.43 ± 0.05 | 74.69 ± 25.01 | 0.97/0.96 |
| | 128 F | 97.67 ± 0.58 | 83.50 ± 2.12 | 98.00 ± 0.00 | 93.56 ± 0.19 | 92.00 ± 0.00 | 75.80 ± 0.16 | 66.19 ± 0.81 | 28.67 ± 0.49 | 93.00 ± 0.00 | 61.53 ± 0.13 | 78.84 ± 21.50 | 0.88/0.86 |
| | 256 F | 95.00 ± 0.00 | 88.50 ± 0.71 | 99.00 ± 0.00 | 93.78 ± 0.19 | 93.00 ± 0.88 | 76.33 ± 0.29 | 74.60 ± 1.21 | 27.82 ± 0.42 | 95.33 ± 0.58 | 63.70 ± 0.14 | 80.75 ± 21.60 | 0.50/0.47 |
| 500 w/clusters (C) | 16 C 7 | 95.00 | 86.00 | 98.00 | 93.67 | 91.67 | 75.10 | 55.52 | 20.50 | 90.00 | 63.57 | 76.90 | 1.00/0.98 |
| | 16 C 10 | 96.00 | 87.00 | 99.00 | 93.00 | 91.33 | 74.17 | 55.14 | 30.29 | 94.00 | 63.09 | 78.30 | 1.00/0.98 |
| | 16 C 25 | 96.00 | 86.00 | 99.00 | 92.71 | 93.00 | 76.42 | 75.42 | 30.40 | 95.00 | 66.07 | 81.00 | 1.00/0.95 |
| | 64 C 5 | 95.00 | 84.00 | 99.00 | 93.67 | 92.67 | 76.45 | 76.43 | 27.49 | 96.00 | 64.10 | 80.48 | 0.97/0.93 |
| | 64 C 7 | 98.00 | 88.00 | 99.00 | 94.00 | 93.33 | 76.42 | 76.67 | 28.48 | 96.00 | 68.00 | 81.79 | 0.97/0.91 |
| 1000 w/clusters (C) | 16 C 25 | 95.00 | 84.00 | 99.00 | 93.67 | 90.67 | 76.43 | 56.71 | 30.20 | 97.00 | 64.61 | 78.73 | 1.00/0.97 |

Table 10: **Absolute** In-Distribution ROUGE-1 scores for various tasks and methods

| Model Type | Method Type | Tasks | | | | | | | | | | Average | Para. Saved |
|---|---|---|---|---|---|---|---|---|---|---|---|---|---|
| | | task039 | task190 | task280 | task290 | task391 | task442 | task620 | task1342 | task1391 | task1598 | | |
| | base | 0.00 ± 0.00 | 0.00 ± 0.00 | 0.02 ± 0.00 | 0.00 ± 0.00 | 0.00 ± 0.00 | 0.00 ± 0.00 | 0.00 ± 0.00 | 0.00 ± 0.00 | 0.00 ± 0.00 | 0.00 ± 0.00 | 0.00 ± 0.01 | 1.00 / 1.00 |
| | lora | 1.00 ± 0.00 | 1.00 ± 0.00 | 1.00 ± 0.00 | 1.00 ± 0.00 | 1.00 ± 0.00 | 1.00 ± 0.00 | 1.00 ± 0.00 | 1.00 ± 0.00 | 1.00 ± 0.00 | 1.00 ± 0.00 | 1.00 ± 0.00 | 0.00 / 0.00 |
| TIES | 10 | 0.69 ± 0.00 | 0.57 ± 0.02 | 0.45 ± 0.04 | 0.10 ± 0.01 | 0.57 ± 0.03 | 0.00 ± 0.00 | 0.39 ± 0.01 | 0.21 ± 0.00 | 0.82 ± 0.01 | 0.00 ± 0.00 | 0.38 ± 0.28 | 1.00 / 1.00 |
| | 50 | 0.45 ± 0.00 | 0.41 ± 0.00 | 0.18 ± 0.05 | 0.03 ± 0.01 | 0.70 ± 0.02 | 0.00 ± 0.00 | 0.36 ± 0.00 | 0.21 ± 0.00 | 0.32 ± 0.04 | 0.00 ± 0.00 | 0.27 ± 0.22 | 1.00 / 1.00 |
| | 100 | 0.41 ± 0.00 | 0.40 ± 0.01 | 0.20 ± 0.05 | 0.01 ± 0.02 | 0.65 ± 0.00 | 0.00 ± 0.00 | 0.36 ± 0.00 | 0.21 ± 0.00 | 0.01 ± 0.00 | 0.00 ± 0.00 | 0.23 ± 0.22 | 1.00 / 1.00 |
| | 500 | 0.22 ± 0.00 | 0.26 ± 0.00 | 0.01 ± 0.00 | 0.00 ± 0.00 | 0.60 ± 0.00 | 0.00 ± 0.00 | 0.32 ± 0.00 | 0.07 ± 0.00 | 0.01 ± 0.00 | 0.00 ± 0.00 | 0.15 ± 0.20 | 1.00 / 1.00 |
| SVD | SVD 2 | 0.98 ± 0.03 | 1.07 ± 0.02 | 1.00 ± 0.00 | 0.99 ± 0.01 | 0.98 ± 0.01 | 0.98 ± 0.03 | 0.94 ± 0.01 | 1.03 ± 0.17 | 1.00 ± 0.01 | 0.15 ± 0.29 | 0.91 ± 0.28 | 0.88 / 0.88 |
| | SVD 4 | 0.99 ± 0.04 | 1.04 ± 0.01 | 1.00 ± 0.00 | 1.00 ± 0.00 | 1.01 ± 0.02 | 1.11 ± 0.00 | 0.97 ± 0.02 | 0.99 ± 0.13 | 0.90 ± 0.17 | 1.00 ± 0.00 | 1.00 ± 0.08 | 0.75 / 0.75 |
| | SVD 8 | 1.00 ± 0.00 | 1.02 ± 0.01 | 1.00 ± 0.00 | 1.00 ± 0.00 | 1.00 ± 0.01 | 1.02 ± 0.05 | 1.00 ± 0.00 | 1.00 ± 0.00 | 1.01 ± 0.00 | 1.00 ± 0.00 | 1.00 ± 0.02 | 0.50 / 0.50 |
| | SVD 16 | 1.00 ± 0.00 | 1.00 ± 0.00 | 1.00 ± 0.00 | 1.00 ± 0.00 | 1.00 ± 0.00 | 1.00 ± 0.00 | 1.00 ± 0.00 | 0.99 ± 0.01 | 1.00 ± 0.00 | 1.00 ± 0.00 | 1.00 ± 0.00 | 0.00 / 0.00 |
| 10 diagonal (D) | 16 D | 1.02 ± 0.01 | 1.01 ± 0.01 | 1.00 ± 0.00 | 1.01 ± 0.02 | 0.96 ± 0.01 | 1.11 ± 0.11 | 0.89 ± 0.03 | 1.19 ± 0.04 | 0.99 ± 0.02 | 0.33 ± 0.58 | 0.95 ± 0.27 | 1.00 / 0.90 |
| | 32 D | 1.01 ± 0.01 | 1.05 ± 0.01 | 1.00 ± 0.00 | 0.98 ± 0.01 | 1.02 ± 0.02 | 1.11 ± 0.00 | 0.93 ± 0.01 | 1.10 ± 0.04 | 1.00 ± 0.01 | 0.67 ± 0.58 | 0.98 ± 0.19 | 1.00 / 0.80 |
| | 64 D | 1.00 ± 0.00 | 1.03 ± 0.01 | 1.00 ± 0.00 | 1.00 ± 0.00 | 1.02 ± 0.02 | 1.11 ± 0.00 | 0.99 ± 0.01 | 1.00 ± 0.00 | 1.01 ± 0.00 | 0.67 ± 0.58 | 0.98 ± 0.19 | 1.00 / 0.60 |
| | 128 D | 1.00 ± 0.00 | 1.01 ± 0.01 | 1.00 ± 0.00 | 1.00 ± 0.00 | 1.00 ± 0.01 | 1.00 ± 0.00 | 1.00 ± 0.01 | 1.00 ± 0.00 | 1.00 ± 0.00 | 1.00 ± 0.00 | 1.00 ± 0.00 | 1.00 / 0.20 |
| | 256 D | 1.00 ± 0.00 | 1.00 ± 0.00 | 1.00 ± 0.00 | 1.00 ± 0.00 | 1.00 ± 0.00 | 1.00 ± 0.00 | 1.00 ± 0.00 | 1.00 ± 0.00 | 1.00 ± 0.00 | 1.00 ± 0.00 | 1.00 ± 0.00 | 1.00 / -0.60 |
| 10 full (F) | 16 F | 1.02 ± 0.01 | 1.06 ± 0.01 | 1.00 ± 0.00 | 1.00 ± 0.00 | 0.97 ± 0.03 | 1.15 ± 0.06 | 0.92 ± 0.02 | 1.17 ± 0.00 | 1.01 ± 0.01 | 0.67 ± 0.58 | 1.00 ± 0.20 | 1.00/0.90 |
| | 32 F | 1.02 ± 0.01 | 1.04 ± 0.01 | 1.00 ± 0.00 | 0.98 ± 0.01 | 1.00 ± 0.01 | 1.11 ± 0.00 | 0.92 ± 0.00 | 1.02 ± 0.04 | 1.00 ± 0.01 | 1.00 ± 0.00 | 1.01 ± 0.05 | 0.99/0.79 |
| | 64 F | 1.00 ± 0.00 | 1.03 ± 0.01 | 1.00 ± 0.00 | 1.00 ± 0.00 | 1.01 ± 0.01 | 1.07 ± 0.06 | 1.01 ± 0.01 | 1.00 ± 0.00 | 1.01 ± 0.00 | 1.00 ± 0.00 | 1.01 ± 0.03 | 0.97/0.57 |
| | 128 F | 1.00 ± 0.00 | 1.01 ± 0.01 | 1.00 ± 0.00 | 1.00 ± 0.00 | 1.00 ± 0.00 | 1.00 ± 0.00 | 1.00 ± 0.00 | 1.00 ± 0.00 | 1.00 ± 0.00 | 1.00 ± 0.00 | 1.00 ± 0.00 | 0.88/0.07 |
| | 256 F | 1.00 ± 0.00 | 1.00 ± 0.00 | 1.00 ± 0.00 | 1.00 ± 0.00 | 1.00 ± 0.00 | 1.00 ± 0.00 | 1.00 ± 0.00 | 1.00 ± 0.01 | 1.00 ± 0.00 | 1.00 ± 0.00 | 1.00 ± 0.00 | 0.50/-1.10 |
| 50 diagonal (D) | 16 D | 0.91 ± 0.06 | 0.98 ± 0.01 | 1.00 ± 0.00 | 0.91 ± 0.09 | 0.78 ± 0.05 | 0.89 ± 0.29 | 0.34 ± 0.06 | 0.50 ± 0.45 | 0.86 ± 0.07 | 0.00 ± 0.00 | 0.72 ± 0.35 | 1.00 / 0.98 |
| | 32 D | 1.00 ± 0.02 | 1.02 ± 0.02 | 1.00 ± 0.00 | 1.00 ± 0.01 | 0.90 ± 0.03 | 0.85 ± 0.42 | 0.56 ± 0.04 | 0.98 ± 0.23 | 0.98 ± 0.01 | 0.00 ± 0.00 | 0.83 ± 0.34 | 1.00 / 0.96 |
| | 64 D | 1.02 ± 0.00 | 1.05 ± 0.02 | 1.00 ± 0.00 | 1.00 ± 0.01 | 0.95 ± 0.02 | 1.15 ± 0.17 | 0.81 ± 0.03 | 1.14 ± 0.00 | 1.01 ± 0.01 | 0.00 ± 0.00 | 0.91 ± 0.33 | 1.00 / 0.92 |
| | 128 D | 1.01 ± 0.01 | 1.08 ± 0.03 | 1.00 ± 0.00 | 1.00 ± 0.01 | 0.95 ± 0.02 | 1.04 ± 0.06 | 0.92 ± 0.03 | 1.21 ± 0.07 | 1.00 ± 0.00 | 0.67 ± 0.58 | 0.99 ± 0.20 | 1.00 / 0.84 |
| | 256 D | 1.01 ± 0.01 | 1.03 ± 0.01 | 1.00 ± 0.00 | 1.00 ± 0.01 | 1.01 ± 0.02 | 1.11 ± 0.00 | 0.95 ± 0.04 | 1.02 ± 0.04 | 1.00 ± 0.01 | 1.00 ± 0.00 | 1.01 ± 0.04 | 1.00 / 0.68 |
| 50 full (F) | 16 F | 0.96 ± 0.05 | 1.00 ± 0.01 | 1.00 ± 0.01 | 0.95 ± 0.04 | 0.87 ± 0.01 | 1.04 ± 0.06 | 0.31 ± 0.08 | 0.98 ± 0.23 | 0.97 ± 0.02 | 0.00 ± 0.00 | 0.81 ± 0.35 | 1.00/0.98 |
| | 32 F | 1.02 ± 0.00 | 1.00 ± 0.02 | 1.00 ± 0.00 | 1.00 ± 0.00 | 0.92 ± 0.03 | 1.15 ± 0.06 | 0.73 ± 0.04 | 1.17 ± 0.04 | 1.01 ± 0.02 | 0.00 ± 0.00 | 0.90 ± 0.33 | 0.99/0.95 |
| | 64 F | 1.02 ± 0.01 | 1.06 ± 0.03 | 1.00 ± 0.00 | 1.00 ± 0.01 | 0.96 ± 0.02 | 1.22 ± 0.00 | 0.94 ± 0.02 | 1.17 ± 0.04 | 1.00 ± 0.01 | 0.00 ± 0.00 | 0.94 ± 0.33 | 0.97/0.89 |
| | 128 F | 1.02 ± 0.00 | 1.06 ± 0.01 | 1.00 ± 0.00 | 0.99 ± 0.00 | 0.99 ± 0.02 | 1.15 ± 0.06 | 0.92 ± 0.01 | 1.10 ± 0.08 | 1.00 ± 0.01 | 1.00 ± 0.00 | 1.02 ± 0.07 | 0.88/0.72 |
| | 256 F | 1.00 ± 0.00 | 1.02 ± 0.00 | 1.00 ± 0.00 | 1.00 ± 0.00 | 1.00 ± 0.01 | 1.04 ± 0.06 | 0.99 ± 0.00 | 1.00 ± 0.00 | 1.01 ± 0.00 | 1.00 ± 0.00 | 1.01 ± 0.02 | 0.50/0.18 |
| 100 diagonal (D) | 16 D | 0.54 ± 0.16 | 0.89 ± 0.06 | 0.90 ± 0.04 | 0.89 ± 0.05 | 0.42 ± 0.08 | 0.44 ± 0.05 | 0.08 ± 0.02 | 0.00 ± 0.00 | 0.76 ± 0.05 | 0.00 ± 0.00 | 0.49 ± 0.35 | 1.00 / 0.99 |
| | 32 D | 0.85 ± 0.15 | 0.98 ± 0.01 | 1.00 ± 0.00 | 0.86 ± 0.13 | 0.70 ± 0.14 | 0.74 ± 0.28 | 0.28 ± 0.07 | 0.48 ± 0.55 | 0.91 ± 0.02 | 0.00 ± 0.00 | 0.68 ± 0.36 | 1.00 / 0.98 |
| | 64 D | 1.00 ± 0.04 | 1.01 ± 0.01 | 1.00 ± 0.00 | 0.98 ± 0.02 | 0.88 ± 0.04 | 1.07 ± 0.06 | 0.58 ± 0.09 | 1.10 ± 0.04 | 0.96 ± 0.02 | 0.00 ± 0.00 | 0.86 ± 0.32 | 1.00 / 0.96 |
| | 128 D | 1.01 ± 0.00 | 1.02 ± 0.01 | 1.00 ± 0.00 | 1.01 ± 0.01 | 0.95 ± 0.02 | 1.11 ± 0.00 | 0.81 ± 0.06 | 1.14 ± 0.00 | 0.99 ± 0.01 | 0.00 ± 0.00 | 0.91 ± 0.33 | 1.00 / 0.92 |
| | 256 D | 1.00 ± 0.00 | 1.06 ± 0.03 | 1.00 ± 0.00 | 1.00 ± 0.00 | 0.96 ± 0.01 | 1.11 ± 0.11 | 0.92 ± 0.02 | 1.21 ± 0.07 | 1.00 ± 0.01 | 0.00 ± 0.00 | 0.93 ± 0.33 | 1.00 / 0.84 |
| 100 full (F) | 16 F | 0.85 ± 0.03 | 0.97 ± 0.03 | 0.97 ± 0.03 | 0.95 ± 0.06 | 0.80 ± 0.02 | 0.81 ± 0.17 | 0.29 ± 0.04 | 0.60 ± 0.34 | 0.87 ± 0.02 | 0.00 ± 0.00 | 0.71 ± 0.33 | 1.00/0.99 |
| | 32 F | 0.99 ± 0.02 | 0.99 ± 0.01 | 1.00 ± 0.00 | 1.00 ± 0.01 | 0.90 ± 0.03 | 1.04 ± 0.06 | 0.55 ± 0.04 | 1.07 ± 0.07 | 0.96 ± 0.01 | 0.00 ± 0.00 | 0.85 ± 0.32 | 0.99/0.97 |
| | 64 F | 1.02 ± 0.00 | 1.00 ± 0.02 | 1.00 ± 0.00 | 1.00 ± 0.01 | 0.94 ± 0.01 | 1.04 ± 0.06 | 0.78 ± 0.01 | 1.14 ± 0.00 | 0.99 ± 0.02 | 0.00 ± 0.00 | 0.89 ± 0.31 | 0.97/0.93 |
| | 128 F | 1.01 ± 0.01 | 1.05 ± 0.01 | 1.00 ± 0.00 | 0.98 ± 0.00 | 0.98 ± 0.01 | 1.15 ± 0.06 | 0.94 ± 0.01 | 1.21 ± 0.00 | 1.01 ± 0.00 | 0.33 ± 0.58 | 0.97 ± 0.28 | 0.88/0.80 |
| | 256 F | 1.01 ± 0.01 | 1.03 ± 0.01 | 1.00 ± 0.00 | 1.00 ± 0.00 | 1.02 ± 0.01 | 1.19 ± 0.06 | 0.93 ± 0.01 | 1.02 ± 0.04 | 1.01 ± 0.00 | 1.00 ± 0.00 | 1.02 ± 0.06 | 0.50/0.34 |
| 100 w/clusters (C) | 16 C 5 | 1.13 ± 0.01 | 1.03 ± 0.01 | 1.00 ± 0.00 | 1.00 ± 0.01 | 0.97 ± 0.03 | 1.24 ± 0.09 | 0.88 ± 0.06 | 1.42 ± 0.07 | 1.05 ± 0.01 | 0.65 ± 0.46 | 1.04 ± 0.19 | 1.00/0.95 |
| | 16 C 7 | 1.12 ± 0.01 | 1.01 ± 0.01 | 1.00 ± 0.00 | 1.00 ± 0.00 | 0.98 ± 0.03 | 1.16 ± 0.11 | 0.92 ± 0.05 | 1.45 ± 0.04 | 1.03 ± 0.01 | 0.69 ± 0.49 | 1.04 ± 0.18 | 1.00/0.93 |
| 500 diagonal (D) | 16 D | 0.22 ± 0.10 | 0.55 ± 0.03 | 0.81 ± 0.05 | 0.33 ± 0.49 | 0.70 ± 0.03 | 0.15 ± 0.17 | 0.03 ± 0.01 | 0.00 ± 0.00 | 0.76 ± 0.06 | 0.00 ± 0.00 | 0.35 ± 0.35 | 1.00 / 1.00 |
| | 32 D | 0.27 ± 0.18 | 0.55 ± 0.08 | 0.82 ± 0.02 | 0.49 ± 0.37 | 0.75 ± 0.01 | 0.22 ± 0.11 | 0.05 ± 0.05 | 0.02 ± 0.04 | 0.79 ± 0.04 | 0.00 ± 0.00 | 0.39 ± 0.34 | 1.00 / 1.00 |
| | 64 D | 0.40 ± 0.04 | 0.63 ± 0.11 | 0.89 ± 0.04 | 0.91 ± 0.01 | 0.80 ± 0.01 | 0.48 ± 0.06 | 0.13 ± 0.04 | 0.05 ± 0.08 | 0.82 ± 0.02 | 0.00 ± 0.00 | 0.51 ± 0.35 | 1.00 / 0.99 |
| | 128 D | 0.61 ± 0.04 | 0.92 ± 0.02 | 0.97 ± 0.03 | 0.93 ± 0.05 | 0.80 ± 0.00 | 0.74 ± 0.17 | 0.22 ± 0.11 | 0.12 ± 0.08 | 0.85 ± 0.00 | 0.00 ± 0.00 | 0.61 ± 0.36 | 1.00 / 0.98 |
| | 256 D | 0.95 ± 0.02 | 0.99 ± 0.01 | 1.00 ± 0.00 | 1.00 ± 0.01 | 0.86 ± 0.01 | 0.85 ± 0.28 | 0.28 ± 0.07 | 0.55 ± 0.39 | 0.92 ± 0.02 | 0.00 ± 0.00 | 0.73 ± 0.36 | 1.00 / 0.97 |
| 500 full (F) | 16 F | 0.21 ± 0.02 | 0.43 ± 0.07 | 0.78 ± 0.01 | 0.90 ± 0.00 | 0.86 ± 0.01 | 0.59 ± 0.06 | 0.21 ± 0.01 | 0.12 ± 0.04 | 0.83 ± 0.01 | 0.00 ± 0.00 | 0.50 ± 0.34 | 1.00/1.00 |
| | 32 F | 0.54 ± 0.08 | 0.54 ± 0.04 | 0.93 ± 0.02 | 0.92 ± 0.01 | 0.90 ± 0.01 | 0.63 ± 0.13 | 0.26 ± 0.02 | 0.14 ± 0.00 | 0.86 ± 0.02 | 0.00 ± 0.00 | 0.57 ± 0.34 | 0.99/0.99 |
| | 64 F | 0.99 ± 0.03 | 0.96 ± 0.01 | 0.94 ± 0.01 | 1.01 ± 0.03 | 0.87 ± 0.00 | 1.04 ± 0.17 | 0.36 ± 0.00 | 0.14 ± 0.00 | 0.91 ± 0.01 | 0.00 ± 0.00 | 0.71 ± 0.39 | 0.97/0.96 |
| | 128 F | 1.02 ± 0.01 | 0.97 ± 0.02 | 0.99 ± 0.00 | 1.00 ± 0.01 | 0.92 ± 0.00 | 1.15 ± 0.06 | 0.61 ± 0.00 | 1.07 ± 0.00 | 0.98 ± 0.00 | 0.00 ± 0.00 | 0.87 ± 0.33 | 0.88/0.86 |
| | 256 F | 1.03 ± 0.00 | 1.03 ± 0.01 | 1.00 ± 0.00 | 1.00 ± 0.01 | 0.95 ± 0.03 | 1.00 ± 0.00 | 0.78 ± 0.01 | 1.07 ± 0.00 | 1.00 ± 0.01 | 0.00 ± 0.00 | 0.88 ± 0.31 | 0.50/0.47 |
| 500 w/clusters (C) | 16 C 7 | 1.08 | 1.00 | 0.99 | 1.00 | 0.92 | 1.01 | 0.39 | 0.62 | 0.98 | 0.00 | 0.80 | 1.00/0.98 |
| | 16 C 10 | 1.10 | 1.01 | 1.00 | 0.98 | 0.91 | 1.01 | 0.37 | 1.51 | 1.02 | 0.00 | 0.89 | 1.00/0.98 |
| | 16 C 25 | 1.10 | 1.00 | 1.00 | 0.99 | 0.97 | 1.12 | 0.81 | 1.42 | 1.03 | 0.00 | 0.95 | 1.00/0.95 |
| | 64 C 5 | 1.09 | 0.98 | 1.00 | 1.00 | 0.96 | 1.12 | 0.83 | 1.33 | 1.04 | 0.00 | 0.94 | 0.97/0.93 |
| | 64 C 7 | 1.13 | 1.02 | 1.00 | 1.01 | 0.98 | 1.12 | 0.90 | 1.42 | 1.04 | 0.00 | 0.96 | 0.97/0.91 |
| 1000 w/clusters (C) | 16 C 25 | 1.09 | 0.98 | 1.00 | 1.00 | 0.89 | 1.01 | 0.39 | 1.42 | 1.05 | 0.00 | 0.88 | 1.00/0.97 |

Table 11: **Relative** In-Distribution exact match scores for various tasks and methods

| Model Type | Method Type | Tasks | | | | | | | | | | Average | Para. Saved |
|---|---|---|---|---|---|---|---|---|---|---|---|---|---|
| | | task039 | task190 | task280 | task290 | task391 | task442 | task620 | task1342 | task1391 | task1598 | | |
| | base | 8.59 ± 0.08 | 9.15 ± 0.00 | 2.55 ± 0.00 | 2.88 ± 0.00 | 2.34 ± 0.00 | 3.46 ± 0.04 | 6.40 ± 0.18 | 5.55 ± 0.00 | 8.60 ± 0.00 | 2.67 ± 0.00 | 5.19 ± 2.65 | 1.00 / 1.00 |
| | lora | 0.36 ± 0.01 | 0.17 ± 0.00 | 0.01 ± 0.00 | 0.12 ± 0.00 | 0.11 ± 0.00 | 0.76 ± 0.02 | 1.17 ± 0.07 | 1.94 ± 0.00 | 0.16 ± 0.00 | 0.85 ± 0.00 | 0.57 ± 0.59 | 0.00 / 0.00 |
| SVD | SVD 2 | 0.32 ± 0.01 | 0.15 ± 0.00 | 0.01 ± 0.00 | 0.12 ± 0.00 | 0.10 ± 0.00 | 0.76 ± 0.02 | 1.13 ± 0.08 | 1.94 ± 0.00 | 0.13 ± 0.00 | 0.97 ± 0.00 | 0.57 ± 0.60 | 0.88 / 0.88 |
| | SVD 4 | 0.33 ± 0.01 | 0.16 ± 0.00 | 0.01 ± 0.00 | 0.12 ± 0.00 | 0.11 ± 0.00 | 0.76 ± 0.02 | 1.14 ± 0.08 | 1.94 ± 0.00 | 0.14 ± 0.00 | 0.86 ± 0.00 | 0.56 ± 0.59 | 0.75 / 0.75 |
| | SVD 8 | 0.35 ± 0.01 | 0.17 ± 0.00 | 0.01 ± 0.00 | 0.12 ± 0.00 | 0.11 ± 0.00 | 0.77 ± 0.02 | 1.16 ± 0.07 | 1.94 ± 0.00 | 0.15 ± 0.00 | 0.84 ± 0.00 | 0.51 ± 0.59 | 0.50 / 0.50 |
| | SVD 16 | 0.36 ± 0.01 | 0.17 ± 0.00 | 0.01 ± 0.00 | 0.12 ± 0.00 | 0.11 ± 0.00 | 0.76 ± 0.02 | 1.14 ± 0.06 | 1.94 ± 0.00 | 0.16 ± 0.00 | 0.85 ± 0.00 | 0.56 ± 0.59 | 0.00 / 0.00 |
| 10 diagonal (D) | 16 D | 0.33 ± 0.01 | 0.15 ± 0.01 | 0.01 ± 0.00 | 0.12 ± 0.00 | 0.10 ± 0.00 | 0.76 ± 0.03 | 1.13 ± 0.08 | 1.95 ± 0.01 | 0.14 ± 0.00 | 1.00 ± 0.02 | 0.57 ± 0.61 | 1.00 / 0.90 |
| | 32 D | 0.33 ± 0.01 | 0.16 ± 0.00 | 0.01 ± 0.00 | 0.12 ± 0.00 | 0.10 ± 0.00 | 0.75 ± 0.02 | 1.11 ± 0.07 | 1.93 ± 0.00 | 0.14 ± 0.01 | 0.88 ± 0.00 | 0.55 ± 0.60 | 1.00 / 0.80 |
| | 64 D | 0.35 ± 0.01 | 0.17 ± 0.00 | 0.01 ± 0.00 | 0.12 ± 0.00 | 0.11 ± 0.00 | 0.75 ± 0.02 | 1.11 ± 0.07 | 1.94 ± 0.00 | 0.15 ± 0.00 | 0.84 ± 0.00 | 0.55 ± 0.59 | 1.00 / 0.60 |
| | 128 D | 0.35 ± 0.01 | 0.17 ± 0.00 | 0.01 ± 0.00 | 0.12 ± 0.00 | 0.11 ± 0.00 | 0.75 ± 0.02 | 1.11 ± 0.07 | 1.94 ± 0.00 | 0.16 ± 0.00 | 0.84 ± 0.00 | 0.56 ± 0.59 | 1.00 / 0.20 |
| | 256 D | 0.36 ± 0.01 | 0.17 ± 0.00 | 0.01 ± 0.00 | 0.12 ± 0.00 | 0.11 ± 0.00 | 0.75 ± 0.02 | 1.12 ± 0.07 | 1.94 ± 0.00 | 0.16 ± 0.00 | 0.85 ± 0.00 | 0.56 ± 0.59 | 1.00 / -0.60 |
| 10 full (F) | 16 F | 0.33 ± 0.00 | 0.15 ± 0.00 | 0.01 ± 0.00 | 0.12 ± 0.00 | 0.10 ± 0.00 | 0.76 ± 0.02 | 1.20 ± 0.02 | 1.95 ± 0.00 | 0.13 ± 0.00 | 0.97 ± 0.00 | 0.57 ± 0.61 | 1.00/0.90 |
| | 32 F | 0.33 ± 0.01 | 0.16 ± 0.00 | 0.01 ± 0.00 | 0.12 ± 0.00 | 0.10 ± 0.00 | 0.75 ± 0.00 | 1.11 ± 0.07 | 1.94 ± 0.00 | 0.14 ± 0.00 | 0.86 ± 0.00 | 0.55 ± 0.60 | 0.99/0.79 |
| | 64 F | 0.34 ± 0.01 | 0.16 ± 0.00 | 0.01 ± 0.00 | 0.12 ± 0.00 | 0.11 ± 0.00 | 0.75 ± 0.02 | 1.11 ± 0.07 | 1.94 ± 0.00 | 0.15 ± 0.00 | 0.84 ± 0.00 | 0.55 ± 0.59 | 0.97/0.57 |
| | 128 F | 0.35 ± 0.01 | 0.17 ± 0.00 | 0.01 ± 0.00 | 0.12 ± 0.00 | 0.11 ± 0.00 | 0.75 ± 0.02 | 1.12 ± 0.07 | 1.94 ± 0.00 | 0.16 ± 0.00 | 0.84 ± 0.00 | 0.56 ± 0.59 | 0.88/0.07 |
| | 256 F | 0.36 ± 0.01 | 0.17 ± 0.00 | 0.01 ± 0.00 | 0.12 ± 0.00 | 0.11 ± 0.00 | 0.75 ± 0.02 | 1.12 ± 0.07 | 1.94 ± 0.00 | 0.16 ± 0.00 | 0.85 ± 0.00 | 0.56 ± 0.59 | 0.50/-1.10 |
| 50 diagonal (D) | 16 D | 0.61 ± 0.06 | 0.19 ± 0.02 | 0.03 ± 0.01 | 0.29 ± 0.04 | 0.36 ± 0.04 | 0.95 ± 0.05 | 1.73 ± 0.21 | 2.66 ± 0.22 | 0.32 ± 0.11 | 1.98 ± 0.01 | 0.91 ± 0.88 | 1.00 / 0.98 |
| | 32 D | 0.37 ± 0.02 | 0.16 ± 0.00 | 0.01 ± 0.00 | 0.19 ± 0.03 | 0.18 ± 0.01 | 0.85 ± 0.05 | 1.37 ± 0.14 | 2.12 ± 0.05 | 0.16 ± 0.00 | 1.65 ± 0.03 | 0.71 ± 0.73 | 1.00 / 0.96 |
| | 64 D | 0.33 ± 0.02 | 0.15 ± 0.00 | 0.01 ± 0.00 | 0.12 ± 0.00 | 0.10 ± 0.00 | 0.79 ± 0.02 | 1.12 ± 0.08 | 1.97 ± 0.01 | 0.13 ± 0.01 | 1.13 ± 0.03 | 0.59 ± 0.63 | 1.00 / 0.92 |
| | 128 D | 0.33 ± 0.01 | 0.15 ± 0.00 | 0.01 ± 0.00 | 0.12 ± 0.00 | 0.10 ± 0.00 | 0.76 ± 0.03 | 1.10 ± 0.05 | 1.93 ± 0.01 | 0.14 ± 0.00 | 0.93 ± 0.01 | 0.56 ± 0.60 | 1.00 / 0.84 |
| | 256 D | 0.34 ± 0.01 | 0.16 ± 0.00 | 0.01 ± 0.00 | 0.12 ± 0.00 | 0.10 ± 0.00 | 0.76 ± 0.03 | 1.11 ± 0.05 | 1.93 ± 0.00 | 0.15 ± 0.00 | 0.85 ± 0.00 | 0.55 ± 0.59 | 1.00 / 0.68 |
| 50 full (F) | 16 F | 0.47 ± 0.06 | 0.17 ± 0.00 | 0.02 ± 0.00 | 0.20 ± 0.02 | 0.19 ± 0.04 | 0.86 ± 0.03 | 1.71 ± 0.10 | 2.20 ± 0.04 | 0.17 ± 0.01 | 1.84 ± 0.07 | 0.78 ± 0.80 | 1.00/0.98 |
| | 32 F | 0.36 ± 0.02 | 0.16 ± 0.00 | 0.01 ± 0.00 | 0.14 ± 0.00 | 0.11 ± 0.00 | 0.80 ± 0.03 | 1.14 ± 0.08 | 2.00 ± 0.01 | 0.14 ± 0.00 | 1.32 ± 0.02 | 0.62 ± 0.65 | 0.99/0.95 |
| | 64 F | 0.33 ± 0.01 | 0.15 ± 0.00 | 0.01 ± 0.00 | 0.12 ± 0.00 | 0.10 ± 0.00 | 0.77 ± 0.03 | 1.10 ± 0.06 | 1.94 ± 0.00 | 0.13 ± 0.00 | 1.02 ± 0.00 | 0.57 ± 0.61 | 0.97/0.89 |
| | 128 F | 0.33 ± 0.01 | 0.16 ± 0.00 | 0.00 ± 0.00 | 0.12 ± 0.00 | 0.10 ± 0.00 | 0.76 ± 0.03 | 1.11 ± 0.05 | 1.93 ± 0.00 | 0.14 ± 0.00 | 0.87 ± 0.00 | 0.55 ± 0.60 | 0.88/0.72 |
| | 256 F | 0.35 ± 0.01 | 0.16 ± 0.00 | 0.01 ± 0.00 | 0.12 ± 0.00 | 0.11 ± 0.00 | 0.76 ± 0.03 | 1.11 ± 0.05 | 1.94 ± 0.00 | 0.15 ± 0.00 | 0.84 ± 0.00 | 0.55 ± 0.59 | 0.50/0.18 |
| 100 diagonal (D) | 16 D | 1.69 ± 0.49 | 0.26 ± 0.04 | 0.18 ± 0.07 | 0.34 ± 0.02 | 1.01 ± 0.20 | 1.45 ± 0.10 | 3.59 ± 0.25 | 3.72 ± 0.72 | 0.44 ± 0.20 | 2.37 ± 0.09 | 1.51 ± 1.32 | 1.00 / 0.99 |
| | 32 D | 0.67 ± 0.24 | 0.18 ± 0.01 | 0.06 ± 0.05 | 0.31 ± 0.06 | 0.35 ± 0.08 | 1.04 ± 0.15 | 1.97 ± 0.13 | 2.88 ± 0.70 | 0.22 ± 0.01 | 2.12 ± 0.07 | 0.98 ± 0.98 | 1.00 / 0.98 |
| | 64 D | 0.39 ± 0.06 | 0.16 ± 0.00 | 0.01 ± 0.00 | 0.18 ± 0.02 | 0.14 ± 0.01 | 0.86 ± 0.02 | 1.39 ± 0.07 | 2.18 ± 0.04 | 0.17 ± 0.00 | 1.79 ± 0.02 | 0.73 ± 0.76 | 1.00 / 0.96 |
| | 128 D | 0.32 ± 0.00 | 0.15 ± 0.00 | 0.01 ± 0.00 | 0.12 ± 0.00 | 0.10 ± 0.00 | 0.79 ± 0.02 | 1.19 ± 0.02 | 2.00 ± 0.01 | 0.14 ± 0.01 | 1.24 ± 0.04 | 0.61 ± 0.65 | 1.00 / 0.92 |
| | 256 D | 0.32 ± 0.00 | 0.15 ± 0.00 | 0.01 ± 0.00 | 0.12 ± 0.00 | 0.10 ± 0.00 | 0.77 ± 0.02 | 1.16 ± 0.00 | 1.94 ± 0.00 | 0.13 ± 0.00 | 0.96 ± 0.01 | 0.56 ± 0.61 | 1.00 / 0.84 |
| 100 full (F) | 16 F | 0.66 ± 0.07 | 0.19 ± 0.01 | 0.03 ± 0.01 | 0.25 ± 0.02 | 0.29 ± 0.02 | 0.99 ± 0.07 | 2.50 ± 0.51 | 2.63 ± 0.03 | 0.24 ± 0.02 | 2.21 ± 0.08 | 1.00 ± 1.01 | 1.00/0.99 |
| | 32 F | 0.40 ± 0.01 | 0.17 ± 0.00 | 0.01 ± 0.00 | 0.15 ± 0.01 | 0.13 ± 0.01 | 0.85 ± 0.02 | 1.53 ± 0.12 | 2.17 ± 0.06 | 0.15 ± 0.01 | 1.93 ± 0.04 | 0.75 ± 0.80 | 0.99/0.93 |
| | 64 F | 0.34 ± 0.01 | 0.15 ± 0.00 | 0.01 ± 0.00 | 0.12 ± 0.00 | 0.11 ± 0.00 | 0.79 ± 0.01 | 1.23 ± 0.07 | 1.98 ± 0.01 | 0.15 ± 0.00 | 1.26 ± 0.01 | 0.61 ± 0.65 | 0.97/0.93 |
| | 128 F | 0.32 ± 0.00 | 0.15 ± 0.00 | 0.01 ± 0.00 | 0.12 ± 0.00 | 0.10 ± 0.00 | 0.77 ± 0.02 | 1.16 ± 0.01 | 1.94 ± 0.00 | 0.13 ± 0.00 | 0.99 ± 0.01 | 0.57 ± 0.61 | 0.88/0.80 |
| | 256 F | 0.33 ± 0.00 | 0.16 ± 0.00 | 0.00 ± 0.00 | 0.12 ± 0.00 | 0.10 ± 0.00 | 0.76 ± 0.02 | 1.15 ± 0.01 | 1.93 ± 0.00 | 0.14 ± 0.00 | 0.86 ± 0.00 | 0.56 ± 0.60 | 0.50/0.34 |
| 100 w/clusters (C) | 16 C 5 | 0.34 ± 0.01 | 0.15 ± 0.00 | 0.01 ± 0.00 | 0.14 ± 0.01 | 0.11 ± 0.00 | 0.79 ± 0.02 | 0.97 ± 0.27 | 1.97 ± 0.00 | 0.13 ± 0.00 | 0.77 ± 0.25 | 0.54 ± 0.58 | 1.00/0.95 |
| | 16 C 7 | 0.34 ± 0.01 | 0.15 ± 0.00 | 0.01 ± 0.00 | 0.13 ± 0.00 | 0.10 ± 0.00 | 0.78 ± 0.02 | 0.96 ± 0.27 | 1.96 ± 0.00 | 0.14 ± 0.00 | 0.74 ± 0.22 | 0.53 ± 0.57 | 1.00/0.93 |
| 500 diagonal (D) | 16 D | 2.95 ± 0.28 | 0.73 ± 0.29 | 0.27 ± 0.09 | 0.67 ± 0.28 | 0.52 ± 0.07 | 2.06 ± 0.30 | 4.85 ± 0.31 | 3.94 ± 0.42 | 0.50 ± 0.05 | 2.50 ± 0.03 | 1.94 ± 1.59 | 1.00 / 1.00 |
| | 32 D | 2.33 ± 0.30 | 0.62 ± 0.17 | 0.24 ± 0.05 | 0.50 ± 0.16 | 0.37 ± 0.07 | 1.86 ± 0.25 | 4.73 ± 0.35 | 3.81 ± 0.59 | 0.39 ± 0.04 | 2.46 ± 0.05 | 1.77 ± 1.57 | 1.00 / 1.00 |
| | 64 D | 1.67 ± 0.18 | 0.43 ± 0.16 | 0.13 ± 0.04 | 0.29 ± 0.02 | 0.23 ± 0.02 | 1.32 ± 0.28 | 3.99 ± 0.36 | 3.41 ± 0.28 | 0.32 ± 0.05 | 2.35 ± 0.11 | 1.45 ± 1.39 | 1.00 / 0.99 |
| | 128 D | 1.12 ± 0.02 | 0.23 ± 0.04 | 0.04 ± 0.03 | 0.21 ± 0.04 | 0.22 ± 0.03 | 1.08 ± 0.06 | 3.05 ± 0.87 | 3.09 ± 0.37 | 0.26 ± 0.03 | 2.31 ± 0.04 | 1.19 ± 1.21 | 1.00 / 0.98 |
| | 256 D | 0.54 ± 0.03 | 0.18 ± 0.01 | 0.01 ± 0.00 | 0.16 ± 0.01 | 0.15 ± 0.01 | 0.92 ± 0.08 | 2.42 ± 0.14 | 2.51 ± 0.13 | 0.19 ± 0.01 | 2.09 ± 0.02 | 0.94 ± 0.99 | 1.00 / 0.97 |
| 500 full (F) | 16 F | 2.14 ± 0.06 | 0.70 ± 0.04 | 0.28 ± 0.00 | 0.27 ± 0.01 | 0.21 ± 0.00 | 1.14 ± 0.04 | 3.06 ± 0.27 | 2.71 ± 0.01 | 0.34 ± 0.01 | 2.21 ± 0.01 | 1.33 ± 1.09 | 1.00/1.00 |
| | 32 F | 1.17 ± 0.07 | 0.48 ± 0.03 | 0.08 ± 0.04 | 0.21 ± 0.01 | 0.17 ± 0.00 | 0.99 ± 0.04 | 2.69 ± 0.10 | 2.47 ± 0.02 | 0.25 ± 0.02 | 2.11 ± 0.04 | 1.08 ± 0.99 | 0.99/0.99 |
| | 64 F | 0.51 ± 0.03 | 0.21 ± 0.04 | 0.02 ± 0.00 | 0.17 ± 0.01 | 0.14 ± 0.00 | 0.88 ± 0.04 | 2.19 ± 0.14 | 2.34 ± 0.03 | 0.20 ± 0.00 | 1.97 ± 0.02 | 0.89 ± 0.91 | 0.97/0.96 |
| | 128 F | 0.39 ± 0.01 | 0.16 ± 0.00 | 0.01 ± 0.00 | 0.13 ± 0.00 | 0.11 ± 0.00 | 0.81 ± 0.03 | 1.42 ± 0.07 | 2.03 ± 0.01 | 0.16 ± 0.00 | 1.71 ± 0.01 | 0.71 ± 0.74 | 0.88/0.86 |
| | 256 F | 0.32 ± 0.01 | 0.15 ± 0.00 | 0.01 ± 0.00 | 0.12 ± 0.00 | 0.10 ± 0.00 | 0.77 ± 0.01 | 1.18 ± 0.04 | 1.96 ± 0.00 | 0.14 ± 0.01 | 1.25 ± 0.00 | 0.61 ± 0.65 | 0.50/0.47 |
| 500 w/clusters (C) | 16 C 7 | 0.40 | 0.18 | 0.01 | 0.15 | 0.13 | 0.90 | 2.03 | 2.21 | 0.16 | 1.50 | 0.77 | 1.00/0.98 |
| | 16 C 10 | 0.36 | 0.16 | 0.01 | 0.14 | 0.13 | 0.87 | 2.19 | 2.04 | 0.15 | 1.38 | 0.74 | 1.00/0.98 |
| | 16 C 25 | 0.32 | 0.16 | 0.01 | 0.13 | 0.10 | 0.81 | 1.28 | 1.96 | 0.12 | 1.07 | 0.60 | 1.00/0.95 |
| | 64 C 5 | 0.36 | 0.16 | 0.01 | 0.12 | 0.10 | 0.80 | 1.17 | 1.98 | 0.14 | 1.17 | 0.60 | 0.97/0.93 |
| | 64 C 7 | 0.34 | 0.15 | 0.01 | 0.12 | 0.10 | 0.79 | 1.14 | 1.96 | 0.13 | 1.08 | 0.58 | 0.97/0.91 |
| 1000 w/clusters (C) | 16 C 25 | 0.37 | 0.16 | 0.01 | 0.13 | 0.13 | 0.86 | 2.12 | 2.04 | 0.14 | 1.35 | 0.73 | 1.00/0.97 |

Table 12: **Absolute** In-Distribution test loss for various tasks and methods

| Model Type | Method Type | Tasks | | | | | | | | | | Average | Para. Saved |
|---|---|---|---|---|---|---|---|---|---|---|---|---|---|
| | | task039 | task190 | task280 | task290 | task391 | task442 | task620 | task1342 | task1391 | task1598 | | |
| | base | $0.00 \pm 0.00$ | $0.00 \pm 0.00$ | $1.00 \pm 0.00$ | $0.00 \pm 0.00$ | $0.00 \pm 0.00$ | $0.00 \pm 0.00$ | $0.00 \pm 0.00$ | $0.00 \pm 0.00$ | $0.00 \pm 0.00$ | $0.00 \pm 0.00$ | $0.10 \pm 0.30$ | 1.00 / 1.00 |
| | lora | $100.00 \pm 0.00$ | $100.00 \pm 0.00$ | $100.00 \pm 0.00$ | $100.00 \pm 0.00$ | $100.00 \pm 0.00$ | $100.00 \pm 0.00$ | $100.00 \pm 0.00$ | $100.00 \pm 0.00$ | $100.00 \pm 0.00$ | $100.00 \pm 0.00$ | $100.00 \pm 0.00$ | 0.00 / 0.00 |
| TIES | 10 | $41.00 \pm 0.00$ | $53.67 \pm 0.58$ | $44.33 \pm 4.04$ | $10.33 \pm 0.58$ | $46.33 \pm 4.04$ | $1.00 \pm 0.00$ | $8.00 \pm 0.00$ | $8.00 \pm 0.00$ | $76.67 \pm 1.15$ | $1.00 \pm 0.00$ | $29.03 \pm 25.69$ | 1.00 / 1.00 |
| | 50 | $24.00 \pm 0.00$ | $38.67 \pm 0.58$ | $17.67 \pm 4.62$ | $2.00 \pm 0.00$ | $56.33 \pm 0.58$ | $1.00 \pm 0.00$ | $8.00 \pm 0.00$ | $8.00 \pm 0.00$ | $29.67 \pm 2.89$ | $0.00 \pm 0.00$ | $18.53 \pm 18.07$ | 1.00 / 1.00 |
| | 100 | $22.00 \pm 0.00$ | $38.00 \pm 0.00$ | $18.67 \pm 4.62$ | $1.00 \pm 1.73$ | $51.67 \pm 4.62$ | $1.00 \pm 0.00$ | $8.00 \pm 0.00$ | $7.33 \pm 0.58$ | $2.00 \pm 0.00$ | $0.00 \pm 0.00$ | $14.97 \pm 17.20$ | 1.00 / 1.00 |
| | 500 | $8.00 \pm 0.00$ | $25.00 \pm 0.00$ | $1.00 \pm 0.00$ | $0.00 \pm 0.00$ | $59.00 \pm 0.00$ | $0.00 \pm 0.00$ | $3.00 \pm 0.00$ | $6.00 \pm 0.00$ | $2.00 \pm 0.00$ | $0.00 \pm 0.00$ | $9.90 \pm 18.12$ | 1.00 / 1.00 |
| SVD | SVD 2 | $88.33 \pm 0.65$ | $91.91 \pm 0.94$ | $100.00 \pm 0.00$ | $97.25 \pm 0.45$ | $92.83 \pm 0.39$ | $76.50 \pm 1.51$ | $66.00 \pm 1.41$ | $58.08 \pm 1.16$ | $98.67 \pm 0.49$ | $5.83 \pm 0.94$ | $77.42 \pm 27.69$ | 0.88 / 0.88 |
| | SVD 4 | $93.00 \pm 0.00$ | $96.64 \pm 0.50$ | $100.00 \pm 0.00$ | $100.00 \pm 0.00$ | $96.75 \pm 0.87$ | $88.83 \pm 1.53$ | $90.67 \pm 1.23$ | $72.17 \pm 0.58$ | $98.67 \pm 0.49$ | $16.67 \pm 1.78$ | $85.24 \pm 24.39$ | 0.75 / 0.75 |
| | SVD 8 | $98.89 \pm 0.60$ | $98.55 \pm 0.52$ | $100.00 \pm 0.00$ | $100.00 \pm 0.00$ | $99.42 \pm 0.51$ | $93.44 \pm 0.73$ | $97.22 \pm 0.44$ | $82.78 \pm 1.64$ | $99.00 \pm 0.00$ | $60.00 \pm 0.87$ | $93.70 \pm 11.59$ | 0.50 / 0.50 |
| | SVD 16 | $100.00 \pm 0.00$ | $100.00 \pm 0.00$ | $100.00 \pm 0.00$ | $100.00 \pm 0.00$ | $100.00 \pm 0.00$ | $99.67 \pm 0.50$ | $99.50 \pm 0.55$ | $99.67 \pm 1.15$ | $100.00 \pm 0.00$ | $98.11 \pm 0.78$ | $99.69 \pm 0.68$ | 0.00 / 0.00 |
| 10 diagonal (D) | 16 D | $83.33 \pm 1.53$ | $88.33 \pm 0.58$ | $100.00 \pm 0.00$ | $97.00 \pm 2.00$ | $88.33 \pm 1.15$ | $57.00 \pm 1.00$ | $48.67 \pm 3.21$ | $50.67 \pm 4.93$ | $97.67 \pm 1.53$ | $5.33 \pm 1.15$ | $71.63 \pm 29.53$ | 1.00 / 0.90 |
| | 32 D | $93.00 \pm 1.00$ | $95.33 \pm 0.58$ | $100.00 \pm 0.00$ | $98.00 \pm 1.00$ | $93.67 \pm 1.53$ | $80.67 \pm 2.31$ | $78.67 \pm 1.15$ | $68.00 \pm 1.73$ | $98.33 \pm 0.58$ | $14.67 \pm 2.52$ | $82.03 \pm 24.99$ | 1.00 / 0.80 |
| | 64 D | $99.00 \pm 0.00$ | $97.00 \pm 1.00$ | $100.00 \pm 0.00$ | $100.00 \pm 0.00$ | $98.00 \pm 0.00$ | $90.67 \pm 1.53$ | $95.33 \pm 1.15$ | $79.67 \pm 1.53$ | $99.00 \pm 0.00$ | $55.00 \pm 4.36$ | $91.37 \pm 13.78$ | 1.00 / 0.60 |
| | 128 D | $100.00 \pm 0.00$ | $99.33 \pm 0.58$ | $100.00 \pm 0.00$ | $100.00 \pm 0.00$ | $100.00 \pm 0.00$ | $96.67 \pm 1.53$ | $98.33 \pm 1.15$ | $95.67 \pm 2.31$ | $100.00 \pm 0.00$ | $91.67 \pm 3.51$ | $98.17 \pm 2.94$ | 1.00 / 0.20 |
| | 256 D | $100.00 \pm 0.00$ | $100.00 \pm 0.00$ | $100.00 \pm 0.00$ | $100.00 \pm 0.00$ | $100.00 \pm 0.00$ | $100.00 \pm 0.00$ | $100.00 \pm 0.00$ | $99.33 \pm 1.15$ | $100.00 \pm 0.00$ | $95.00 \pm 1.00$ | $99.43 \pm 1.57$ | 1.00 / -0.60 |
| 10 full (F) | 16 F | $83.00 \pm 2.00$ | $93.00 \pm 1.00$ | $100.00 \pm 0.00$ | $98.33 \pm 0.58$ | $91.67 \pm 0.58$ | $64.33 \pm 3.21$ | $59.33 \pm 1.15$ | $52.67 \pm 1.53$ | $98.33 \pm 0.58$ | $6.33 \pm 1.15$ | $74.70 \pm 28.71$ | 1.00/0.90 |
| | 32 F | $91.33 \pm 0.58$ | $96.00 \pm 1.00$ | $100.00 \pm 0.00$ | $98.33 \pm 0.58$ | $94.33 \pm 0.58$ | $84.00 \pm 2.00$ | $83.00 \pm 1.73$ | $70.33 \pm 1.53$ | $99.00 \pm 1.00$ | $22.00 \pm 2.65$ | $83.83 \pm 22.82$ | 0.99/0.79 |
| | 64 F | $99.00 \pm 0.00$ | $97.33 \pm 0.58$ | $100.00 \pm 0.00$ | $100.00 \pm 0.00$ | $99.33 \pm 1.15$ | $91.33 \pm 1.53$ | $96.33 \pm 0.58$ | $81.67 \pm 2.31$ | $99.00 \pm 0.00$ | $58.33 \pm 1.53$ | $92.23 \pm 12.76$ | 0.97/0.57 |
| | 128 F | $99.67 \pm 0.58$ | $99.33 \pm 0.58$ | $100.00 \pm 0.00$ | $100.00 \pm 0.00$ | $100.00 \pm 0.00$ | $97.67 \pm 1.15$ | $100.00 \pm 0.00$ | $95.67 \pm 1.15$ | $100.00 \pm 0.00$ | $91.00 \pm 1.00$ | $98.33 \pm 2.89$ | 0.88/0.07 |
| | 256 F | $100.00 \pm 0.00$ | $100.00 \pm 0.00$ | $100.00 \pm 0.00$ | $100.00 \pm 0.00$ | $100.00 \pm 0.00$ | $99.67 \pm 0.58$ | $99.67 \pm 0.58$ | $99.67 \pm 0.58$ | $100.00 \pm 0.00$ | $98.00 \pm 1.00$ | $99.70 \pm 0.70$ | 0.50/-1.10 |
| 50 diagonal (D) | 16 D | $52.67 \pm 4.51$ | $86.67 \pm 3.06$ | $100.00 \pm 0.00$ | $85.00 \pm 3.46$ | $65.33 \pm 3.79$ | $25.33 \pm 5.03$ | $10.00 \pm 1.00$ | $10.67 \pm 10.26$ | $81.00 \pm 6.56$ | $0.00 \pm 0.00$ | $51.67 \pm 36.18$ | 1.00 / 0.98 |
| | 32 D | $69.67 \pm 3.21$ | $88.67 \pm 1.53$ | $100.00 \pm 0.00$ | $95.00 \pm 2.00$ | $80.00 \pm 3.00$ | $36.67 \pm 3.51$ | $17.00 \pm 2.65$ | $26.33 \pm 5.03$ | $95.00 \pm 2.00$ | $0.00 \pm 0.00$ | $60.83 \pm 36.02$ | 1.00 / 0.96 |
| | 64 D | $79.67 \pm 2.52$ | $91.00 \pm 1.00$ | $100.00 \pm 0.00$ | $97.67 \pm 0.58$ | $88.00 \pm 1.00$ | $52.00 \pm 1.00$ | $36.67 \pm 5.69$ | $41.33 \pm 1.15$ | $96.00 \pm 1.00$ | $0.33 \pm 0.58$ | $68.27 \pm 32.66$ | 1.00 / 0.92 |
| | 128 D | $90.00 \pm 1.00$ | $91.33 \pm 0.58$ | $100.00 \pm 0.00$ | $98.33 \pm 0.58$ | $90.67 \pm 2.08$ | $73.67 \pm 2.08$ | $63.67 \pm 1.53$ | $56.33 \pm 0.58$ | $98.00 \pm 0.00$ | $7.33 \pm 1.15$ | $76.93 \pm 27.79$ | 1.00 / 0.84 |
| | 256 D | $94.67 \pm 1.58$ | $96.33 \pm 0.58$ | $100.00 \pm 0.00$ | $99.67 \pm 0.58$ | $96.33 \pm 1.15$ | $87.33 \pm 0.58$ | $87.00 \pm 2.65$ | $71.67 \pm 1.53$ | $99.67 \pm 0.58$ | $31.67 \pm 1.15$ | $86.43 \pm 20.41$ | 1.00 / 0.68 |
| 50 full (F) | 16 F | $61.67 \pm 3.06$ | $89.67 \pm 1.15$ | $99.67 \pm 0.58$ | $90.67 \pm 2.52$ | $78.33 \pm 3.51$ | $34.00 \pm 1.00$ | $7.00 \pm 3.46$ | $25.00 \pm 6.24$ | $90.00 \pm 1.00$ | $0.00 \pm 0.00$ | $57.60 \pm 36.59$ | 1.00/0.98 |
| | 32 F | $71.00 \pm 1.00$ | $89.00 \pm 1.73$ | $100.00 \pm 0.00$ | $98.00 \pm 0.00$ | $85.00 \pm 1.00$ | $47.00 \pm 1.73$ | $29.00 \pm 3.00$ | $35.00 \pm 2.00$ | $98.00 \pm 1.00$ | $0.00 \pm 0.00$ | $65.20 \pm 34.00$ | 0.99/0.95 |
| | 64 F | $81.67 \pm 0.58$ | $93.67 \pm 1.15$ | $100.00 \pm 0.00$ | $98.33 \pm 0.58$ | $90.67 \pm 2.08$ | $61.67 \pm 1.53$ | $54.33 \pm 1.53$ | $51.33 \pm 1.15$ | $98.33 \pm 0.58$ | $3.33 \pm 0.58$ | $73.33 \pm 29.89$ | 0.97/0.89 |
| | 128 F | $91.00 \pm 1.00$ | $94.33 \pm 0.58$ | $100.00 \pm 0.00$ | $99.00 \pm 0.00$ | $93.33 \pm 1.53$ | $81.67 \pm 0.58$ | $75.00 \pm 1.73$ | $67.67 \pm 2.08$ | $98.67 \pm 0.58$ | $16.67 \pm 0.58$ | $81.73 \pm 24.46$ | 0.88/0.72 |
| | 256 F | $97.00 \pm 0.00$ | $98.00 \pm 0.00$ | $100.00 \pm 0.00$ | $100.00 \pm 0.00$ | $99.67 \pm 0.58$ | $92.00 \pm 0.00$ | $94.33 \pm 1.15$ | $79.67 \pm 1.15$ | $100.00 \pm 0.00$ | $57.33 \pm 2.52$ | $91.70 \pm 13.11$ | 0.50/0.18 |
| 100 diagonal (D) | 16 D | $33.00 \pm 8.19$ | $79.33 \pm 5.69$ | $89.33 \pm 5.51$ | $80.00 \pm 3.61$ | $35.33 \pm 6.03$ | $4.00 \pm 1.73$ | $3.00 \pm 1.00$ | $0.00 \pm 0.00$ | $71.33 \pm 4.51$ | $0.00 \pm 0.00$ | $39.53 \pm 36.15$ | 1.00 / 0.99 |
| | 32 D | $51.00 \pm 7.81$ | $90.00 \pm 1.00$ | $100.00 \pm 0.00$ | $88.00 \pm 7.00$ | $58.67 \pm 11.68$ | $17.67 \pm 10.26$ | $7.67 \pm 2.89$ | $9.33 \pm 12.86$ | $86.33 \pm 2.52$ | $0.00 \pm 0.00$ | $50.87 \pm 38.43$ | 1.00 / 0.98 |
| | 64 D | $68.00 \pm 2.65$ | $87.33 \pm 1.53$ | $100.00 \pm 0.00$ | $94.33 \pm 4.04$ | $80.33 \pm 2.08$ | $38.00 \pm 3.00$ | $19.67 \pm 4.51$ | $28.33 \pm 1.53$ | $92.67 \pm 1.15$ | $0.33 \pm 0.58$ | $60.90 \pm 34.91$ | 1.00 / 0.96 |
| | 128 D | $82.00 \pm 2.00$ | $90.00 \pm 2.00$ | $100.00 \pm 0.00$ | $97.33 \pm 0.58$ | $85.33 \pm 0.58$ | $55.33 \pm 2.08$ | $34.33 \pm 3.79$ | $36.67 \pm 2.52$ | $94.67 \pm 0.58$ | $0.00 \pm 0.00$ | $67.57 \pm 32.95$ | 1.00 / 0.92 |
| | 256 D | $90.00 \pm 1.00$ | $93.00 \pm 2.00$ | $100.00 \pm 0.00$ | $97.67 \pm 0.58$ | $91.67 \pm 0.58$ | $71.67 \pm 5.13$ | $59.67 \pm 1.53$ | $58.00 \pm 0.00$ | $97.67 \pm 0.58$ | $4.00 \pm 1.00$ | $76.33 \pm 28.88$ | 1.00 / 0.84 |
| 100 full (F) | 16 F | $49.00 \pm 2.00$ | $89.67 \pm 3.21$ | $97.00 \pm 3.00$ | $84.33 \pm 3.06$ | $65.33 \pm 2.52$ | $20.67 \pm 8.33$ | $6.33 \pm 2.08$ | $8.33 \pm 4.73$ | $81.33 \pm 2.08$ | $0.00 \pm 0.00$ | $50.20 \pm 37.06$ | 1.00/0.99 |
| | 32 F | $65.00 \pm 3.46$ | $90.33 \pm 1.53$ | $100.00 \pm 0.00$ | $96.33 \pm 1.53$ | $80.00 \pm 2.65$ | $41.33 \pm 3.21$ | $16.00 \pm 0.00$ | $29.33 \pm 2.08$ | $92.00 \pm 2.65$ | $0.00 \pm 0.00$ | $61.03 \pm 35.43$ | 0.99/0.97 |
| | 64 F | $72.33 \pm 0.58$ | $89.67 \pm 1.53$ | $100.00 \pm 0.00$ | $97.67 \pm 0.58$ | $86.00 \pm 1.00$ | $53.00 \pm 1.00$ | $35.33 \pm 1.53$ | $38.00 \pm 1.73$ | $94.67 \pm 0.58$ | $0.00 \pm 0.00$ | $66.67 \pm 32.54$ | 0.97/0.93 |
| | 128 F | $84.33 \pm 1.53$ | $92.33 \pm 1.53$ | $100.00 \pm 0.00$ | $98.00 \pm 0.00$ | $91.33 \pm 0.58$ | $68.67 \pm 0.58$ | $56.00 \pm 1.00$ | $57.67 \pm 1.15$ | $99.00 \pm 0.00$ | $5.33 \pm 0.58$ | $75.27 \pm 28.67$ | 0.88/0.80 |
| | 256 F | $91.67 \pm 1.15$ | $96.67 \pm 0.58$ | $100.00 \pm 0.00$ | $100.00 \pm 0.00$ | $94.33 \pm 0.58$ | $84.67 \pm 0.58$ | $78.00 \pm 0.00$ | $69.67 \pm 0.58$ | $99.00 \pm 0.00$ | $22.00 \pm 1.00$ | $83.60 \pm 23.08$ | 0.50/0.34 |
| 100 w/clusters (C) | 16 C 5 | $74.67 \pm 0.94$ | $91.00 \pm 0.82$ | $100.00 \pm 0.00$ | $96.67 \pm 1.25$ | $87.67 \pm 1.70$ | $53.67 \pm 2.05$ | $40.67 \pm 2.87$ | $41.00 \pm 4.55$ | $97.67 \pm 1.25$ | $0.67 \pm 0.94$ | $68.37 \pm 31.46$ | 1.00/0.95 |
| | 16 C 7 | $77.67 \pm 0.47$ | $90.33 \pm 1.25$ | $100.00 \pm 0.00$ | $97.33 \pm 0.94$ | $90.33 \pm 2.05$ | $58.67 \pm 0.94$ | $49.00 \pm 2.16$ | $49.00 \pm 0.82$ | $97.67 \pm 0.47$ | $3.67 \pm 1.25$ | $71.37 \pm 29.48$ | 1.00/0.93 |
| 500 diagonal (D) | 16 D | $8.00 \pm 3.61$ | $51.50 \pm 3.54$ | $79.67 \pm 4.93$ | $28.00 \pm 42.44$ | $56.67 \pm 2.89$ | $0.67 \pm 1.15$ | $0.33 \pm 0.58$ | $0.00 \pm 0.00$ | $71.67 \pm 6.03$ | $0.00 \pm 0.00$ | $28.90 \pm 33.54$ | 1.00 / 1.00 |
| | 32 D | $14.33 \pm 11.02$ | $52.50 \pm 9.19$ | $80.67 \pm 2.08$ | $43.00 \pm 31.48$ | $60.67 \pm 0.58$ | $0.67 \pm 1.15$ | $1.67 \pm 1.15$ | $0.00 \pm 0.00$ | $74.33 \pm 4.04$ | $0.00 \pm 0.00$ | $32.10 \pm 33.43$ | 1.00 / 1.00 |
| | 64 D | $25.67 \pm 1.15$ | $62.50 \pm 12.02$ | $87.33 \pm 4.04$ | $78.33 \pm 1.15$ | $65.33 \pm 3.06$ | $5.33 \pm 3.51$ | $3.67 \pm 1.15$ | $1.00 \pm 1.73$ | $76.67 \pm 2.31$ | $0.00 \pm 0.00$ | $39.83 \pm 35.80$ | 1.00 / 0.99 |
| | 128 D | $38.33 \pm 3.21$ | $85.50 \pm 2.12$ | $96.00 \pm 3.00$ | $81.33 \pm 2.31$ | $65.67 \pm 1.15$ | $11.67 \pm 4.73$ | $5.33 \pm 2.08$ | $2.00 \pm 1.00$ | $80.00 \pm 5.00$ | $0.00 \pm 0.00$ | $45.24 \pm 37.78$ | 1.00 / 0.98 |
| | 256 D | $53.33 \pm 0.58$ | $91.00 \pm 2.83$ | $100.00 \pm 0.00$ | $89.00 \pm 2.65$ | $76.00 \pm 2.00$ | $20.67 \pm 6.81$ | $6.00 \pm 1.73$ | $12.00 \pm 8.00$ | $86.33 \pm 2.31$ | $0.00 \pm 0.00$ | $52.14 \pm 38.64$ | 1.00 / 0.97 |
| 500 full (F) | 16 F | $8.33 \pm 2.08$ | $41.00 \pm 5.66$ | $76.67 \pm 0.58$ | $78.00 \pm 0.00$ | $72.67 \pm 0.58$ | $6.00 \pm 0.00$ | $5.67 \pm 0.58$ | $0.00 \pm 0.00$ | $78.00 \pm 1.00$ | $0.00 \pm 0.00$ | $36.48 \pm 35.46$ | 1.00/1.00 |
| | 32 F | $33.67 \pm 4.16$ | $51.00 \pm 1.41$ | $92.67 \pm 1.53$ | $77.00 \pm 1.73$ | $75.00 \pm 2.00$ | $14.33 \pm 1.53$ | $8.00 \pm 0.00$ | $0.00 \pm 0.00$ | $80.67 \pm 1.53$ | $0.00 \pm 0.00$ | $42.97 \pm 35.80$ | 0.99/0.99 |
| | 64 F | $56.00 \pm 2.65$ | $85.50 \pm 0.71$ | $94.33 \pm 0.58$ | $89.33 \pm 2.89$ | $74.33 \pm 1.15$ | $36.33 \pm 1.15$ | $9.00 \pm 1.00$ | $2.67 \pm 1.15$ | $84.00 \pm 1.00$ | $0.00 \pm 0.00$ | $52.03 \pm 36.92$ | 0.97/0.96 |
| | 128 F | $69.33 \pm 0.58$ | $88.50 \pm 0.71$ | $99.00 \pm 0.00$ | $96.33 \pm 1.53$ | $80.33 \pm 1.15$ | $45.00 \pm 2.00$ | $16.33 \pm 0.58$ | $31.00 \pm 1.73$ | $92.00 \pm 0.00$ | $0.00 \pm 0.00$ | $60.86 \pm 35.07$ | 0.88/0.86 |
| | 256 F | $79.67 \pm 0.58$ | $89.50 \pm 0.71$ | $100.00 \pm 0.00$ | $97.67 \pm 0.58$ | $87.33 \pm 0.58$ | $57.00 \pm 1.00$ | $35.00 \pm 1.00$ | $42.00 \pm 1.00$ | $95.00 \pm 1.00$ | $0.00 \pm 0.00$ | $67.59 \pm 32.67$ | 0.50/0.47 |
| 500 w/clusters (C) | 16 C 7 | 63.00 | 90.00 | 99.00 | 96.00 | 78.00 | 31.00 | 9.00 | 15.00 | 89.00 | 1.00 | 57.10 | 1.00/0.98 |
| | 16 C 10 | 69.00 | 93.00 | 100.00 | 98.00 | 81.00 | 34.00 | 8.00 | 33.00 | 95.00 | 1.00 | 61.20 | 1.00/0.98 |
| | 16 C 25 | 79.00 | 90.00 | 100.00 | 97.00 | 88.00 | 53.00 | 38.00 | 48.00 | 98.00 | 0.00 | 69.10 | 1.00/0.95 |
| | 64 C 5 | 77.00 | 88.00 | 100.00 | 98.00 | 89.00 | 56.00 | 39.00 | 42.00 | 99.00 | 0.00 | 68.80 | 0.97/0.93 |
| | 64 C 7 | 76.00 | 90.00 | 100.00 | 97.00 | 89.00 | 60.00 | 48.00 | 49.00 | 99.00 | 3.00 | 71.10 | 0.97/0.91 |
| 1000 w/clusters (C) | 16 C 25 | 73.00 | 90.00 | 100.00 | 98.00 | 77.00 | 39.00 | 8.00 | 34.00 | 96.00 | 1.00 | 61.60 | 1.00/0.97 |

Table 13: **Absolute** In-Distribution agreement for various tasks and methods

| Model Type | Method Type | Tasks | | | | | | | | | | Average |
|---|---|---|---|---|---|---|---|---|---|---|---|---|
| | | task039 | task190 | task280 | task290 | task391 | task442 | task620 | task1342 | task1391 | task1598 | |
| SVD | SVD 2 | 0.29 ± 0.00 | 0.43 ± 0.00 | 0.31 ± 0.00 | 0.40 ± 0.00 | 0.38 ± 0.00 | 0.31 ± 0.00 | 0.37 ± 0.00 | 0.31 ± 0.00 | 0.42 ± 0.00 | 0.30 ± 0.00 | 0.35 ± 0.05 |
| | SVD 4 | 0.16 ± 0.00 | 0.24 ± 0.00 | 0.16 ± 0.00 | 0.25 ± 0.00 | 0.23 ± 0.00 | 0.17 ± 0.00 | 0.22 ± 0.00 | 0.16 ± 0.00 | 0.25 ± 0.00 | 0.16 ± 0.00 | 0.20 ± 0.04 |
| | SVD 8 | 0.06 ± 0.00 | 0.09 ± 0.00 | 0.06 ± 0.00 | 0.11 ± 0.00 | 0.10 ± 0.00 | 0.07 ± 0.00 | 0.09 ± 0.00 | 0.06 ± 0.00 | 0.11 ± 0.00 | 0.06 ± 0.00 | 0.08 ± 0.02 |
| 10 diagonal (D) | 16 D | 0.37 ± 0.02 | 0.51 ± 0.02 | 0.36 ± 0.01 | 0.57 ± 0.02 | 0.55 ± 0.00 | 0.39 ± 0.02 | 0.49 ± 0.01 | 0.36 ± 0.02 | 0.53 ± 0.03 | 0.39 ± 0.01 | 0.45 ± 0.08 |
| | 32 D | 0.21 ± 0.01 | 0.28 ± 0.00 | 0.20 ± 0.01 | 0.35 ± 0.00 | 0.33 ± 0.01 | 0.22 ± 0.01 | 0.31 ± 0.01 | 0.20 ± 0.01 | 0.32 ± 0.01 | 0.22 ± 0.00 | 0.26 ± 0.06 |
| | 64 D | 0.10 ± 0.00 | 0.11 ± 0.01 | 0.09 ± 0.00 | 0.18 ± 0.01 | 0.18 ± 0.00 | 0.10 ± 0.00 | 0.15 ± 0.01 | 0.09 ± 0.00 | 0.14 ± 0.00 | 0.09 ± 0.00 | 0.12 ± 0.04 |
| | 128 D | 0.02 ± 0.00 | 0.01 ± 0.00 | 0.02 ± 0.00 | 0.03 ± 0.00 | 0.04 ± 0.00 | 0.02 ± 0.00 | 0.03 ± 0.00 | 0.02 ± 0.00 | 0.02 ± 0.00 | 0.02 ± 0.00 | 0.03 ± 0.01 |
| | 256 D | 0.00 ± 0.00 | 0.00 ± 0.00 | 0.00 ± 0.00 | 0.00 ± 0.00 | 0.00 ± 0.00 | 0.00 ± 0.00 | 0.00 ± 0.00 | 0.00 ± 0.00 | 0.00 ± 0.00 | 0.00 ± 0.00 | 0.00 ± 0.00 |
| 10 full (F) | 16 F | 0.35 ± 0.00 | 0.46 ± 0.00 | 0.34 ± 0.00 | 0.51 ± 0.00 | 0.47 ± 0.01 | 0.36 ± 0.01 | 0.45 ± 0.01 | 0.35 ± 0.01 | 0.49 ± 0.00 | 0.35 ± 0.01 | 0.41 ± 0.06 |
| | 32 F | 0.20 ± 0.00 | 0.24 ± 0.00 | 0.20 ± 0.00 | 0.30 ± 0.00 | 0.29 ± 0.00 | 0.22 ± 0.00 | 0.27 ± 0.00 | 0.20 ± 0.00 | 0.27 ± 0.00 | 0.21 ± 0.00 | 0.24 ± 0.04 |
| | 64 F | 0.10 ± 0.00 | 0.10 ± 0.00 | 0.09 ± 0.00 | 0.13 ± 0.00 | 0.13 ± 0.00 | 0.10 ± 0.00 | 0.12 ± 0.00 | 0.09 ± 0.00 | 0.12 ± 0.00 | 0.10 ± 0.00 | 0.11 ± 0.02 |
| | 128 F | 0.02 ± 0.00 | 0.02 ± 0.00 | 0.02 ± 0.00 | 0.01 ± 0.00 | 0.02 ± 0.00 | 0.02 ± 0.00 | 0.02 ± 0.00 | 0.02 ± 0.00 | 0.01 ± 0.00 | 0.02 ± 0.00 | 0.02 ± 0.00 |
| | 256 F | 0.00 ± 0.00 | 0.00 ± 0.00 | 0.00 ± 0.00 | 0.00 ± 0.00 | 0.00 ± 0.00 | 0.00 ± 0.00 | 0.00 ± 0.00 | 0.00 ± 0.00 | 0.00 ± 0.00 | 0.00 ± 0.00 | 0.00 ± 0.00 |
| 50 diagonal (D) | 16 D | 0.66 ± 0.01 | 0.69 ± 0.01 | 0.88 ± 0.01 | 0.76 ± 0.03 | 0.95 ± 0.02 | 0.91 ± 0.01 | 0.83 ± 0.02 | 0.88 ± 0.03 | 0.72 ± 0.02 | 0.88 ± 0.02 | 0.82 ± 0.10 |
| | 32 D | 0.50 ± 0.01 | 0.52 ± 0.02 | 0.73 ± 0.01 | 0.58 ± 0.03 | 0.88 ± 0.03 | 0.79 ± 0.03 | 0.72 ± 0.01 | 0.75 ± 0.01 | 0.57 ± 0.02 | 0.75 ± 0.01 | 0.68 ± 0.12 |
| | 64 D | 0.34 ± 0.01 | 0.37 ± 0.01 | 0.52 ± 0.00 | 0.38 ± 0.01 | 0.71 ± 0.02 | 0.58 ± 0.01 | 0.54 ± 0.00 | 0.56 ± 0.00 | 0.44 ± 0.01 | 0.58 ± 0.01 | 0.50 ± 0.11 |
| | 128 D | 0.21 ± 0.01 | 0.22 ± 0.01 | 0.31 ± 0.00 | 0.22 ± 0.01 | 0.51 ± 0.01 | 0.42 ± 0.01 | 0.38 ± 0.00 | 0.39 ± 0.00 | 0.27 ± 0.00 | 0.40 ± 0.00 | 0.33 ± 0.10 |
| | 256 D | 0.10 ± 0.00 | 0.12 ± 0.00 | 0.16 ± 0.00 | 0.10 ± 0.00 | 0.29 ± 0.01 | 0.21 ± 0.00 | 0.19 ± 0.00 | 0.23 ± 0.01 | 0.15 ± 0.00 | 0.20 ± 0.00 | 0.18 ± 0.06 |
| 50 full (F) | 16 F | 0.57 ± 0.01 | 0.60 ± 0.01 | 0.86 ± 0.01 | 0.71 ± 0.02 | 0.95 ± 0.01 | 0.88 ± 0.01 | 0.81 ± 0.00 | 0.83 ± 0.01 | 0.67 ± 0.01 | 0.86 ± 0.01 | 0.78 ± 0.12 |
| | 32 F | 0.47 ± 0.01 | 0.48 ± 0.01 | 0.71 ± 0.00 | 0.55 ± 0.01 | 0.78 ± 0.01 | 0.69 ± 0.01 | 0.69 ± 0.00 | 0.65 ± 0.01 | 0.53 ± 0.01 | 0.71 ± 0.00 | 0.63 ± 0.11 |
| | 64 F | 0.33 ± 0.00 | 0.35 ± 0.00 | 0.45 ± 0.00 | 0.36 ± 0.00 | 0.56 ± 0.00 | 0.50 ± 0.00 | 0.47 ± 0.00 | 0.49 ± 0.00 | 0.39 ± 0.00 | 0.49 ± 0.00 | 0.44 ± 0.08 |
| | 128 F | 0.19 ± 0.00 | 0.21 ± 0.00 | 0.25 ± 0.00 | 0.19 ± 0.00 | 0.35 ± 0.00 | 0.30 ± 0.00 | 0.28 ± 0.00 | 0.31 ± 0.00 | 0.24 ± 0.00 | 0.30 ± 0.00 | 0.26 ± 0.05 |
| | 256 F | 0.09 ± 0.00 | 0.10 ± 0.00 | 0.10 ± 0.00 | 0.08 ± 0.00 | 0.16 ± 0.00 | 0.13 ± 0.00 | 0.12 ± 0.00 | 0.15 ± 0.00 | 0.11 ± 0.00 | 0.13 ± 0.00 | 0.12 ± 0.02 |
| 100 diagonal (D) | 16 D | 0.90 ± 0.01 | 0.85 ± 0.01 | 0.87 ± 0.03 | 0.88 ± 0.02 | 0.68 ± 0.02 | 0.91 ± 0.01 | 0.97 ± 0.01 | 0.98 ± 0.01 | 0.96 ± 0.01 | 1.00 ± 0.00 | 0.90 ± 0.09 |
| | 32 D | 0.83 ± 0.02 | 0.77 ± 0.00 | 0.77 ± 0.01 | 0.78 ± 0.00 | 0.55 ± 0.02 | 0.79 ± 0.01 | 0.94 ± 0.02 | 0.94 ± 0.03 | 0.87 ± 0.00 | 0.98 ± 0.01 | 0.82 ± 0.12 |
| | 64 D | 0.67 ± 0.00 | 0.63 ± 0.00 | 0.59 ± 0.02 | 0.63 ± 0.01 | 0.40 ± 0.00 | 0.62 ± 0.00 | 0.86 ± 0.02 | 0.82 ± 0.02 | 0.71 ± 0.03 | 0.93 ± 0.00 | 0.68 ± 0.15 |
| | 128 D | 0.49 ± 0.01 | 0.47 ± 0.00 | 0.42 ± 0.01 | 0.45 ± 0.00 | 0.27 ± 0.02 | 0.44 ± 0.01 | 0.73 ± 0.01 | 0.69 ± 0.02 | 0.59 ± 0.02 | 0.80 ± 0.02 | 0.53 ± 0.16 |
| | 256 D | 0.32 ± 0.00 | 0.31 ± 0.00 | 0.26 ± 0.01 | 0.30 ± 0.00 | 0.15 ± 0.01 | 0.28 ± 0.00 | 0.51 ± 0.02 | 0.51 ± 0.02 | 0.40 ± 0.01 | 0.61 ± 0.01 | 0.36 ± 0.14 |
| 100 full (F) | 16 F | 0.88 ± 0.00 | 0.82 ± 0.00 | 0.84 ± 0.01 | 0.86 ± 0.00 | 0.67 ± 0.01 | 0.88 ± 0.01 | 0.99 ± 0.00 | 0.96 ± 0.01 | 0.91 ± 0.01 | 1.00 ± 0.00 | 0.88 ± 0.09 |
| | 32 F | 0.78 ± 0.00 | 0.72 ± 0.00 | 0.73 ± 0.00 | 0.74 ± 0.00 | 0.52 ± 0.00 | 0.74 ± 0.01 | 0.94 ± 0.01 | 0.89 ± 0.00 | 0.77 ± 0.02 | 0.99 ± 0.00 | 0.78 ± 0.13 |
| | 64 F | 0.60 ± 0.00 | 0.57 ± 0.00 | 0.57 ± 0.00 | 0.57 ± 0.00 | 0.39 ± 0.00 | 0.56 ± 0.00 | 0.76 ± 0.00 | 0.73 ± 0.00 | 0.60 ± 0.00 | 0.83 ± 0.01 | 0.62 ± 0.12 |
| | 128 F | 0.40 ± 0.00 | 0.38 ± 0.00 | 0.35 ± 0.00 | 0.37 ± 0.00 | 0.25 ± 0.00 | 0.37 ± 0.00 | 0.52 ± 0.00 | 0.54 ± 0.00 | 0.45 ± 0.00 | 0.60 ± 0.00 | 0.42 ± 0.10 |
| | 256 F | 0.21 ± 0.00 | 0.20 ± 0.00 | 0.18 ± 0.00 | 0.19 ± 0.00 | 0.13 ± 0.00 | 0.19 ± 0.00 | 0.30 ± 0.00 | 0.34 ± 0.00 | 0.26 ± 0.00 | 0.38 ± 0.00 | 0.24 ± 0.08 |
| 100 w/clusters (C) | 16 C 5 | 0.46 ± 0.01 | 0.46 ± 0.00 | 0.45 ± 0.00 | 0.47 ± 0.01 | 0.61 ± 0.01 | 0.65 ± 0.01 | 0.61 ± 0.00 | 0.64 ± 0.02 | 0.45 ± 0.00 | 0.59 ± 0.01 | 0.54 ± 0.08 |
| | 16 C 7 | 0.41 ± 0.01 | 0.42 ± 0.01 | 0.39 ± 0.01 | 0.43 ± 0.01 | 0.51 ± 0.01 | 0.56 ± 0.01 | 0.53 ± 0.01 | 0.55 ± 0.01 | 0.42 ± 0.01 | 0.54 ± 0.01 | 0.48 ± 0.06 |
| 500 diagonal (D) | 16 D | 0.97 ± 0.00 | 0.73 ± 0.00 | 0.96 ± 0.00 | 1.00 ± 0.00 | 0.99 ± 0.00 | 0.96 ± 0.01 | 0.90 ± 0.00 | 0.92 ± 0.00 | 1.00 ± 0.00 | 1.00 ± 0.00 | 0.94 ± 0.08 |
| | 32 D | 0.96 ± 0.00 | 0.70 ± 0.00 | 0.92 ± 0.01 | 0.98 ± 0.01 | 0.96 ± 0.01 | 0.93 ± 0.01 | 0.86 ± 0.00 | 0.89 ± 0.00 | 1.00 ± 0.00 | 1.00 ± 0.00 | 0.92 ± 0.09 |
| | 64 D | 0.90 ± 0.01 | 0.65 ± 0.00 | 0.86 ± 0.01 | 0.96 ± 0.02 | 0.90 ± 0.01 | 0.87 ± 0.01 | 0.81 ± 0.00 | 0.83 ± 0.01 | 0.99 ± 0.01 | 1.00 ± 0.00 | 0.88 ± 0.10 |
| | 128 D | 0.82 ± 0.01 | 0.60 ± 0.00 | 0.76 ± 0.00 | 0.90 ± 0.02 | 0.83 ± 0.01 | 0.78 ± 0.02 | 0.74 ± 0.00 | 0.74 ± 0.01 | 0.97 ± 0.01 | 1.00 ± 0.00 | 0.81 ± 0.12 |
| | 256 D | 0.59 ± 0.02 | 0.51 ± 0.00 | 0.56 ± 0.01 | 0.81 ± 0.02 | 0.70 ± 0.02 | 0.55 ± 0.02 | 0.57 ± 0.01 | 0.54 ± 0.01 | 0.91 ± 0.01 | 1.00 ± 0.01 | 0.67 ± 0.17 |
| 500 full (F) | 16 F | 0.94 ± 0.00 | 0.67 ± 0.00 | 0.88 ± 0.00 | 1.00 ± 0.00 | 0.98 ± 0.00 | 0.90 ± 0.00 | 0.82 ± 0.00 | 0.83 ± 0.00 | 1.00 ± 0.00 | 1.00 ± 0.00 | 0.90 ± 0.10 |
| | 32 F | 0.88 ± 0.00 | 0.61 ± 0.00 | 0.81 ± 0.00 | 0.97 ± 0.01 | 0.94 ± 0.00 | 0.84 ± 0.00 | 0.75 ± 0.00 | 0.77 ± 0.00 | 0.99 ± 0.00 | 0.99 ± 0.00 | 0.86 ± 0.12 |
| | 64 F | 0.80 ± 0.00 | 0.55 ± 0.00 | 0.72 ± 0.00 | 0.86 ± 0.00 | 0.82 ± 0.01 | 0.76 ± 0.00 | 0.67 ± 0.00 | 0.70 ± 0.00 | 0.94 ± 0.00 | 0.99 ± 0.00 | 0.78 ± 0.13 |
| | 128 F | 0.64 ± 0.00 | 0.46 ± 0.00 | 0.60 ± 0.00 | 0.74 ± 0.00 | 0.65 ± 0.00 | 0.63 ± 0.00 | 0.56 ± 0.00 | 0.58 ± 0.00 | 0.85 ± 0.00 | 0.96 ± 0.00 | 0.67 ± 0.14 |
| | 256 F | 0.43 ± 0.00 | 0.35 ± 0.00 | 0.44 ± 0.00 | 0.55 ± 0.00 | 0.49 ± 0.00 | 0.45 ± 0.00 | 0.40 ± 0.00 | 0.42 ± 0.00 | 0.67 ± 0.00 | 0.84 ± 0.00 | 0.50 ± 0.14 |
| 500 w/clusters (C) | 16 C 7 | 0.68 | 0.70 | 0.64 | 0.72 | 0.85 | 0.90 | 0.93 | 0.92 | 0.71 | 0.83 | 0.79 |
| | 16 C 10 | 0.61 | 0.65 | 0.61 | 0.66 | 0.84 | 0.86 | 0.88 | 0.84 | 0.62 | 0.76 | 0.73 |
| | 16 C 25 | 0.42 | 0.41 | 0.42 | 0.44 | 0.57 | 0.64 | 0.63 | 0.62 | 0.40 | 0.58 | 0.51 |
| | 64 C 5 | 0.49 | 0.49 | 0.45 | 0.51 | 0.64 | 0.66 | 0.62 | 0.67 | 0.50 | 0.65 | 0.57 |
| | 64 C 7 | 0.45 | 0.45 | 0.41 | 0.45 | 0.56 | 0.58 | 0.55 | 0.59 | 0.44 | 0.57 | 0.51 |
| 1000 w/clusters (C) | 16 C 25 | 0.58 | 0.64 | 0.54 | 0.64 | 0.81 | 0.87 | 0.90 | 0.84 | 0.57 | 0.74 | 0.71 |

Table 14: **Reconstruction error** In-Distribution for various tasks and methods

| Model Type | Method Type | Tasks | | | | | | | | | | Average |
|---|---|---|---|---|---|---|---|---|---|---|---|---|
| | | task039 | task190 | task280 | task290 | task391 | task442 | task620 | task1342 | task1391 | task1598 | |
| 10 full (F) | 16 F | 0.46 ± 0.01 | 0.63 ± 0.00 | 0.50 ± 0.00 | 0.55 ± 0.01 | 0.50 ± 0.00 | 0.49 ± 0.01 | 0.50 ± 0.01 | 0.50 ± 0.01 | 0.61 ± 0.01 | 0.47 ± 0.01 | 0.52 ± 0.06 |
| | 32 F | 0.30 ± 0.01 | 0.37 ± 0.00 | 0.31 ± 0.00 | 0.35 ± 0.00 | 0.34 ± 0.00 | 0.31 ± 0.00 | 0.33 ± 0.00 | 0.31 ± 0.00 | 0.38 ± 0.00 | 0.30 ± 0.00 | 0.33 ± 0.03 |
| | 64 F | 0.15 ± 0.00 | 0.15 ± 0.00 | 0.16 ± 0.00 | 0.17 ± 0.00 | 0.17 ± 0.00 | 0.16 ± 0.00 | 0.16 ± 0.00 | 0.16 ± 0.00 | 0.17 ± 0.00 | 0.15 ± 0.00 | 0.16 ± 0.01 |
| 50 full (F) | 16 F | 0.80 ± 0.02 | 0.82 ± 0.01 | 0.85 ± 0.01 | 0.90 ± 0.02 | 0.78 ± 0.01 | 0.95 ± 0.01 | 0.76 ± 0.01 | 0.75 ± 0.01 | 0.79 ± 0.01 | 0.82 ± 0.01 | 0.82 ± 0.06 |
| | 32 F | 0.65 ± 0.01 | 0.67 ± 0.01 | 0.72 ± 0.01 | 0.76 ± 0.02 | 0.65 ± 0.01 | 0.82 ± 0.02 | 0.66 ± 0.01 | 0.65 ± 0.01 | 0.67 ± 0.02 | 0.69 ± 0.00 | 0.69 ± 0.06 |
| | 64 F | 0.50 ± 0.01 | 0.52 ± 0.00 | 0.52 ± 0.00 | 0.55 ± 0.01 | 0.52 ± 0.01 | 0.62 ± 0.01 | 0.54 ± 0.01 | 0.51 ± 0.00 | 0.54 ± 0.01 | 0.57 ± 0.00 | 0.54 ± 0.03 |
| 100 full (F) | 16 F | 0.93 ± 0.02 | 0.90 ± 0.02 | 0.93 ± 0.01 | 0.91 ± 0.02 | 0.88 ± 0.03 | 0.98 ± 0.01 | 0.96 ± 0.01 | 0.78 ± 0.00 | 0.82 ± 0.00 | 0.93 ± 0.02 | 0.90 ± 0.06 |
| | 32 F | 0.87 ± 0.01 | 0.81 ± 0.01 | 0.85 ± 0.02 | 0.80 ± 0.01 | 0.79 ± 0.02 | 0.91 ± 0.00 | 0.90 ± 0.01 | 0.74 ± 0.01 | 0.70 ± 0.02 | 0.85 ± 0.02 | 0.82 ± 0.07 |
| | 64 F | 0.65 ± 0.04 | 0.69 ± 0.01 | 0.71 ± 0.01 | 0.67 ± 0.01 | 0.64 ± 0.01 | 0.76 ± 0.01 | 0.77 ± 0.01 | 0.67 ± 0.00 | 0.61 ± 0.00 | 0.75 ± 0.06 | 0.69 ± 0.06 |
| 500 full (F) | 16 F | 0.98 ± 0.04 | 0.98 ± 0.01 | 0.99 ± 0.01 | 1.00 ± 0.00 | 0.99 ± 0.00 | 0.96 ± 0.05 | 0.93 ± 0.10 | 0.94 ± 0.09 | 1.00 ± 0.00 | 0.99 ± 0.00 | 0.98 ± 0.05 |
| | 32 F | 0.92 ± 0.07 | 0.84 ± 0.20 | 0.92 ± 0.10 | 0.98 ± 0.02 | 0.97 ± 0.02 | 0.89 ± 0.08 | 0.82 ± 0.13 | 0.84 ± 0.11 | 0.99 ± 0.00 | 0.99 ± 0.02 | 0.92 ± 0.10 |
| | 64 F | 0.80 ± 0.00 | 0.67 ± 0.21 | 0.78 ± 0.11 | 0.90 ± 0.07 | 0.86 ± 0.08 | 0.76 ± 0.00 | 0.67 ± 0.00 | 0.70 ± 0.00 | 0.96 ± 0.03 | 0.99 ± 0.00 | 0.81 ± 0.13 |

Table 15: **Reconstruction error on random LoRAs** The error is larger in comparison to reconstructing trained (i.e., non-random) LoRAs in Table 14 for the corresponding compression methods.

| Model Type | Method Type | Tasks | | | | | | | | | | Average |
|---|---|---|---|---|---|---|---|---|---|---|---|---|
| | | task039 | task190 | task280 | task290 | task391 | task442 | task620 | task1342 | task1391 | task1598 | |
| | base | 24.44 | 1.60 | 19.13 | 39.22 | 10.27 | 35.46 | 7.85 | 6.22 | 17.82 | 38.87 | 20.09 |
| | lora | 95.00 | 86.00 | 99.00 | 93.67 | 94.33 | 74.88 | 74.40 | 26.68 | 95.00 | 50.32 | 78.93 |
| 10 full (F) | 32 F | 97.00 | 90.00 | 99.00 | 93.33 | 94.67 | 74.09 | 72.13 | 27.83 | 94.00 | 50.71 | 79.28 |
| | 64 F | 95.00 | 89.00 | 99.00 | 93.67 | 94.67 | 74.29 | 74.80 | 26.63 | 96.00 | 51.04 | 79.41 |
| 50 full (F) | 32 F | 96.00 | 88.00 | 99.00 | 93.67 | 92.33 | 72.30 | 75.97 | 29.89 | 94.00 | 45.68 | 78.68 |
| | 64 F | 98.00 | 89.00 | 99.00 | 93.67 | 93.33 | 72.74 | 76.50 | 29.33 | 96.00 | 45.71 | 79.33 |
| 100 full (F) | 32 F | 92.10 | 83.00 | 99.00 | 93.67 | 92.00 | 71.09 | 63.29 | 27.87 | 88.00 | 42.36 | 75.24 |
| | 64 F | 97.00 | 87.00 | 99.00 | 93.67 | 92.33 | 72.23 | 74.69 | 29.98 | 95.00 | 44.71 | 78.56 |
| 500 full (F) | 32 F | 68.92 | 43.00 | 87.00 | 91.67 | 90.67 | 70.08 | 51.16 | 14.40 | 83.00 | 41.97 | 64.19 |
| | 64 F | 93.50 | 78.00 | 91.00 | 92.33 | 90.33 | 72.55 | 57.49 | 15.44 | 85.00 | 42.31 | 71.80 |

Table 16: **Performance with convergence** In-Distribution **Rouge-L**

Table 17: Agreement Comparison. 100 LoRAs

| Configuration | | Agreement (%) |
|---|---|---|
| Base Model | | 83.015 |
| Uncompressed LoRAs | | 100.000 |
| **Joint Compression** | | |
| Diagonal | Rank 8 | 87.032 |
| | Rank 16 | 88.908 |
| | Rank 32 | 91.545 |
| | Rank 64 | 94.659 |
| Full | Rank 8 | 87.686 |
| | Rank 16 | 90.163 |
| | Rank 32 | 94.018 |
| | Rank 64 | 96.918 |

Table 18: Performance Comparison. 100 LoRAs

| Configuration | | Average Performance |
|---|---|---|
| Base Model | | 32.28 |
| Uncompressed LoRAs | | 48.32 |
| **Join Compression** | | |
| Diagonal | Rank 8 | 41.90 |
| | Rank 16 | 45.44 |
| | Rank 32 | 46.89 |
| | Rank 64 | 47.43 |
| Full | Rank 8 | 43.88 |
| | Rank 16 | 45.79 |
| | Rank 32 | 46.83 |
| | Rank 64 | 47.66 |

Table 19: Task-Based Performance Evaluation Across Different Models and Ranks

| Task | Base Model | LoRA | Diagonal R8 | Diagonal R16 | Diagonal R32 | Diagonal R64 |
|---|---|---|---|---|---|---|
| Causal Judgement | 57.47 | 64.37 | 55.17 | 58.62 | 58.62 | 58.62 |
| Date Understanding | 15.33 | 23.33 | 20.67 | 22.00 | 21.33 | 22.67 |
| Formal Fallacies | 51.33 | 56.00 | 52.67 | 52.67 | 53.33 | 54.67 |
| Hyperbaton | 6.67 | 68.00 | 57.33 | 63.33 | 67.33 | 68.00 |
| Logical Deduction (5 Objects) | 21.33 | 37.33 | 32.00 | 36.67 | 37.33 | 37.33 |
| Logical Deduction (7 Objects) | 12.67 | 44.00 | 31.33 | 42.67 | 44.67 | 45.33 |
| Movie Recommendation | 62.67 | 67.33 | 62.00 | 64.67 | 66.67 | 67.33 |
| Object Counting | 34.67 | 38.00 | 35.33 | 36.67 | 36.67 | 38.00 |
| Snarks | 50.00 | 61.54 | 53.85 | 56.41 | 58.97 | 57.69 |
| Temporal Sequences | 16.67 | 23.33 | 18.67 | 20.67 | 24.00 | 24.67 |
| **Average** | 32.88 | 48.32 | 41.90 | 45.44 | 46.89 | 47.43 |

Table 20: Task-Based Performance Evaluation Across Different Models and Ranks

| Task | Base Model | LoRA | Full R8 | Full R16 | Full R32 | Full R64 |
|---|---|---|---|---|---|---|
| Causal Judgement | 57.47 | 64.37 | 56.32 | 57.47 | 58.62 | 60.92 |
| Date Understanding | 15.33 | 23.33 | 19.33 | 22.00 | 22.67 | 22.67 |
| Formal Fallacies | 51.33 | 56.00 | 51.33 | 52.67 | 53.33 | 56.00 |
| Hyperbaton | 6.67 | 68.00 | 63.33 | 66.00 | 69.33 | 68.00 |
| Logical Deduction (5 Objects) | 21.33 | 37.33 | 35.33 | 36.00 | 35.33 | 37.33 |
| Logical Deduction (7 Objects) | 12.67 | 44.00 | 40.00 | 44.67 | 44.67 | 44.67 |
| Movie Recommendation | 62.67 | 67.33 | 63.33 | 65.33 | 67.33 | 67.33 |
| Object Counting | 34.67 | 38.00 | 35.33 | 36.67 | 37.33 | 37.33 |
| Snarks | 50.00 | 61.54 | 53.85 | 55.13 | 57.69 | 58.97 |
| Temporal Sequences | 16.67 | 23.33 | 20.67 | 22.00 | 22.00 | 23.33 |
| **Average** | 32.88 | 48.32 | 43.88 | 45.79 | 46.83 | 47.66 |

