# OpenReview forum: "Compress then Serve: Serving Thousands of LoRA Adapters with Little Overhead"
_ICML.cc/2025/Conference — ICML 2025 poster_

### Official Review · Reviewer_ZhJy · 2025-03-06

**Overall Recommendation:** 3

**Summary:**

This paper propose a method to efficiently serve numerous (thousands) of LoRA adapters for large language models. They propose a joint diagonalization-based compression method that significantly reduces storage and serving overhead while preserving model performance. To effectively scale the LoRA to the advertised thousands, they further propose a clustering strategy to solve the bottleneck from r. In their experiment, they train 1000 LoRA adapters to demonstrate their approach preserves most of the performance and achieves an improvement on throughput compared with the vLLM multi-LoRA solution.

**Claims And Evidence:**

Yes, claims made in the submission supported by clear and convincing evidence (experimental results).

**Essential References Not Discussed:**

No

**Experimental Designs Or Analyses:**

The proposed method is tested on a task suite (Wang et al., 2022) with over 1000 tasks but it is only tested on a Mistral model.

**Methods And Evaluation Criteria:**

Yes, the proposed method make sense for the problem or application at hand (achieving improvement on throughput and storage while preserving most of the performance).

**Other Comments Or Suggestions:**

No

**Other Strengths And Weaknesses:**

It might be difficult for adding a new LoRA adapters on the compressed old LoRAs (the trained up and down projections might be specific to the old LoRAs)

**Questions For Authors:**

How to route the different inputs to those LoRA adapters, do we need a classifier?

**Relation To Broader Scientific Literature:**

1. LoRA literature, while original LoRA considers single task scenario.
2. The proposed JD approach is related to model compression and merging
3. Efficient inference method such as vLLM.

**Theoretical Claims:**

The proposed JD method has notable limitations due to its reliance on a shared basis, potentially restricting its effectiveness with orthogonal or highly uncorrelated LoRA adapters. Corollary 1 indicates that when LoRA update matrices are orthogonal, reconstruction error can become significant. The approach implicitly assumes some level of similarity among adapters, which is not just a trade-off but a limitation in the method’s applicability to diverse scenarios.

---

> ### Author Rebuttal · Authors · 2025-04-01
>
> Thank you for the review. Please see our responses below.
>
> ---
>
> ## 1. Shared Basis and Clustering Approach
>
> Our method relies on a shared basis for each cluster. Although this approach implicitly assumes some similarity among LoRA adapters, the clustering strategy is designed to work effectively even when reconstruction errors (as measured in the L2 sense) are high. This is because the merging effect within clusters—akin to averaging weights—preserves high performance, as evidenced by our experiments on 1000 LoRAs across multiple languages.
>
> ## 2. Evaluation on Additional Model Architectures
>
> While our current evaluation is performed on a Mistral model, prior work (including the original *LoRA: Low-Rank Adaptation of Large Language Models* paper) shows that LoRA operates similarly across various transformer-based architectures. Nonetheless, we can run experiments on an additional LLM before the camera-ready version to further substantiate the generalizability of our approach.
>
> ## 3. Incorporating New LoRA Adapters
>
> Adding new LoRA adapters to an already compressed set can be challenging because the trained up and down projections are tailored to the existing adapters. Our recommendation is to rerun the joint compression algorithm when new LoRAs are introduced. As we mention in Section 6.5, this can be managed via batched cron jobs (e.g., on a daily schedule), where new LoRAs are initially compressed as individual clusters and then incorporated into the overall joint compression process.
>
> ## 4. Routing and Selection of LoRA Adapters
>
> The deployment framework is designed such that the server maintains the full collection of LoRA adapters. Each request includes the identifier of the specific LoRA to be used, thereby eliminating the need for an additional classifier to route inputs. This straightforward mechanism ensures that the correct adapter is selected for each task.
>
> ---
>
> We hope these clarifications address your concerns regarding the reliance on a shared basis, the evaluation on a single model, the integration of new LoRA adapters, and the routing mechanism for adapter selection.

---

### Official Review · Reviewer_tRKB · 2025-03-15

**Overall Recommendation:** 4

**Summary:**

This work focuses on a multi-LoRA serving system and significantly enhances throughput. The key approach involves compressing a collection of LoRA adapters to share a common low-rank space. This joint compression effectively reduces the total number of parameters during inference, leading to improved serving efficiency. Experimental results demonstrate that the end-to-end throughput is significantly improved while maintaining model performance.


### Update after Rebuttal ###

My concerns have been addressed by the responses. I keep my original score.

**Claims And Evidence:**

Yes, the method is evaluated based on both end-to-end throughput and performance, demonstrating its effectiveness.

**Essential References Not Discussed:**

No

**Experimental Designs Or Analyses:**

Both the thoughput and performance evaluation is sound.

**Methods And Evaluation Criteria:**

yes.

**Other Comments Or Suggestions:**

None

**Other Strengths And Weaknesses:**

- For the clustering strategy, how to better decide the optimial number of clusters?

- If the distribution of different LoRA models varies significantly, will it impact the effectiveness of joint compression?

**Questions For Authors:**

None

**Relation To Broader Scientific Literature:**

The work is highly related with multi-lora serving system. Notable literatures like S-LoRA and Punica.

**Theoretical Claims:**

I reviewed Theorem 1 and Corollary 1, and the results are valid, providing theoretical support for the proposed method.

---

> ### Author Rebuttal · Authors · 2025-04-01
>
> Thank you for the review. Please see our responses below.
>
> ---
>
> ## 1. Deciding the Optimal Number of Clusters
>
> Please see Section 6.5 for hyperparameter recommendations. Determining the optimal number of clusters does require some hyperparameter tuning; however, our experiments indicate that the method is not highly sensitive to this parameter. In **Appendix G**, we outline a practical tuning procedure that leverages the L2 reconstruction error from a single LoRA module to guide the selection of an appropriate number of clusters for the entire collection. This approach simplifies the tuning process and ensures robust performance.
>
> ## 2. Impact of LoRA Model Distribution on Joint Compression
>
> The motivation for introducing the clustering strategy was based on the observation that joint compression is considerably more sensitive to variations in the LoRA models when performed directly. In our experiments—covering 1000 LoRAs across diverse tasks and languages—we found that the inherent variability of the LoRA models can negatively impact compression effectiveness. By clustering the LoRAs, the impact of this diversity is significantly reduced, thereby stabilizing and enhancing the performance of the joint compression.
>
> ---
>
> We hope these clarifications address your concerns regarding the optimal clustering strategy and its role in mitigating the effects of LoRA diversity. We appreciate your insightful feedback.

---

> > ### Comment · Reviewer_tRKB · 2025-04-04
> >
> > Thank you for the responses that addressed my concerns. I will maintain my original positive score.

---

### Official Review · Reviewer_wRya · 2025-03-17

**Overall Recommendation:** 3

**Summary:**

The paper addresses the challenge of efficiently serving large numbers of LoRA adapters in real-time inference settings. Existing solutions require frequent loading and offloading of LoRAs due to limited GPU memory. The authors introduce a joint compression technique where multiple LoRAs are compressed into a shared basis with LoRA-specific scaling matrices. This reduces memory footprint and improves throughput.

**Claims And Evidence:**

1. "Throughput Improvement (1.6× Speedup) & Memory Efficiency". Supported.
2. "compressed LoRAs retain up to 99% of original performance." Supported in Figures 2 and 3.
3. "Compression Enhances Generalization" No.  There is no clear causal explanation for why compression might enhance generalization.

**Essential References Not Discussed:**

Related works that are enough for understanding.

**Experimental Designs Or Analyses:**

I think there are reasonable.

**Methods And Evaluation Criteria:**

1. For the evaluation part, the Figure 4, to compare Throughput, why not also compare the SOTA s-lora? Even s-lora needs to reload the lora adapters.
2. Could authors also clarify the evaluation hardware platform?
3. Could authors also clarify the overhead of compression?

**Other Comments Or Suggestions:**

No other Comments.

**Other Strengths And Weaknesses:**

Strengths: It proposes 1. Novel Compression Method for Serving LoRAs. 2. Clustering-based compression allows efficient inference.

Weaknesses: 1. While the method scales to 1000+ LoRAs, its feasibility for 100,000+ LoRAs is not explored, which could be relevant for large-scale commercial deployments.
2. Choosing optimal rank and number of clusters requires hyperparameter tuning, which may increase complexity in real-world deployments.

**Questions For Authors:**

1. For the evaluation part, the Figure 4, to compare Throughput, why not also compare the SOTA s-lora? Even s-lora needs to reload the lora adapters.
2. Could authors also clarify the evaluation hardware platform?
3. Could authors also clarify the overhead of compression?

**Relation To Broader Scientific Literature:**

1. It contributes to Model Compression and Matrix Factorization. It extends classical compression methods by introducing joint diagonalization (JD), which compresses multiple LoRAs simultaneously instead of handling them individually.
2. It integrates compression into LLM serving—unlike previous methods that focus on better memory allocation, scheduling, or kernel optimizations.

**Theoretical Claims:**

I think there are no issues.

---

> ### Author Rebuttal · Authors · 2025-04-01
>
> Thank you for the review. Please see our responses below.
>
> ---
>
> ## 1. Throughput Comparison and SOTA s-LoRA
>
> The vLLM multi-LoRA baseline in our experiments already incorporates advanced optimizations such as efficient scheduling and non-blocking CPU-GPU communication when swapping LoRAs, as well as techniques from S-LoRA. Consequently, the throughput comparisons in Figure 4 inherently reflect these state-of-the-art improvements without the need for an additional baseline comparison to S-LoRA.
>
> ## 2. Evaluation Hardware Platform
>
> Experiments were conducted on an **H100 80GB GPU** with memory consumption capped at 40%. This setting is intended to simulate scenarios where service providers might serve many LoRAs on more economical hardware with lower memory capacity than high-end GPUs.
>
> ## 3. Compression Overhead
>
> In **Figure 5** of the Appendix, we present detailed measurements of memory load, data transfer time, and forward-pass performance for both standard LoRA and our joint diagonalization approach (JD-LoRA). This analysis provides a clear view of the compression overhead and its impact on overall performance. Please note that the whole compression process can be done on CPU without interfering with serving of the LLMs. E.g. new LoRAs are temporarily compressed using SVD individually, and then a cron job, say, runs once a day, to compress the full set of LoRAs using joint compression.
>
> ## 4. Scalability to 100,000+ LoRAs
>
> While our current work scales to 1000+ LoRAs—currently the world’s largest open collection of LoRAs with documented training parameters—we believe that scaling to 100,000+ LoRAs is feasible in principle by scaling the number of clusters in our algorithm.
>
> ## 5. Hyperparameter Tuning
>
> Please see Section 6.5 for hyperparameter tuning. Although choosing the optimal rank and number of clusters requires some hyperparameter tuning, our experiments indicate that the method is not overly sensitive to these choices. In **Appendix G**, we describe a practical tuning procedure that leverages the reconstruction error (in the L2 sense) from a single LoRA module to efficiently determine appropriate hyperparameters for a collection of 1000 LoRAs, thereby reducing deployment complexity.
>
> ---
>
> We hope these clarifications address your concerns regarding throughput, hardware, compression overhead, scalability, and hyperparameter tuning. We appreciate your feedback and believe these additions significantly strengthen our work.

---

### Official Review · Reviewer_RG6N · 2025-03-17

**Overall Recommendation:** 4

**Summary:**

The authors propose a method that efficiently handle the problem of serving thousands of LoRA adapters for LLMs when apply on many tasks by compressing them into a shared basis with LoRA-specific scaling matrices. With number of LoRA become larger, to scale further, they use clustering-based compression, reducing memory usage while preserving performance. Their approach improves inference throughput by 1.6 times in vLLM, achieving 80% of the speed of a single LoRA while handling thousands of adapters efficiently.

**Claims And Evidence:**

+ The paper provides some theoretical analysis to understand the role of the joint diagonalization method and from that understand how it motivates the clustering approach. Their analysis point out that well-clustered in LoRAs make reconstruction error low and vice versa. They also conduct many experiments to convince their claims.

+ However, the paper does not show why certain LoRAs cluster well together (is it based on task similarity, weight structure?).

**Essential References Not Discussed:**

I see some prior works related to multi-adapter systems in LLM but are not refer in paper, such as [1], the author might need to compare their method with this one. Except that, I think the paper referred enough related works for understanding the key contributions of the paper.

[1] Chameleon: Adaptive Caching and Scheduling for Many-Adapter LLM Inference Environments. Arxiv 2024

**Experimental Designs Or Analyses:**

This paper propose a new method that help to compress many LoRAs adapters to save memory usage but not decrease too much in performance. To validate the benefits of proposed method in compression and performance, reconstruction error, they train LoRAs on 1000 natural instruction tasks. After that, they visualize clearly the  performance of compressed LoRAs relative to uncompressed ones. They also visualize the relation between reconstruction error and relative performance and see that with JD-clustering, reconstruction error is even less critical for performance. Therefore, I think the design of experiments in this paper is make sense to clarify the benefits of proposed method.

**Methods And Evaluation Criteria:**

I think the proposed methods and evaluation criteria are appropriate for the problem of multi-LoRAs serving. The joint diagonalization (JD) and clustering-based compression effectively address memory constraints and inference efficiency, while preserving the performance. The evaluation on 1000 LoRAs (for 1000 tasks) and throughput in vLLM aligns with real-world multi-LoRAs systems.

**Other Comments Or Suggestions:**

I do not have any suggestion.

**Other Strengths And Weaknesses:**

Strengths:
+ The paper proposed a novel method that can help compress multi-LoRAs efficiently without losing performance too much. This is very useful for edge systems.
+ They provide detailly the theory supporting for the method.
+ They conduct experiments on 1000 natural instruction tasks (1000 LoRAs adapters) to show the benefits of the method on compression, reconstruction error vs performance, and throughput, with all show good results.

Weaknesses:
+ While clustering improves compression efficiency, the paper lacks theoretical analysis on why certain LoRAs cluster well.
+ They should provide more on whether clustering LoRAs in multi-domain cases (LoRAs are trained on different domains) can effect the performance to strengthen the findings.
+ I think task diversity is a little bit limited.

**Questions For Authors:**

I do not have any question.

**Relation To Broader Scientific Literature:**

The paper improves LoRA scalability by introducing joint diagonalization (JD) and compression based on clustering, which help reduce memory overhead while preserving performance. It builds on LoRA (Hu et al., 2021) and multi-LoRA inference (S-LoRA, Sheng et al., 2023) but surpasses them by enabling efficient serving of thousands of adapters. Inspired by SVD-based compression (Meng et al., 2024), it groups (through clustering) and compresses LoRAs jointly, improving throughput 1.6× over vLLM without decreasing too much in performance compared to uncompressed LoRAs. This work bridges PEFT, model compression, and scalable inference, open for future optimizations in multi-LoRA systems.

**Theoretical Claims:**

I think the proofs for all theorems in the paper is correct and make sense.

---

> ### Author Rebuttal · Authors · 2025-04-01
>
> Thank you for the review. Please see our responses below.
>
> ---
>
> ## 1. Clustering Basis: Weight Structure vs. Task Similarity
>
> Our approach relies solely on the LoRA weights, meaning that the clustering is driven by the intrinsic weight structure rather than an explicit measure of task similarity. While we recognize that the link between weight structure and task similarity is an area ripe for further exploration, our focus is on how these weight patterns facilitate compression.
>
> ## 2. Theoretical Analysis of Clustering Behavior
>
> Our theoretical section argues that a tight clustering of LoRAs is not a prerequisite for success. Indeed we expect (and observe) very large reconstruction errors (in the L2 sense) for the LoRA weight matrices.
>
> Instead, we argue that in addition to simple weight matrix recovery, there is also a very important effect of our approach more akin to merging of LoRAs by averaging their weights (see the later parts of the theoretical section). We believe this explains why the observed LLM outputs do not degrade at all until the 2-norm reconstruction error becomes very large indeed (see Figure 3). This is intuitive given the inner product structure of the attention mechanism in transformers.
>
> Nonetheless, as demonstrated in the merging literature, combining too many LoRAs eventually leads to degradation. This insight motivated our strategy of performing joint diagonalization on smaller, independent clusters. Another indication for the diversity is the low results of TIES merging (Appendix H.3), which might have succeeded otherwise.
>
> ## 3. Multi-Domain Clustering and Task Diversity
>
> Our experiments include LoRAs trained across multiple domains and languages, which reinforces the importance of robust clustering in multi-domain settings. The diversity of our task set is further underscored by **Table 3** in the Appendix, which details all 1000 tasks sourced from *Super-Natural Instructions: Generalization via Declarative Instructions on 1600+ NLP Tasks*. Despite the high reconstruction errors—indicating that the LoRAs are not overly similar—the clustering process effectively exploits the underlying structure, leading to improved compression efficiency.
>
> ---
>
> We hope these clarifications adequately address your concerns regarding the clustering rationale, the theoretical underpinnings of our approach, and the diversity of our task set. We appreciate your feedback and are open to any further suggestions.

---

> > ### Comment · Reviewer_RG6N · 2025-04-05
> >
> > Thank you for the responses. I am satisfied with the rebuttal and increase my rating.

---

### Official Review · Reviewer_agtv · 2025-03-19

**Overall Recommendation:** 3

**Summary:**

This paper introduces a novel framework for efficiently managing a large set of LoRA adapters. The authors present a joint diagonalization based (JD) algorithm in both a full and a diagonal variant, which compresses multiple LoRA weights by decomposing them into a shared basis and adapter specific scaling matrices. For scenarios with a large number of adapters, the paper proposes a clustered JD algorithm to enhance scalability. In addition, the paper provides a theoretical analysis of the reconstruction error for JD full. Experiments on 1000 natural instruction tasks demonstrate that the proposed compression strategy can improve throughput while preserving model performance.

**Claims And Evidence:**

- The submission’s claims regarding performance preservation and throughput enhancement are supported by experimental findings involving varying numbers of LoRA adapters, in some cases up to 1,000.
- Results from the JD-cluster, which integrates a large number of LoRA adapters, demonstrate the scalability of the proposed methods.
- Theoretical analysis of error bounds on the reconstruction error are provided.

**Essential References Not Discussed:**

N/A

**Experimental Designs Or Analyses:**

- The authors employ the JD-cluster algorithm with $k$ clusters to compress a large number of $n$ LoRAs. However, the study does not provide a detailed analysis of how  $k$ varies as $n$ increases.
- In addition, the comparative evaluation is limited to the original uncompressed LoRAs and an SVD-based compression approach. ncluding additional baselines would help to strengthen the contributions of the proposed method.

**Methods And Evaluation Criteria:**

- The proposed JD algorithm is both novel and grounded in a sound theoretical framework. Employing JD-cluster to compress a large number of LoRAs is appropriate for the problem at hand.
- However, while the authors conduct experiments on 1,000 natural instruction tasks, it remains unclear whether these tasks and their corresponding LoRA adapters are sufficiently diverse. It is possible that correlated tasks and LoRA weights inflate the observed performance improvements, particularly given that the compressed LoRAs can surpass the originals (as evidenced in Figure 2). To address this concern, it would be valuable to provide additional analysis of LoRA weight diversity (e.g., through suitable visualizations) and to test the method on a wider variety of tasks.
- Additionally, the reliance on the Mistral-7B-Instruct-v0 model might constrain the scope of the evaluation. Exploring the applicability of the proposed approach on other model architectures would better demonstrate its generalizability and practical utility.

**Other Comments Or Suggestions:**

N/A

**Other Strengths And Weaknesses:**

- The paper offers practical implementations that enhance the clarity and applicability of the proposed method.
- The experimental results are presented in a manner that can be somewhat difficult to follow.

**Questions For Authors:**

1. Could the authors provide a quantitative or qualitative analysis of the LoRA weight diversity (e.g., through visualizations or statistical measures)?  If the weights appear largely similar, it might raise concerns about the method’s capacity for broad generalization; conversely, robust evidence of weight diversity would strengthen the paper’s claims.
2. Could you elaborate on how does number of clusters $k$ vary with the number of LoRA adapters $n$, and how sensitive is the algorithm’s performance to different values of $k$?
3. Could the proposed method be extended or adapted to other parameter-efficient fine-tuning approaches, such as prompt-tuning?

**Relation To Broader Scientific Literature:**

- The paper addresses an important challenge in deploying large-scale machine learning models, particularly in real-time applications where serving numerous LoRA adapters efficiently is paramount. This concern aligns with a growing body of work focused on model compression and optimization, such as knowledge distillation, network pruning, and factorization-based methods.
- The proposed JD method is a novel approach that broadens current model compression strategies. This innovation has the potential to advance both theoretical understanding and practical applications in the domain of efficient model adaptation, further bridging the gap between research and real-world deployment.

**Theoretical Claims:**

The provided proofs and theoretical claims appear to be correct and internally consistent. No major flaws or inconsistencies were identified.

---

> ### Author Rebuttal · Authors · 2025-04-01
>
> Thank you for the review. Please see our responses below.
>
> ---
>
> ## 1. LoRA Adapter Diversity and Task Coverage
>
> Please see **Table 3** in the Appendix, which lists all 1000 tasks drawn from *Super-Natural Instructions: Generalization via Declarative Instructions on 1600+ NLP Tasks*. This dataset covers a wide range of tasks, including multiple languages, and provides detailed insight into our data collection protocol.
>
> Moreover, the relatively high reconstruction error observed in our experiments indicates that the LoRA adapters are not overly similar. Another indication for the diversity is the low results of TIES merging (Appendix H.3), which might have succeeded otherwise. The significant gains from clustering further suggest a meaningful clustered structure among tasks.
>
> In response to your suggestion, we will add an analysis of cosine similarity between LoRA adapters prior to compression. If you have specific similarity thresholds or metrics in mind, we welcome your recommendations.
>
> ## 2. Evaluation on Additional Model Architectures
>
> We trained an unprecedented number of LoRA adapters on the Mistral-7B-Instruct-v0 model to rigorously validate our approach.
>
> While we acknowledge that additional experiments on more architectures could provide further insights, our current evaluation robustly supports our claims, and we hope you consider this work as a meaningful step forward. Still, we aim to extend our evaluation to include results from another model architecture, which further demonstrates the generalizability and practical utility of our method.
>
> ## 3. Analysis of Clustering Parameter Sensitivity
>
> In **Section G** of the Appendix and Figure 6, we study how the number of clusters \(K\) varies with the number of LoRA adapters \(N\) (with experiments conducted for \(N=100\) and \(N=500\)). This analysis helps clarify the sensitivity of the algorithm’s performance with respect to different values of \(K\). Further, see Section 6.5 about selecting hyperparameters.
>
> ## 4. Comparative Evaluation and Extensions to Other Methods
>
> In addition to the original uncompressed LoRAs and the SVD-based compression baseline, we compare to Ties-Merging in Appendix H.3 (see Table 7). Regarding potential extensions, we acknowledge that other parameter-efficient fine-tuning methods such as prompt-tuning might also benefit from exploiting shared structures across tasks. We agree that exploring this possibility is an interesting direction for future work.
>
> ---
>
> We hope that these clarifications and our planned experiments sufficiently address your concerns.

---

### Official Review · Reviewer_ikf9 · 2025-03-21

**Overall Recommendation:** 4

**Summary:**

This paper considers the problem of serving a large amount of LoRA adapters for the same LLM. This is a very practical scenarios where each LoRA adapter correspond to one specific task. If one naively switches between different adapters, the throughput will degrade a lot when the number of adapters is large. So in this paper, the authors propose to compress all LoRA adapters together by finding a share basis for all. The proposed method significantly reduce the serving overhead when the number of adapters is up to 1000.

Overall I think the method is very smart and direct. The throughput gain over the naive solution is very signifiant.

**Claims And Evidence:**

Yes

**Essential References Not Discussed:**

NA

**Experimental Designs Or Analyses:**

Yes

**Methods And Evaluation Criteria:**

Yes

**Other Comments Or Suggestions:**

NA

**Other Strengths And Weaknesses:**

NA

**Questions For Authors:**

NA

**Relation To Broader Scientific Literature:**

This paper provides useful insights on how to serve a large number of lora adapters in practice.

**Theoretical Claims:**

Yes

---

> ### Author Rebuttal · Authors · 2025-04-01
>
> We thank the reviewer for recognizing the novelty of our method and its throughput gains over naive solutions.
>
> We emphasize that our joint compression approach effectively addresses the GPU memory constraints and the overhead associated with loading and unloading LoRA adapters and is accompanied by both theoretical and empirical validation.
>
> If there is any additional clarification or experiment we can provide to further strengthen your confidence in our approach, please let us know. Otherwise, we would greatly appreciate it if you could consider revising your score upward based on our response.
>
> Thank you again for your valuable feedback.

---

### Decision · Program_Chairs · 2025-05-01

**Decision:**

Accept (poster)

**Comment:**

In the paper, the authors proposed methods and algorithms to handle a large set of LoRA adapters for LLMs. All the reviewers are positive about the contributions of the papers, including: (1) the novelty of the joint diagonalization based and clustering-based compression algorithms that have solid theoretical guarantee; (2) the experiments are sufficiently extensive and appropriate to evaluate the performance of the proposed algorithms; (3) the writing and presentation of the paper are good. After the rebuttal, most of the remaining concerns of the reviewers were addressed, and all the reviewers are happy with the current stage of the paper.

I believe the contributions and originality of the paper are sufficient for acceptance at ICML. Therefore, I recommend accepting it in its current form. However, I encourage the authors to address the reviewers’ suggestions and integrate their feedback into the camera-ready version of their paper.